# Convergence Theorems for Entropy-Regularized and Distributional Reinforcement Learning

**Yash Jhaveri**[*]
Rutgers University–Newark

**Harley Wiltzer**[*]
Mila–Québec AI Institute
McGill University

**Patrick Shafto**
Rutgers University–Newark

**Marc G. Bellemare**[†]
Mila–Québec AI Institute
McGill University

**David Meger**
Mila–Québec AI Institute
McGill University

## Abstract

In the pursuit of finding an optimal policy, reinforcement learning (RL) methods generally ignore the properties of learned policies apart from their expected return. Thus, even when successful, it is difficult to characterize which policies will be learned and what they will do. In this work, we present a theoretical framework for policy optimization that guarantees convergence to a particular optimal policy, via vanishing entropy regularization and a *temperature decoupling gambit*. Our approach realizes an interpretable, diversity-preserving optimal policy as the regularization temperature vanishes and ensures the convergence of policy derived objects–value functions and return distributions. In a particular instance of our method, for example, the realized policy samples all optimal actions uniformly. Leveraging our temperature decoupling gambit, we present an algorithm that estimates, to arbitrary accuracy, the return distribution associated to its interpretable, diversity-preserving optimal policy.

## 1 Introduction

In generic Markov Decision Processes (MDPs), many optimal policies exist. Thus, while certain policy optimization approaches can ensure convergent approximation to an optimal policy, they do not have control over which states these policies will visit, which actions they will play, or which long-term returns they can achieve. Indeed, the non-uniqueness of optimal policies renders any discussion of the properties of an optimal policy ambiguous, beyond its expected value.

A partial remedy to this problem is to regularize the RL objective in order to induce uniqueness. One popular approach to regularization is to penalize the value of a policy according to its KL divergence to a reference policy $\pi^{\text{ref}}$. This branch of RL is known as entropy-regularized RL (ERL). In ERL, for any positive regularization weight $\tau$ (also known as *temperature*), one and only one policy is optimal. Moreover, in a tabular MDP, $\tau$-optimal policies and their derived objects (value functions, occupancy measures, and return distributions) converge to classically optimal policies and their derived objects. However, beyond tabular MDPs, the evolution of $\tau$-optimal quantities, as a function of the temperature, is not well understood. Thus, as we decay the temperature to zero, we are, in some sense, back to where we started: living in ambiguity.

In this work, we introduce a *temperature decoupling gambit*, through which we can guarantee the convergence of resulting policies and their derived objects in the vanishing temperature limit.

---

[*]Equal contribution. Correspondence to `yash.jhaveri@rutgers.edu`, `wiltzerh@mila.quebec`.
[†]CIFAR AI Chair.

Much like how a gambit in chess sacrifices an immediate and shallow proxy of the objective (e.g., material count) for a long term positional advantage, the temperature decoupling gambit plays notably *suboptimal* policies for the $\tau$-ERL objective to ensure convergence to RL optimality as $\tau \to 0$. This scheme entails estimating action-values under a target regularization temperature while playing policies with an amplified temperature. Furthermore, we characterize this limiting policy as a modification of the reference policy which "filters out" suboptimal actions. Even when $\tau$-optimal policies converge in the vanishing temperature limit (such as in tabular MDPs), the limiting policy produced by the temperature decoupling gambit is distinct from the limiting policy found otherwise. The limiting policy found via our gambit preserves, quantifiably, more state-wise action diversity. Moreover, we show that this limiting policy achieves a notion of *reference-optimality* for RL, characterized by a new Bellman-like equation, whose unique fixed point upper bounds the (RL) performance of $\tau$-optimal policies in general.

Our analysis additionally sheds light on the convergence of return distributions—the central objects of study in distributional RL (DRL) [6]. While optimal policies achieve the same return in expectation, they may vary drastically in other statistics, such as variance. In safety-critical applications, for example, understanding the distribution over returns is crucial. DRL provides techniques for estimating return distributions, primarily based on distributional dynamic programming methods which generalize dynamic programming approaches for estimating expected returns. However, it is well-known that existing distributional methods do not produce convergent iterates in the control setting [5]. Leveraging our convergence results for policies in ERL, we define the first algorithm for accurately estimating a reference-optimal return distribution, the return distribution associated to the interpretable, diverse policy realized by the temperature decoupling gambit.

## 2 Preliminaries

Given a Borel set $\mathsf{S} \subset \mathbb{R}^n$, for some $n \in \mathbb{N}$, we let $M(\mathsf{S})$ and $M_b(\mathsf{S})$ denote the space of Borel measurable and bounded Borel measurable functions on $\mathsf{S}$ respectively. We let $\mathscr{P}(\mathsf{S})$ denoted the space of Borel probability measures on $\mathsf{S}$. From now on, measurability will always be with respect to Borel sets. Moreover, for any $\rho \in \mathscr{P}(\mathsf{Y})$ with $\mathsf{Y} \subset \mathbb{R}^m$ and any measurable function $f : \mathsf{Y} \to \mathsf{S}$, the *push-forward of $\rho$ by $f$* is $f_\#\rho := \rho \circ f^{-1} \in \mathscr{P}(\mathsf{S})$. Here $f^{-1}$ is the preimage of $f$.

We single out two particular functions. The function $\mathtt{proj}^{\mathsf{Y}_k} : \mathsf{Y}_1 \times \cdots \times \mathsf{Y}_n \to \mathsf{Y}_k$ defined by $\mathtt{proj}^{\mathsf{Y}_k}(y_1, \ldots, y_k, \ldots, y_n) := y_k$ is the *projection function* of $\mathsf{Y}^n$ onto $\mathsf{Y}_k$. We note that the push-forward of the projection map is marginalization: $\nu^\mu := \mathtt{proj}^{\mathsf{Y}}_\# \mu$ is the $\mathsf{Y}$-marginal of $\mu \in \mathscr{P}(\mathsf{Y} \times \mathsf{Z})$. The *bootstrap function* $\mathsf{b}_{a,b} : \mathbb{R} \to \mathbb{R}$ is defined by $\mathsf{b}_{a,b}(z) := a + bz$ from [6].

Our analysis works with conditional distributions, which we formalize as probability kernels, as well as a tensor-product notation constructing product measures and for disintegrating product measures. For any $\mathsf{Y} \subset \mathbb{R}^m$ and $\mathsf{Z} \subset \mathbb{R}^n$, the space of (Borel) *probability kernels* from $\mathsf{Y}$ to $\mathsf{Z}$, denoted $\mathsf{K}(\mathsf{Y}, \mathscr{P}(\mathsf{Z}))$, is the set of all indexed measures $\lambda$ for which $y \mapsto \lambda_y(\mathsf{S})$ is measurable for each $\mathsf{S} \in \mathscr{B}(\mathsf{Z})$, the Borel subsets of $\mathsf{Z}$. Given $\lambda \in \mathsf{K}(\mathsf{Y}, \mathscr{P}(\mathsf{Z}))$ and $\rho \in \mathscr{P}(\mathsf{Y})$, the *generalized product measure* $\lambda_- \otimes \rho \in \mathscr{P}(\mathsf{Y} \times \mathsf{Z})$ is defined as follows:

$$\int \phi \, \mathrm{d}(\lambda_- \otimes \rho) := \int \left[ \int \phi(y, z) \, \mathrm{d}\lambda_y(z) \right] \mathrm{d}\rho(y) \quad \forall \phi \in M(\mathsf{Y} \times \mathsf{Z}).$$

Additionally, we can disintegrate any $\mu \in \mathscr{P}(\mathsf{Y} \times \mathsf{Z})$ as a generalized product between either of its marginals and the induced conditional probabilities:

$$\mu = \pi^\mu_- \otimes \nu^\mu \quad \text{where} \quad \nu^\mu := \mathtt{proj}^{\mathsf{Y}}_\# \mu \quad \text{and} \quad \pi^\mu \in \mathsf{K}(\mathsf{Y}, \mathscr{P}(\mathsf{Z})).$$

An important subset of $\mathsf{K}(\mathsf{Y}, \mathscr{P}(\mathsf{Z}))$ consists of those kernels with bounded $p$th moments,

$$\overline{\mathsf{K}}^p(\mathsf{Y}, \mathscr{P}(\mathsf{Z})) := \left\{ \lambda \in \mathsf{K}(\mathsf{Y}, \mathscr{P}(\mathsf{Z})) : \sup_{y \in \mathsf{Y}} \int |z|^p \, \mathrm{d}\lambda_y(z) < \infty \right\} \quad \text{for} \quad p \in [1, \infty),$$

which can be metrized as complete metric spaces. In this work, we consider their metrization via the following metrics based on the Wasserstein metrics [40] $d_p$,

$$\overline{d}_p(\lambda, \lambda') := \sup_y d_p(\lambda_y, \lambda'_y) \quad \text{and} \quad d_{p;q,\omega}(\lambda, \lambda') := \left( \int d_p(\lambda_y, \lambda'_y)^q \, \mathrm{d}\omega(y) \right)^{1/q}, \qquad (2.1)$$

where $p, q \in [1, \infty)$ and $\omega \in \mathscr{P}(\mathsf{Y})$. These metrize topologies on $\overline{\mathsf{K}}^p(\mathsf{Y}, \mathscr{P}(\mathsf{Z}))$ akin to the weak topology on probability measures with finite $p$th moments.

## 2.1 Markov Decision Processes and Reinforcement Learning

A discounted MDP is a five-tuple $(\mathsf{X}, \mathsf{A}, P, r, \gamma)$. Here $\mathsf{X} \subset \mathbb{R}^m$ is the *state space*, $\mathsf{A} \subset \mathbb{R}^n$ is the *action space*, $r \in M_b(\mathsf{X} \times \mathsf{A})$ is the *reward function*, and $\gamma \in (0, 1)$ is the *discount factor*.[1]

Central to RL are policies. A *policy* is a probability kernel $\pi \in \mathsf{K}(\mathsf{X}, \mathscr{P}(\mathsf{A}))$. Policies induce state transition kernels $\hat{P}^\pi$ as well as a state-action transition kernels $\check{P}^\pi$, given by

$$\hat{P}^\pi_x := \texttt{proj}^\mathsf{X}_\#(P_{x,\_} \otimes \pi_x) \in \mathscr{P}(\mathsf{X}) \quad \text{and} \quad \check{P}^\pi_{x,a} := \pi_\_ \otimes P_{x,a} \in \mathscr{P}(\mathsf{X} \times \mathsf{A}),$$

respectively. Therefore, policies yield sequences of states as well as state-action pairs, labeled $(S^\pi_t)_{t \geq 0}$ and $(X^\pi_t, A^\pi_t)_{t \geq 0}$ respectively, whose sequences of laws $(\nu^\pi_t)_{t \geq 0}$ and $(\mu^\pi_t)_{t \geq 0}$ are given by

$$\nu^\pi_{t+1} := \hat{P}^\pi \nu^\pi_t \quad \text{with} \quad \nu^\pi_0 := \nu_0 \quad \text{and} \quad \mu^\pi_{t+1} := \check{P}^\pi \mu^\pi_t \quad \text{with} \quad \mu^\pi_0 := \pi_\_ \otimes \nu_0$$

for some $\nu_0 \in \mathscr{P}(\mathsf{X})$. Given $\nu_0 \in \mathscr{P}(\mathsf{X})$, the long-term behavior of any policy $\pi$ can be encoded via its (discounted, state-action) *occupancy measure* $\mu^\pi$, the set of which we denote by $\mathscr{O}(\nu_0)$,

$$\mathscr{O}(\nu_0) := \left\{ \mu^\pi \in \mathscr{P}(\mathsf{X} \times \mathsf{A}) : \mu^\pi := (1 - \gamma) \sum_{t \geq 0} \gamma^t \mu^\pi_t \text{ for some } \pi \in \mathsf{K}(\mathsf{X}, \mathscr{P}(\mathsf{A})) \right\}.$$

Policies also induce *return distribution functions* $\zeta^\pi \in \mathsf{K}(\mathsf{X} \times \mathsf{A}, \mathscr{P}(\mathbb{R}))$ and $\eta^\pi \in \mathsf{K}(\mathsf{X}, \mathscr{P}(\mathbb{R}))$,

$$\zeta^\pi_{x,a} := \text{law}\left( \sum_{t \geq 0} \gamma^t r(X^\pi_t, A^\pi_t) \,\bigg|\, X^\pi_0 = x, A^\pi_0 = a \right) \quad \text{and} \quad \eta^\pi_x := \texttt{proj}^\mathbb{R}_\#(\zeta^\pi_{x,\_} \otimes \pi_x)$$

whose means, the *action-value function* $q^\pi \in M_b(\mathsf{X} \times \mathsf{A})$ and the *value function* $v^\pi \in M_b(\mathsf{X})$,

$$q^\pi(x, a) := \mathbf{E}_{Z \sim \zeta^\pi_{x,a}}[Z] \quad \text{and} \quad v^\pi(x) := \mathbf{E}_{G \sim \eta^\pi_x}[G],$$

lead to the RL objective: find a $\pi^\star \in \mathsf{K}(\mathsf{X}, \mathscr{P}(\mathsf{A}))$ such that $q^{\pi^\star} \geq q^\pi$ for all $\pi$. Such a policy is called *optimal*. Generally, many policies are optimal. However, their associated action-value functions are identical (see [32]). We denote this optimal action-value function by $q^\star$.

## 2.2 Entropy-Regularized Reinforcement Learning

In ERL, the value of a policy is penalized by how far it diverges from a fixed *reference policy* $\pi^{\mathsf{ref}} \in \mathsf{K}(\mathsf{X}, \mathscr{P}(\mathsf{A}))$. In particular, the $\tau$-ERL problem with *temperature* $\tau > 0$ is

$$\sup_{\mu^\pi \in \mathscr{O}(\nu_0)} \mathcal{J}_\tau(\mu) \quad \text{where} \quad \mathcal{J}_\tau(\mu) := \int r \, \mathrm{d}\mu - \tau \mathcal{R}(\mu) \text{ and } \mathcal{R}(\mu) := \int \mathrm{KL}(\pi^\mu_x \,\|\, \pi^{\mathsf{ref}}_x) \, \mathrm{d}\nu^\mu(x).$$

When $\tau = 0$, we recover the linear programming formulation of the (expected-value) RL objective. In ERL, the regularizer $\mathcal{R}$ is strictly convex. Thus, $\mathcal{J}_\tau$ ia strictly concave and its maximizer unique.[2]

**Lemma 2.1.** *The functional* $\mathcal{R} : \mathscr{P}(\mathsf{X} \times \mathsf{A}) \to \mathbb{R}$ *is strictly convex.* [Proof]

Given Lemma 2.1, one might hope that the well-posedness of $\tau$-ERL could be realized through simple, yet power methods like the direct method in the calculus of variations. However, outside the tabular case, this is unclear, for many reasons, the first of which is that $M_b(\mathsf{X} \times \mathsf{A})$ is not separable.

The well-posedness of $\tau$-ERL, however, can be established through other means. In particular, in $\tau$-ERL, only one optimal policy exists, and it is characterized as a Boltzmann–Gibbs (BG) policy.

**Definition 2.2.** Let $q \in M(\mathsf{X} \times \mathsf{A})$ and $\tau > 0$. We denote the *Boltzmann-Gibbs policy associated to q and $\tau$* by $\mathcal{G}_\tau q$, and it is characterized by

$$\mathrm{d}(\mathcal{G}_\tau q)_x(a) := e^{(q(x,a) - (\mathcal{V}_\tau q)(x))/\tau} \, \mathrm{d}\pi^{\mathsf{ref}}_x(a) \quad \text{with} \quad (\mathcal{V}_\tau q)(x) := \tau \log \int e^{q(x,a)/\tau} \, \mathrm{d}\pi^{\mathsf{ref}}_x(a).$$

We note that $(\mathcal{G}_\tau q)_x$ is well-defined if and only if $(\mathcal{V}_\tau q)(x) \in \mathbb{R}$.

---

[1] We expect many of our results can be extended to Polish spaces.

[2] The only work we are aware of that establishes a comparable result is [28]. However, their result is on tabular MDPs and establishes convexity on $\mathscr{O}(\nu_0)$, not on all of $\mathscr{P}(\mathsf{X} \times \mathsf{A})$.

More specifically, it is well-known that the optimal policy of $\tau$-ERL is the BG policy associated to the unique fixed point $q_\tau^\star$ of the *soft Bellman optimality operator* $\mathcal{B}_\tau^\star : M(\mathsf{X} \times \mathsf{A}) \to M(\mathsf{X} \times \mathsf{A})$,

$$(\mathcal{B}_\tau^\star q)(x,a) := r(x,a) + \gamma \int (\mathcal{V}_\tau q)(x')\, \mathrm{d}P_{x,a}(x').$$

(See Lemma A.7.) The following theorem summarizes the well-posedness of $\tau$-ERL.

**Theorem 2.3.** *Let $\tau > 0$. The policy $\pi^{\tau,\star} := \mathcal{G}_\tau q^\star$ is optimal, and uniquely so. More precisely, for all $\nu_0, \nu_0' \in \mathscr{P}(\mathsf{X})$, we have that $\arg\max_{\mathscr{O}(\nu_0)} \mathcal{J}_\tau = \pi^{\tau,\star} = \arg\max_{\mathscr{O}(\nu_0')} \mathcal{J}_\tau.$*  [Proof]

In Appendix A, we prove Theorem 2.3 as well as a collection of supporting and related results that generalize well-known results in tabular MDPs. We include them for completeness.

In the remainder of this work, we study the evolution of $\tau$-optimal objects as $\tau$ vanishes. In the tabular regime, where $M_b(\mathsf{X} \times \mathsf{A})$ is separable, one can establish the existence and uniqueness of a $\tau$-optimal occupancy measure: $\mu_\tau^\star$. Furthermore, under a compatibility assumption, one can prove that the limit of the sequence $(\mu_\tau^\star)_{\tau>0}$ as $\tau$ vanishes exists and is unique as well.

**Assumption 2.4.** The intersection of $\{\arg\sup_{\mathscr{O}(\nu_0)} \mathcal{J}_0\}$ and $\{\mathcal{R} < \infty\}$ is nonempty.

Assumption 2.4 asks that our regularizer isn't identically $+\infty$ on the set of optimal policies. Without such an assumption, $\tau$-ERL and RL have no meaningful relationship, as we shall see in Section 3.

**Theorem 2.5.** *Suppose that $r \in M_b(\mathsf{X} \times \mathsf{A})$ and that $\mathsf{X} \times \mathsf{A}$ is finite. For every $\tau > 0$, let $\mu_\tau^\star$ be the maximizer of $\mathcal{J}_\tau$ over $\mathscr{O}(\nu_0)$. If Assumption 2.4 holds, the sequence $(\mu_\tau^\star)_{\tau>0}$ has a unique setwise limit as $\tau$ tends to zero. This limit $\mu_0^\star$ is the minimizer of $\mathcal{R}$ over $\arg\sup_{\mathscr{O}(\nu_0)} \mathcal{J}_0.$*  [Proof]

Consequently, in the tabular setting, the sequence $(\pi^{\tau,\star})_{\tau>0}$ has a unique limit.

**Remark 2.6.** Even if Theorem 2.5 could be extended to hold true in continuous MDPs, occupancy measure convergence *does not* guarantee policy convergence, outside of the tabular setting. Theorem 2.5 is a statement about a sequence of *joint distributions*. A policy convergence statement would be one about a sequence of conditional distributions (i.e., probability kernels). In general, the convergence of a sequence of joint distributions does not imply the convergence of the associated sequence of conditional distributions with respect to a fixed marginal (see, e.g., [7, Example 10.4.24]). While it is possible that the structure $\mathscr{O}(\nu_0)$ permits a type of policy convergence, we are unaware of any such result for continuous MDPs.

## 3 Convergence to Optimality: The Temperature Decoupling Gambit

While ERL has a unique solution, this identifiability comes at a cost with respect to RL: the resulting policy is suboptimal for RL. In this section, we analyze vanishing-temperature limits in $\tau$-ERL. Our main results for this section—Theorems 3.9 and 3.10—show that policies and their return distributions converge under the scheme of Definition 3.7 to interpretable, optimal limits as $\tau \to 0$.

To understand the ways in which $\tau$-ERL converges to RL, we define a (new) $\pi^{\mathsf{ref}}$-sensitive variant of the Bellman optimality operator, the *Bellman reference-optimality operator*. We call its unique fixed point the *reference-optimal action-value function*.

**Lemma 3.1.** *Let $r \in M_b(\mathsf{X} \times \mathsf{A})$, $\gamma < 1$, and $\mathcal{B}_{\mathsf{ref}}^\star : M(\mathsf{X} \times \mathsf{A}) \to M(\mathsf{X} \times \mathsf{A})$ be defined by*

$$(\mathcal{B}_{\mathsf{ref}}^\star q)(x,a) := r(x,a) + \gamma \int \operatorname{ess\,sup}_{\pi_{x'}^{\mathsf{ref}}} q(x',\cdot)\, \mathrm{d}P_{x,a}(x').$$

*Then $\mathcal{B}_{\mathsf{ref}}^\star$ is a contraction on $M_b(\mathsf{X} \times \mathsf{A})$. Thus, it has a unique fixed point $q_{\mathsf{ref}}^\star$.*  [Proof]

Generally, $q_{\mathsf{ref}}^\star$ is distinct from $q^\star$. Yet, ERL recovers $q_{\mathsf{ref}}^\star$ in the vanishing temperature limit.

**Theorem 3.2.** *We have that $q_\tau^\star \to q_{\mathsf{ref}}^\star$ monotonically as $\tau \to 0$.*  [Proof]

Theorem 3.2 implies that optimal policies, in general, cannot be recovered by taking vanishing temperature limits in ERL. We formalize a notion of *reference-optimality* to highlight this distinction.

**Definition 3.3.** A policy $\pi \in \mathsf{K}(\mathsf{X}, \mathscr{P}(\mathsf{A}))$ is said to be *reference-optimal* (against $\pi^{\mathsf{ref}}$) if $q^\pi \geq q_{\mathsf{ref}}^\star$. Moreover, $\pi$ is said to be $\epsilon$-reference optimal if $q^\pi \geq q_{\mathsf{ref}}^\star - \epsilon$.

Generally, $q^\star_{\text{ref}} < q^\star$. For instance, consider an MDP with one state $\perp$ (a bandit), $\mathsf{A} = [0, 1]$, and $\pi^{\text{ref}}_\perp = \mathcal{U}(\mathsf{A})$. If $r(\perp, \cdot) = \delta_{1/2}$, then $\sup_\mathsf{A} q^\star(\perp, \cdot) = 1$, while $\operatorname{ess\,sup}_{\pi^{\text{ref}}_\perp} q^\star(\perp, \cdot) = 0$. However, in many interesting cases, reference-optimal policies *are* optimal in the classic sense. When $\mathsf{A}$ is discrete and $\pi^{\text{ref}}_x$ is supported on all of $\mathsf{A}$—a ubiquitous assumption in ERL—then indeed $q^\star = q^\star_{\text{ref}}$. Likewise, when $\mathsf{A}$ is continuous and $(P, r)$ satisfy certain regularity conditions, then $q^\star$ is continuous [20]. In these case, a reference-optimal policy is optimal.

When $q^\star_{\text{ref}} \neq q^\star$, even state-of-the-art continuous-control methods, entropy-regularized or otherwise, can at best hope to achieve $q^\star_{\text{ref}}$, and not $q^\star$. This is because, when $q^\star_{\text{ref}} \neq q^\star$, optimal actions form a measure 0 set. And so, even rich policy classes, such as neural-network-parameterized Gaussian policies [19] or diffusion policies [9] will not sample these actions, with probability 1. Thus, moving forward, we establish $q^\star_{\text{ref}}$ as a "skyline" for optimal performance. In other words, we strive to achieve convergence to reference-optimal policies.

Under the next assumption, we can derive convergent policy optimization schemes as $\tau$ tends to zero.

**Assumption 3.4.** A constant $p_{\text{ref}} > 0$ exists for which

$$\inf_{\tau > 0} \inf_{x \in \mathsf{X}} \pi^{\text{ref}}_x \left( \left\{ a \in \mathsf{A} \,:\, q^\star_\tau(x, a) = \operatorname{ess\,sup}_{\pi^{\text{ref}}_x} q^\star_\tau(x, \cdot) \right\} \right) \geq p_{\text{ref}}.$$

**Remark 3.5.** If $\mathsf{A}$ is discrete and $\pi^{\text{ref}}_x$ is uniformly lower bounded, Assumption 3.4 holds. This is a standard assumption. When $\mathsf{A}$ is continuous, this assumption is more difficult to guarantee. Intuitively, it asks that there is enough mass surrounding the optima of the entropy-regularized optimal value functions $q^\star_\tau$ for $\text{KL}((\mathcal{G}_\tau q^\star_\tau)_x \,\|\, \pi^{\text{ref}}_x)$ to remain bounded in the limit.

A result key to the remainder of our work is the following bound on the total variation distance between pairs of BG policies in terms of their temperature and the distance between their potentials.

**Theorem 3.6.** *Let $q, q' \in M(\mathsf{X} \times \mathsf{A})$. For any $\tau > 0$ and any $x \in \mathsf{X}$,*

$$\|(\mathcal{G}_\tau q)_x - (\mathcal{G}_\tau q')_x\|_{\text{TV}}$$
$$\leq \min \left\{ \sqrt{\tau^{-1} \|q(x, \cdot) - q'(x, \cdot)\|_{L^\infty(\pi^{\text{ref}}_x)}}, \frac{1}{2} \sinh \left( 4\tau^{-1} \|q(x, \cdot) - q'(x, \cdot)\|_{L^\infty(\pi^{\text{ref}}_x)} \right) \right\}.$$

*In particular,*

$$\|(\mathcal{G}_\tau q)_x - (\mathcal{G}_\tau q')_x\|_{\text{TV}} \leq \frac{2e - 3}{4} \tau^{-1} \|q(x, \cdot) - q'(x, \cdot)\|_{L^\infty(\pi^{\text{ref}}_x)},$$

*if $\|q(x, \cdot) - q'(x, \cdot)\|_{L^\infty(\pi^{\text{ref}}_x)} < \tau/2$.*                                          [Proof]

While $q^\star_\tau$ and $\mathcal{V}_\tau q^\star_\tau$ converge in the zero-temperature limit, whether or not $\tau$-regularized optimal policies $\pi^{\tau,\star}$ converge is still unclear. Indeed, under Assumption 3.4, $\|q^\star_{\text{ref}} - q^\star_\tau\|_\infty \lesssim \tau$ (see Lemma B.10). However, the log-probabilities of an action $a$ under $\pi^{\tau,\star}$ are amplified by $\tau^{-1}$. Hence, the total variation difference between the BG policy at temperature $\tau$ and potential $q^\star_{\text{ref}}$ and $\pi^{\tau,\star}$ may not vanish as $\tau$ vanishes. Based on this insight, we introduce the *temperature decoupling gambit*.

**Definition 3.7.** Given $\tau > 0$, the *temperature decoupling gambit* specifies an alternate temperature $\sigma = \sigma(\tau)$ and constructs $\pi^{\tau,\sigma} := \mathcal{G}_\tau q^\star_\sigma$. In particular, it requires that $\sigma/\tau \to 0$ as $\tau \to 0$.

At any $\tau > 0$, decoupled-temperature policies $\pi^{\tau,\sigma}$ are necessarily *not* optimal for the $\tau$-regularized problem. Nevertheless, unlike $\pi^{\tau,\star}$, the policies $\pi^{\tau,\sigma}$ produced by the temperature decoupling gambit realize long-term advantages: they have convergence guarantees in the vanishing temperature limit, and they recover an interpretable reference-optimal policy.

**Definition 3.8.** Let $q^\star$ denote the optimal action-value function in a given MDP, and let $\pi^{\text{ref}} \in \mathsf{K}(\mathsf{X}, \mathscr{P}(\mathsf{A}))$. The *optimality-filtered* reference policy $\pi^{\text{ref},\star}$ is defined by

$$\pi^{\text{ref},\star}_x \propto \pi^{\text{ref}}_x \odot \chi_{\mathsf{N}^\star_{\text{ref}}(x)} \quad \text{where} \quad \mathsf{N}^\star_{\text{ref}}(x) := \{a \in \mathsf{A} \,:\, q^\star(x, a) = \operatorname{ess\,sup}_{\pi^{\text{ref}}_x} q^\star(x, \cdot)\}.$$

Here $\chi_\mathsf{Y}$ is the characteristic or indicator function for the measurable set $\mathsf{Y}$.

Heuristically, the optimality-filtered reference $\pi^{\text{ref},\star}_x$ is the *restriction* of $\pi^{\text{ref}}_x$ onto the set of expected-value-optimal actions in the state $x$.[3] When $\pi^{\text{ref}}$ is the uniform random policy, that is, $\pi^{\text{ref}}_x = \mathcal{U}(\mathsf{A})$

---

[3] This is exact when $q^\star = q^\star_{\text{ref}}$.

for all $x \in \mathsf{X}$, we see that $\pi_x^{\mathsf{ref},\star} = \mathcal{U}(\mathsf{N}_{\mathsf{ref}}^{\star}(x))$—the *uniform policy on optimal actions*. In a sense, $\pi^{\mathsf{ref},\star}$ is the *most diverse* (reference-)optimal policy; it does not discriminate between optimal actions.

In general, even when $\pi^{\tau,\star}$ does converge as $\tau$ converges to zero, its limit is different from $\pi^{\mathsf{ref},\star}$. We demonstrate this explicitly in Section 3.1. On the other hand, our next result proves that the temperature decoupling gambit enables convergence to $\pi^{\mathsf{ref},\star}$.[4]

**Theorem 3.9.** *Under Assumption 3.4, if $\sigma = \sigma(\tau)$ is such that $\lim_{\tau \to 0} \sigma/\tau = 0$, then $\pi_x^{\tau,\sigma} \to \pi_x^{\mathsf{ref},\star}$ as $\tau \to 0$, for all $x \in \mathsf{X}$, in TV if $\mathsf{A}$ is discrete and weakly if $\mathsf{A}$ is continuous.*          [Proof]

At the heart of the proof of Theorem 3.9 is the following inequality (a direct consequence of Theorem 3.6 and Lemma B.10), which relates the BG policies at temperature $\tau$ and potentials $q_\sigma^\star$ and $q_{\mathsf{ref}}^\star$:

$$\lim_{\tau \to 0} \sup_x \|(\mathcal{G}_\tau q_\sigma^\star)_x - (\mathcal{G}_\tau q_{\mathsf{ref}}^\star)_x\|_{\mathrm{TV}} \lesssim - \lim_{\tau \to 0} \frac{\sigma}{\tau} \log p_{\mathsf{ref}}.$$

This inequality reduces questions of convergence of $\mathcal{G}_\tau q_\sigma^\star$ to those of $\mathcal{G}_\tau q_{\mathsf{ref}}^\star$ (the vanishing temperature limit of a BG policy with a fixed potential is well-studied). Note that the smaller the fraction $\sigma/\tau$ is, the closer these two policies are. For instance, taking $\sigma(\tau) = \tau^3$ ensures that $\mathcal{G}_\tau q_\sigma^\star$ is more like $\mathcal{G}_\tau q_{\mathsf{ref}}^\star$ than taking $\sigma(\tau) = \tau^2$. In particular, it is from this inequality that the temperature decoupling gambit's requirement that $\sigma/\tau \to 0$ as $\tau \to 0$ arises.

Beyond enabling policy convergence in the vanishing temperature limit, the temperature decoupling gambit also ensures return distribution function convergence.

**Theorem 3.10.** *Suppose $\mathsf{A}$ is discrete and Assumption 3.4 holds. If $\sigma = \sigma(\tau)$ is such that $\sigma/\tau \to 0$ as $\tau \to 0$, then, for any $p, p' \in [1, \infty)$ and $\omega \in \mathscr{P}(\mathsf{X} \times \mathsf{A})$, as $\tau \to 0$, the return distribution functions $\zeta^{\tau,\sigma}$ of the temperature-decoupled policies $\pi^{\tau,\sigma}$ satisfy $d_{p;p',\omega}(\zeta^{\tau,\sigma}, \zeta^{\pi^{\mathsf{ref},\star}}) \to 0$.*          [Proof]

While Theorem 3.10 does not yet provide an algorithm for approximating $\zeta^\star$, this result serves as inspiration for such developments in Section 4.

## 3.1 Numerical Demonstration

In this section, we demonstrate that the policies learned via the temperature decoupling gambit differ from those learned in ERL, even in the presence of stochastic updates.

Figure 3.1 shows a given tristate MDP with two actions (blue: $a_1$; green: $a_2$), as well as learned policies $\hat{\pi}^{\tau,\star}$ and $\hat{\pi}^{\tau,\sigma}$ estimated with soft Q-learning [18]. Here $\pi_x^{\mathsf{ref}} = \mathcal{U}(\mathsf{A})$ for all $x \in \mathsf{X}$ and $\gamma = 0.9$. As this MDP is tabular, Theorem 2.5 implies that the policies $\pi^{\tau,\star}$ converge as $\tau \to 0$. Thus, the temperature decoupling gambit is not necessary to guarantee convergence. Yet we see different limiting behavior. As predicted by Theorem 3.9, the estimates $\hat{\pi}^{\tau,\sigma}$ converge to $\pi^{\mathsf{ref},\star}$, as $\tau \to 0$. With uniform $\pi^{\mathsf{ref}}$, this is the policy that samples all optimal actions, given a state, with equal probability. As $\tau \to 0$, the estimates $\hat{\pi}^{\tau,\star}$ do converge to a different optimal policy. This difference is in $x_0$, where $\hat{\pi}_{x_0}^{\tau,\star}$ collapse to $\delta_{a_1}$. We take $\sigma = \tau^2$, in line with Definition 3.7. The two optimal policies found emphasize different notions of diversity. The limit of $\pi^{\tau,\star}$ filters out optimal actions in order to play actions more uniformly on average with respect to state occupancy in the long term, while the limit of $\pi^{\tau,\sigma}$ looks to maximize state-wise action diversity.

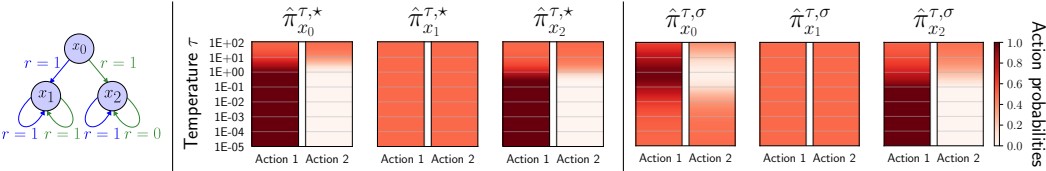

Figure 3.1: Differences between $\hat{\pi}^{\tau,\star}$ and $\hat{\pi}^{\tau,\sigma}$, approximated with soft Q-learning. **Left**: Graphical model of the MDP; arrow colors encode actions. **Center**: Depiction of the estimated policies $\hat{\pi}^{\tau,\star}$ at each state, as $\tau \to 0$. **Right**: Depiction of the estimated policies $\hat{\pi}^{\tau,\sigma}$ at each state, as $\tau \to 0$. **Summary**: Learned policies differ in $x_0$, but are otherwise the same.

---

[4] We discuss the benefits of this optimal policy in Appendix D.

# 4 Convergent Approximation of Optimal Return Distributions

In this section, we formalize a new branch of DRL and introduce distributional ERL (DERL). [5] Our main results in this section, Theorems 4.5, 4.6, and 4.7, establish convergent iterative schemes for approximate (reference-)optimal return distribution estimation. In Section 4.1, we introduce novel soft distributional Bellman operators, for evaluation and for control, and establish the convergence of their iterates. The behavior of the resulting return distribution approximations in the vanishing temperature limit is treated in Section 4.2. To conclude, a simulation is presented in Section 4.3 to illustrate the resulting optimal return distribution approximations.

## 4.1 Entropy-Regularized Distributional Reinforcement Learning

We begin by defining a *soft distributional Bellman operator*, as an analogue to the distributional Bellman operator [5, 35]. It, under certain conditions, computes

$$\bar{\zeta}^{\tau,\pi}_{x,a} := \text{law}\left( r(X^\pi_0, A^\pi_0) + \sum_{t \geq 1} \gamma^t \left( r(X^\pi_t) - \tau \text{KL}(\pi_{X^\pi_t} \,\|\, \pi^{\text{ref}}_{X^\pi_t}) \right) \,\Big|\, X^\pi_0 = x,\, A^\pi_0 = a \right).$$

Notationally, for any $\pi \in \mathsf{K}(\mathsf{X}, \mathscr{P}(\mathsf{A}))$, we define $\mathtt{kl}[\pi] : \mathsf{X} \to \mathbb{R}$ via $\mathtt{kl}[\pi](x) = \text{KL}(\pi_x \,\|\, \pi^{\text{ref}}_x)$.

**Definition 4.1.** For any $\tau > 0$, $\gamma < 1$, and $\pi \in \mathsf{K}(\mathsf{X}, \mathscr{P}(\mathsf{A}))$, the *soft distributional Bellman operator* $\mathcal{T}^\pi_\tau$ is given by

$$(\mathcal{T}^\pi_\tau \bar{\zeta})_{x,a} := \left( \mathtt{b}_{r(x,a),\gamma} \circ \mathtt{proj}^{\mathbb{R}} - \gamma\tau\mathtt{kl}[\pi] \circ \mathtt{proj}^{\mathsf{X}} \right)_\# \left( \bar{\zeta}_{-,-} \otimes \check{P}^\pi_{x,a} \right).$$

**Theorem 4.2.** *If $r \in M_b(\mathsf{X} \times \mathsf{A})$, $\gamma < 1$, and $\pi \in \mathsf{K}(\mathsf{X}, \mathscr{P}(\mathsf{A}))$ is such that*

$$\sup_{x,a} \|\tau\mathtt{kl}[\pi]\|_{L^p(P_{x,a})} < \infty, \tag{4.1}$$

*the soft distributional Bellman operator $\mathcal{T}^\pi_\tau$ is a $\gamma$-contraction in $\overline{d}_p$ for every $\tau \geq 0$. Thus, it has a unique solution to the fixed point equation $\bar{\zeta} = \mathcal{T}^\pi_\tau \bar{\zeta}$, which we denote by $\bar{\zeta}^{\pi,\tau}$.* [Proof]

Next, we move to *policy improvement*. In ERL, improving the action-value function $q$ involves policy evaluation with the policy $\mathcal{G}_\tau q$. We leverage this insight to enable control.

**Definition 4.3.** For any $\tau > 0$, the *soft distributional optimality operator* $\mathcal{T}^\star_\tau$ is given by

$$(\mathcal{T}^\star_\tau \bar{\zeta})_{x,a} := (\mathcal{T}^{\mathcal{G}_\tau \mathcal{Q}\bar{\zeta}}_\tau \bar{\zeta})_{x,a} \equiv \left( \mathtt{b}_{r(x,a),\gamma} \circ \mathtt{proj}^{\mathbb{R}} - \gamma\tau\mathtt{kl}[\mathcal{G}_\tau \mathcal{Q}\bar{\zeta}] \circ \mathtt{proj}^{\mathsf{X}} \right)_\# (\bar{\zeta}_{-,-} \otimes \check{P}^{\mathcal{G}_\tau \mathcal{Q}\bar{\zeta}}_{x,a})$$

where $\mathcal{Q} : \mathsf{K}(\mathsf{X} \times \mathsf{A}, \mathscr{P}(\mathbb{R})) \to M(\mathsf{X} \times \mathsf{A})$ is such that $(\mathcal{Q}\zeta)(x,a) := \mathbb{E}_{Z \sim \zeta_{x,a}}[Z]$.

We proceed by establishing a simple, but useful algebraic property.

**Lemma 4.4.** *For any $\tau > 0$, $\mathcal{Q}\mathcal{T}^\star_\tau = \mathcal{B}^\star_\tau \mathcal{Q}$.* [Proof]

Now we prove that iterates of $\mathcal{T}^\star_\tau$ converge, unlike iterates of $\mathcal{T}^\star$ [5].

**Theorem 4.5.** *For any $\bar{\zeta} \in \overline{\mathsf{K}}^p(\mathsf{X} \times \mathsf{A}, \mathscr{P}(\mathbb{R}))$ and temperature $\tau > 0$ define the iterates $(\bar{\zeta}^n)_{n \in \mathbb{N}}$ given by $\bar{\zeta}^{n+1} = \mathcal{T}^\star_\tau \bar{\zeta}^n$ for $\bar{\zeta}^0 = \mathcal{T}^\star_\tau \bar{\zeta}$. Then, for $\bar{\zeta}^{\tau,\star} := \bar{\zeta}^{\tau,\pi^{\tau,\star}}$,*

$$\overline{d}_p(\bar{\zeta}^n, \bar{\zeta}^{\tau,\star}) \leq C_{p,\tau,\gamma} n\gamma^{n/p} \overline{d}_p(\bar{\zeta}^0, \bar{\zeta}^{\tau,\star}) \quad \text{and} \quad \overline{d}_1(\bar{\zeta}^n, \bar{\zeta}^{\tau,\star}) \leq \frac{1}{(1-\gamma)\sqrt{\tau}} Cn\gamma^n \overline{d}_1(\bar{\zeta}^0, \bar{\zeta}^{\tau,\star}),$$

*where $C, C_{p,\tau,\gamma} < \infty$ are constants depending on $\|r\|_{\sup}$, $(p, \tau, \gamma, \|r\|_{\sup})$ respectively.* [Proof]

Theorem 4.5 leads to stability in entropy-regularized optimal return distribution estimation. In Figure 4.1, we demonstrate the stability of $\mathcal{T}^\star_\tau$ and the instability of $\mathcal{T}^\star$. The iterates defined in Theorem 4.5 converge to *soft return distributions*, which are influenced by stepwise regularization penalties and correspond to policies that are optimal in ERL. To estimate *optimal* return distributions, we must consider vanishing temperature limits.

---

[5] Independently and concurrently, similar results were established by [26] in the fixed-temperature regime, but only with discrete action spaces and $\pi^{\text{ref}}$ being the uniform policy.

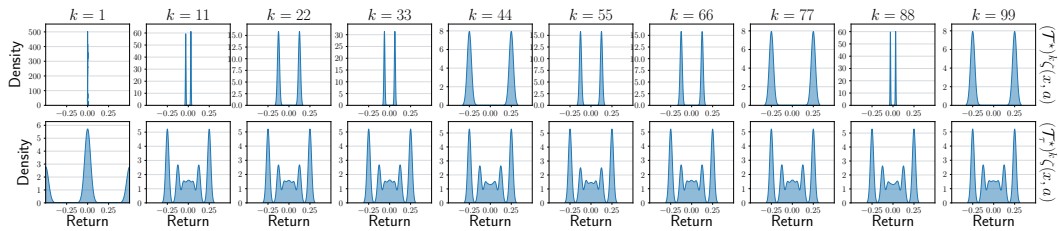

Figure 4.1: Evolution of the *soft* optimality iterates $(\mathcal{T}_\tau^\star)^k \zeta(x, a)$ (bottom row) and the iterates of the distributional optimality operator $(\mathcal{T}^\star)^k \zeta(x, a)$ (top row). Video of entire iterate sequence is available at https://harwiltz.github.io/assets/stable-return-distributions/.

## 4.2 Convergent Optimal Return Distribution Estimation in the Vanishing Temperature Limit

In this section, we instantiate the first methods for computing iterates that approximate reference-optimal return distribution functions in a stable manner.

**Theorem 4.6.** *Suppose Assumption 3.4 holds. Let $p, p' \in [1, \infty)$ and $\omega \in \mathscr{P}(\mathsf{X} \times \mathsf{A})$. For any $\epsilon, \delta > 0$, there exists a $\tau > 0$ for which $d_{p;p',\omega}(\bar{\zeta}^{\tau, \pi^{\tau,\star}}, \zeta^{\pi^{\tau,\star}}) \leq \delta/2$ and $q^{\pi^{\tau,\star}}$ is $\epsilon/2$-reference-optimal. In turn, an $n_{\epsilon,\delta} = n_{\epsilon,\delta}(\tau) \in \mathbb{N}$ exists for which*

$$d_{p;p',\omega}(\bar{\zeta}^n, \zeta^{\pi^{\tau,\star}}) \leq \delta \quad and \quad \mathcal{G}_\tau \mathcal{Q} \bar{\zeta}^n \text{ is } \epsilon\text{-reference-optimal} \quad \forall n \geq n_{\epsilon,\delta}$$

*where $\bar{\zeta}^{n+1} = \mathcal{T}_\tau^\star \bar{\zeta}^n$ and $\bar{\zeta}^0 = \mathcal{T}_\tau^\star \bar{\zeta}$ for any $\bar{\zeta} \in \overline{\mathsf{K}}^p(\mathsf{X} \times \mathsf{A}, \mathscr{P}(\mathbb{R}))$.* [Proof]

Theorem 4.6 is the first example of a convergent iterative scheme for approximating the return distribution of a (reference-)optimal policy. While it ensures convergence to a $\epsilon$-reference-optimal return distribution, it is still not possible a priori to characterize which return distribution will be learned. As $\epsilon \to 0$, there may be no stable trend in the return distribution that will be estimated because $\pi^{\tau,\star}$ may not converge. To achieve (characterizable) convergence to a reference-optimal return distribution, we turn back to the temperature decoupling gambit.

**Theorem 4.7.** *Suppose Assumption 3.4 holds and $\mathsf{A}$ is discrete. Let $p, p' \in [1, \infty)$ and $\omega \in \mathscr{P}(\mathsf{X} \times \mathsf{A})$. For any $\epsilon, \delta > 0$ and $\bar{\zeta}^0 \in \overline{\mathsf{K}}^p(\mathsf{X} \times \mathsf{A}, \mathscr{P}(\mathbb{R}))$, there exists $\tau > 0$, a decoupled $\sigma_\tau > 0$ and $n_{\mathsf{opt}}, n_{\mathsf{eval}} \in \mathbb{N}$ such that*

$$d_{p;p',\omega}(\hat{\zeta}^{n_{\mathsf{eval}}}, \zeta^{\pi^{\mathsf{ref},\star}}) \leq \delta \quad and \quad \mathcal{G}_\tau \mathcal{Q} \hat{\zeta}^{n_{\mathsf{eval}}} \text{ is } \epsilon\text{-reference-optimal}$$

*where $\bar{\zeta}^{n+1} = \mathcal{T}_\sigma^\star \bar{\zeta}^n$, $\hat{\pi}^{\tau,\sigma} = \mathcal{G}_\tau \bar{\zeta}^{n_{\mathsf{opt}}}$, and $\hat{\zeta}^{n+1} = \mathcal{T}_\tau^{\hat{\pi}^{\tau,\sigma}} \hat{\zeta}^n$, for $\hat{\zeta}^0 = \bar{\zeta}^{n_{\mathsf{opt}}}$.* [Proof]

Theorem 4.7 outlines an algorithm for estimating $\zeta^{\pi^{\mathsf{ref},\star}}$. First, approximate $\bar{\zeta}^{\sigma,\star}$ via $n_{\mathsf{opt}}$ applications of $\mathcal{T}_\sigma^\star$ (control). Second, extract the mean: $\hat{q}_\sigma^\star \approx q_\sigma^\star$. Finally, apply $\mathcal{T}_\tau^\pi$ $n_{\mathsf{eval}}$ times, with $\pi = \mathcal{G}_\tau \hat{q}_\sigma^\star$ (evaluation). If $\tau \ll 1$, $\sigma = \tau^2$, for example, and $n_{\mathsf{opt}}, n_{\mathsf{eval}} \gg 1$, then the resulting return distribution is as desired. This ensures convergence, (reference-)optimality, and interpretability of the final iterate.

### 4.3 Numerical Demonstration

Here we validate that $\bar{\zeta}^{\tau,\sigma}$ approximates $\zeta^{\pi^{\mathsf{ref},\star}}$. We consider the MDP given in Figure 4.2. Arrow colors correspond to different actions. Dashed lines represent transitions that occur with probability $1/2$. In this MDP, different optimal policies have distinct return distributions. From $x_1$, the blue action yields return of $2\gamma(1 - \gamma)^{-1}$, while the green action achieves return $4\gamma(1 - \gamma)^{-1}\mathsf{Bernoulli}(1/2)$. In Figures 4.3 and 4.4, we compute estimates $\hat{\zeta}^{\tau,\star} \approx \bar{\zeta}^{\tau,\star}$ and $\hat{\zeta}^{\tau,\sigma} \approx \bar{\zeta}^{\tau,\sigma}$ by (soft) distributional dynamic programming using 64-bit precision and

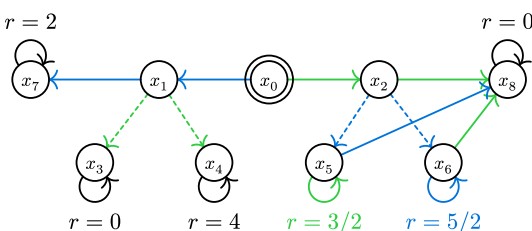

Figure 4.2: An illustrative MDP.

32-bit precision respectively. 32-bit precision is the default in many scientific computing libraries, such as Jax [8]. Here $\gamma = 1/2$, $\pi_x^{\mathsf{ref}} = \mathcal{U}(\mathsf{A})$ for all $x \in \mathsf{X}$, and $\sigma = \tau^2$. We consider $\tau \in \{10^{-(2m+1)} : m = 0, 1, 2, 3, 4\}$. Our simulation is a practical implementation of Theorem 4.7. First, we approximate $n_{\mathsf{opt}} = 1000$ iterative applications of our soft Bellman optimality operator at $\tau$

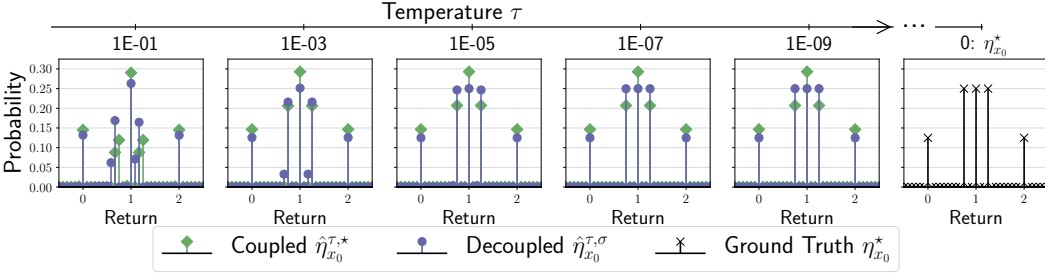

Figure 4.3: Estimates of return distributions via soft distributional dynamic programming—$\hat{\eta}^{\tau,\sigma}$ using the temperature-decoupling gambit and $\hat{\eta}^{\tau,\star}$ without—as $\tau \to 0$. As the temperature vanishes, $\eta^{\tau,\sigma}$ recovers the return distribution of $\pi^{\mathsf{ref},\star}$, shown on the right.

(control). Then, we extract $\hat{q}_\tau^\star$, an approximation of $q_\tau^\star$, and construct two policies: the BG policy at $\tau$ and the BG policy at $\tau^{1/2}$, both with potential $\hat{q}_\tau^\star$. Next we approximate $n_{\mathsf{eval}} = 1000$ iterative applications of our soft Bellman operator (policy evaluation) at temperature $\tau$ with the first policy and at temperature $\tau^{1/2}$ with the second policy. These yield approximations of $\bar{\zeta}^{\tau,\star}$ and $\bar{\zeta}^{\tau,\sigma}$, respectively. Figures 4.3 and 4.4 depict the policy-averaged return distributions $\hat{\eta}_{x_0}^{\tau,\star}$ and $\hat{\eta}_{x_0}^{\tau,\sigma}$ compared to the

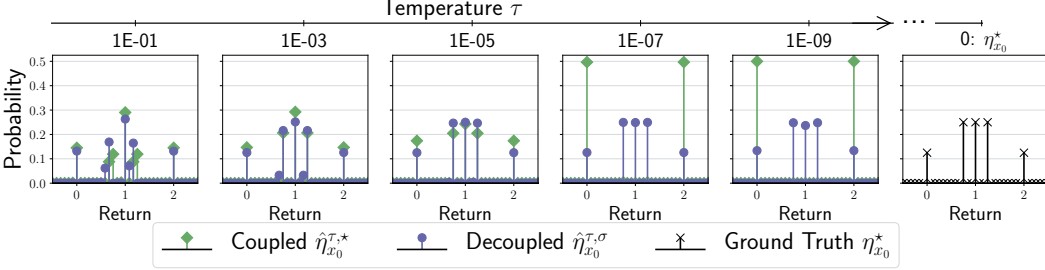

Figure 4.4: Return distribution estimation with vanishing temperature using soft distributional dynamic programming, with 32-bit floating point precision.

baseline $\eta_{x_0}^\star := \mathtt{proj}_\#^{\mathbb{R}}(\zeta_{x_0,\_}^\star \otimes \pi_{x_0}^{\mathsf{ref},\star})$. The iterates are approximated via categorical representations [5, 34] supported on 121 uniformly-spaced atoms on $[-2, 8]$, and MMD projections [43] with the energy distance kernel $\mathcal{E}_{3/2}$. In both figures, we see that the sequence of temperature-decoupled return distribution estimates approximate the return distribution associated to $\pi^{\mathsf{ref},\star}$ (right). Return distributions estimates of $\bar{\zeta}^{\tau,\star}$ also converge to those of optimal policies, as predicted by Theorem 4.6, but we find reach *different* return distributions in each case. While the temperature-decoupling gambit is not impervious to precision issues, it stabilizes BG policy estimation.

## 5   Related Work

Entropy regularization in RL was introduced by [48] for *inverse* RL, where it is necessary to disambiguate optimal policies and identify the most likely reward function to explain demonstrated behavior. ERL with $\pi^{\mathsf{ref}}$ as the uniform policy—termed *maximum entropy* or *MaxEnt* RL, has been highly influential in deep reinforcement learning. Heuristically, MaxEnt RL encourages policies to be more uniform, thereby enhancing exploration, sample-efficiency, behavioral diversity [29, 18, 17], as well as robustness [16, 2, 11, 12]. Heuristic approaches to adaptive temperature schemes in deep

MaxEnt RL have been effective in practice [19, 47]. Policy optimization in MaxEnt RL has been shown to be equivalent to a form of inference, conditional on a notion of behavioral optimality, in a certain graphical model [24, 14], and further characterizations of MaxEnt RL have lead to principled algorithms for efficient exploration [30, 39]. Alternative forms of regularized RL objectives and optimizers have been proposed and analyzed [25, 31, 36, 4, 37, 15].

Policy optimization algorithms for entropy-regularization in general are presented and analyzed by [28]—these methods apply to tabular MDPs and fixed nonzero temperature. [27] provide improved convergence rates for entropy-regularized policy optimization. They also derive convergence results in the vanishing temperature limit, but only in the bandit setting. Exceptionally, [23], based on the work of [1], studies global convergence of policy gradient methods in continuous entropy-regularized MDPs, for fixed and vanishing temperature, with neural network policies via mean-field analysis. However, their analysis requires an extra regularization term to a distribution over neurons, precluding convergence to an optimum of RL. To the best of our knowledge, our work is the first to introduce a convergent policy optimization scheme for general MDPs in the vanishing temperature limit.

Entropy regularization in DRL is largely unexplored. [22] experimented with an adaptation of Rainbow [21] to MaxEntRL, but without analysis or formalism. The concurrent work of [26] also introduced soft distributional Bellman operators, but did not study vanishing temperature limits, and did not establish convergence rates for iterates of $\mathcal{T}_\tau^\star$ even for fixed $\tau$. Moreover, the work of [26] established convergence only in the case of discrete $\mathsf{A}$, and only for a uniform reference policy. Works have investigated the challenges of estimating optimal return distributions [5, 42], and more generally, the influence of particular tractable distribution representations on learning dynamics and fixed point accuracy [45, 46, 43, 3]. In [6], the authors show that distributional analogues of $\mathcal{B}^\star$ produce iterates that converge when there is a unique (deterministic) optimal policy. The interplay between policy optimization stability and return distributions was studied in [33]. Their empirical study found that distributions of returns following stochastic policy gradient updates tend to have long left tails, and called for methods to guide policies into smoother regions ("quiet" neighborhoods) of the *return landscape*, the manifold of policy returns across parameters. This study focused primarily on deterministic policy gradient methods.

## 6 Discussion

In this work, we have investigated policy and return distribution convergence as the temperature vanishes in ERL. Our findings motivate iterative schemes for achieving convergence results beyond expected returns. However, they come with several limitations. In particular, while we have established policy convergence via the temperature-decoupling gambit, this convergence qualitative. As a consequence, our ability to derive approximation algorithms for $\zeta^\pi$ with $\pi = \pi^{\mathsf{ref},\star}$ is limited; it is a priori unclear which temperatures are required for $\zeta^{\tau,\sigma}$ to be an $\epsilon$-approximation of $\zeta^\pi$ with $\pi = \pi^{\mathsf{ref},\star}$ in $d_{p;p,\omega}$ and, therefore, to deploy for iterative applications of $\mathcal{T}_\tau^\star$ or $\mathcal{T}_\tau^\pi$ with $\pi = \mathcal{G}_\tau q_\sigma^\star$. At the moment, however, our results ensure that by progressively annealing $\tau$, the scheme discussed in Theorem 4.7 will approach $\zeta^\pi$ with $\pi = \pi^{\mathsf{ref},\star}$. Nevertheless, quantifying Theorem 3.10 is an exciting direction for future work. Another exciting direction for future work is to try to incorporate the temperature-decoupling gambit into the many algorithms in ERL/RL.

## Acknowledgments and Disclosure of Funding

The authors wish to thank Wesley Chung, Mark Rowland, Jesse Farebrother, Arnav Kumar Jain, Siddarth Venkatraman, Athanasios Vasileiadis, Aditya Mahajan, and Doina Precup for helpful comments and discussions. HW was supported by the National Sciences and Engineering Research Council of Canada (NSERC) and the Fonds de Recherche du Québec. MGB was supported by the Canada CIFAR AI Chair program and NSERC. This work was supported in part by DARPA HR0011-23-9-0050.

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

# A  Entropy-Regularized RL in Continuous MDPs

Here we prove Theorem 2.3 as well as a collection of supporting and related results that generalize well-known results in tabular MDPs.

We start with a characterization the geometry of the space of occupancy measures. The following result extends the well-known counterpart in tabular MDPs [13, 38, 10] to continuous MDPs. While certain parts of this result are proved by [20], not all connections are made, which we state here for the first time.

**Theorem A.1.** *Let $\mathscr{O}(\nu_0) = \{\mu^\pi : \pi \in \mathsf{K}(\mathsf{X}, \mathscr{P}(\mathsf{A}))\}$ the space of all occupancy measures under the initial state distribution $\nu_0 \in \mathscr{P}(\mathsf{X})$. Then $\mathscr{O}(\nu_0)$ is equivalent to the space of all $\mu \in \mathscr{P}(\mathsf{X} \times \mathsf{A})$ that satisfy*

$$\mathtt{proj}^{\mathsf{X}}_{\#} \mu(\mathsf{E}) = (1 - \gamma)\nu_0(\mathsf{E}) + \gamma \int P_{x,a}(\mathsf{E})\, \mathrm{d}\mu(x, a) \quad \forall \mathsf{E} \subset \mathsf{X}\ \textit{Borel}. \tag{A.1}$$

*The space $\mathscr{O}(\nu_0)$ is convex, it is closed under setwise convergence.*

Before proceeding with the proof of Theorem A.1, we recall the *state occupancy measures* $\nu^\pi$, given by

$$\nu^\pi := (1 - \gamma) \sum_{t \geq 0} \gamma^t \nu_t^\pi,$$

where $(\nu_t^\pi)_{t \geq 1}$ is the sequence of laws generated by $\hat{P}^\pi$ starting at $\nu_0$.

**Proposition A.2.** *Let $\nu_t^\pi$ and $\mu_t^\pi$, for $t \geq 1$ denote the laws generated by $\hat{P}^\pi$ and $\check{P}^\pi$ starting at $\nu_0$ and $\mu_0^\pi = \pi_- \otimes \nu_0$. Then $\mu_t^\pi = \pi_- \otimes \nu_t^\pi$, for all $t \geq 1$. Hence, given $\pi$ and $\nu_0$, the state marginal of the associated occupancy measure $\mu^\pi$ is the associated state occupancy measure $\nu^\pi$.*

The proof of this proposition will use the following lemma.

**Lemma A.3.** *Under the hypotheses of Proposition A.2, for every $t \geq 1$, the conditional probabilities of $\mu_t^\pi$ with respect to its state marginal are $\pi_-$.*

*Proof.* It suffices to prove that the conditional probabilities of $\mu_1^\pi$ are $\pi_-$. Let $\nu_1$ denote the state marginal of $\mu_1^\pi$. By definition,

$$\int \psi(x')\, \mathrm{d}\nu_1(x') = \int \psi(x')\, \mathrm{d}\mu_1^\pi(x', a') = \int \left[ \int \psi(x')\, \mathrm{d}P_{x,a}(x') \right] \mathrm{d}\mu_0^\pi(x, a).$$

Thus, for any $\varphi \in M_b(\mathsf{X} \times \mathsf{A})$, with $\psi(x') := \int \varphi(x', a')\, \mathrm{d}\pi_{x'}(a')$, observe that

$$\begin{aligned}
\int \left[ \int \varphi(x', a')\, \mathrm{d}\pi_{x'}(a') \right] \mathrm{d}\nu_1(x') &= \int \psi(x')\, \mathrm{d}\nu_1(x') \\
&= \int \left[ \int \psi(x')\, \mathrm{d}P_{x,a}(x') \right] \mathrm{d}\mu_0^\pi(x, a) \\
&= \int \left[ \int \left[ \int \varphi(x', a')\, \mathrm{d}\pi_{x'}(a') \right] \mathrm{d}P_{x,a}(x') \right] \mathrm{d}\mu_0^\pi(x, a) \\
&= \int \left[ \int \varphi(x', a')\, \mathrm{d}\check{P}_{x,a}^\pi(x') \right] \mathrm{d}\mu_0^\pi(x, a) \\
&= \int \varphi(x, a)\, \mathrm{d}\mu_1^\pi(x, a).
\end{aligned}$$

So the conditional probabilities of $\mu_1^\pi$ with respect to $\nu_1$ are $\pi_x$, as desired. $\square$

*Proof of Proposition A.2.* By Lemma A.3, it suffices to show that the state marginal of $\mu_1^\pi$ is $\nu_1^\pi$. This holds:

$$\int \psi(x')\,\mathrm{d}\nu_1^\pi(x') = \int \left[\int \psi(x')\,\mathrm{d}\hat{P}_x^\pi(x')\right]\mathrm{d}\nu_0(x)$$

$$= \int \left[\int \left[\int \psi(x')\,\mathrm{d}P_{x,a}(x')\right]\mathrm{d}\pi_x(a)\right]\mathrm{d}\nu_0(x)$$

$$= \int \left[\int \psi(x')\,\mathrm{d}P_{x,a}(x')\right]\mathrm{d}(\pi_x \otimes \nu_0)(x,a)$$

$$= \int \left[\int \psi(x')\,\mathrm{d}P_{x,a}(x')\right]\mathrm{d}\mu_0^\pi(x,a).$$

By this computation and Lemma A.3 applied successively to each pair $(\mu_{t+1}^\pi, \mu_t^\pi)$ for every $t \geq 1$, we deduce that $\mu_t^\pi = \pi_- \otimes \nu_t^\pi$, for all $t \geq 1$. Finally, by the linearity of the integral, we conclude. Indeed,

$$\mu^\pi := (1-\gamma)\sum_{t\geq 0}\gamma^t\mu_t^\pi = (1-\gamma)\sum_{t\geq 0}\gamma^t(\pi_- \otimes \nu_t^\pi) = \pi_- \otimes (1-\gamma)\sum_{t\geq 0}\gamma^t\nu_t^\pi =: \pi_- \otimes \nu^\pi.$$

$\square$

*Proof of Theorem A.1.* We prove this theorem in three steps.

**Step 1:** $\mathscr{O}(\nu_0) = \mathscr{F}(\nu_0)$. First, recall that $\mathrm{proj}_{\#}^{\mathsf{X}}\mu^\pi = \nu^\pi$ for any policy $\pi$, by Proposition A.2. Thus, we have that for any $\pi$ and any Borel $\mathsf{E} \subset \mathsf{X}$,

$$\mathrm{proj}_{\#}^{\mathsf{X}}\mu^\pi(\mathsf{E}) = \nu^\pi(\mathsf{E}) = (1-\gamma)\nu_0(\mathsf{E}) + \gamma(1-\gamma)\sum_{t\geq 0}\gamma^t\nu_{t+1}^\pi(\mathsf{E})$$

$$= (1-\gamma)\nu_0(\mathsf{E}) + \gamma(1-\gamma)\sum_{t\geq 0}\gamma^t\int P_{x,a}(\mathsf{E})\,\mathrm{d}\mu_t^\pi(x,a)$$

$$= (1-\gamma)\nu_0(\mathsf{E}) + \gamma\int P_{x,a}(\mathsf{E})\,\mathrm{d}\mu^\pi(x,a).$$

This shows that $\mathscr{O}(\nu_0) \subset \mathscr{F}(\nu_0)$. It remains to show that $\mathscr{F}(\nu_0) \subset \mathscr{O}(\nu_0)$. Let $\mu \in \mathscr{F}(\nu_0)$, and let $\pi^\mu$ denote its conditional action probabilities with respect to its state marginal $\nu^\mu$—that is, $\mu = \pi_-^\mu \otimes \nu^\mu$. Moreover, let $\phi_0$ be any bounded measurable function. By the definition of $P$, we note that (A.1) can be written as

$$\int \phi_0(x_0)\,\mathrm{d}\nu^\mu(x_0) = (1-\gamma)\int \phi_0(x_0)\,\mathrm{d}\nu_0(x_0) + \gamma\int\left[\int \phi_0(x_1)\,\mathrm{d}\hat{P}_{x_0}^{\pi^\mu}(x_1)\right]\mathrm{d}\nu^\mu(x_0).$$

Defining $\phi_1(x) = \int_{\mathsf{X}}\phi_0(x')\,\mathrm{d}\hat{P}_x^{\pi^\mu}(x')$, the rightmost term $\int_{\mathsf{X}}\phi_1(x_0)\,\mathrm{d}\nu^\mu(x_0)$ can be again expanded via (A.1),

$$\int \phi_0(x_0)\,\mathrm{d}\nu^\mu(x_0) = (1-\gamma)\int(\phi_0(x_0) + \gamma\phi_1(x_0))\,\mathrm{d}\nu_0(x_0)$$

$$+ \gamma^2\int\left[\int \phi_1(x_0)\,\mathrm{d}\hat{P}_{x_0}^{\pi^\mu}(x_1)\right]\mathrm{d}\nu^\mu(x_0).$$

Continuing, we define $\phi_{n+1}(x) = \int_{\mathsf{X}}\phi_n(x')\,\mathrm{d}\hat{P}_x^{\pi^\mu}(x')$, which is bounded and measurable for each $n \in \mathbb{N}$, yielding

$$\int \phi_0(x_0)\,\mathrm{d}\nu^\mu(x_0) = \underbrace{(1-\gamma)\int\sum_{k=0}^n\gamma^k\phi_k(x_0)\,\mathrm{d}\nu_0(x_0)}_{\mathrm{I}_n}$$

$$+ \underbrace{\gamma^{n-1}\int\left[\int \phi_n(x_0)\,\mathrm{d}\hat{P}_{x_0}^{\pi^\mu}(x_1)\right]\mathrm{d}\nu^\mu(x_0)}_{\mathrm{II}_n}.$$

By the definition of $\phi_n$, we have that

$$\mathrm{I}_n = (1-\gamma) \int \phi_0(x_0) \, \mathrm{d}\nu_0(x_0) + (1-\gamma)\gamma \int \left[ \int \phi_0(x_1) \, \mathrm{d}\hat{P}^{\pi^\mu}_{x_0}(x_1) \right] \mathrm{d}\nu_0(x_0) + \dots$$

$$= (1-\gamma) \int \phi_0(x_0) \, \mathrm{d}\nu_0(x_0) + (1-\gamma)\gamma \int \phi_0(x_0) \, \mathrm{d}\nu^{\pi^\mu}_1(x_0) + \dots$$

$$= \int \phi_0(x_0)(1-\gamma) \sum_{k=0}^{n} \gamma^k \mathrm{d}\nu^{\pi^\mu}_k(x_0).$$

Moreover, by the boundedness of $\phi_n$, we deduce that $\mathrm{II}_n \to 0$. Substituting, we have

$$\int \phi_0(x_0) \, \mathrm{d}\nu^\mu(x_0) = \lim_{n\to\infty} \mathrm{I}_n + \lim_{n\to\infty} \mathrm{II}_n$$

$$= \int \phi_0(x_0) \lim_{n\to\infty} (1-\gamma) \sum_{k=0}^{n} \gamma^k \, \mathrm{d}\nu^{\pi^\mu}(x_0)$$

$$= \int \phi_0(x_0) \mathrm{d}\nu^{\pi^\mu}(x_0).$$

Since $\phi_0$ was an arbitrary bounded and measurable function, it follows that $\nu^\mu = \nu^{\pi^\mu}$. Thus, $\mu = \pi_- \otimes \nu^\mu = \mu^{\pi^\mu}$—the occupancy measure for the policy $\pi^\mu$. Consequently, any $\mu \in \mathscr{F}(\nu_0)$ is a member of $\mathscr{O}(\nu_0)$.

**Step 2: $\mathscr{O}(\nu_0)$ is convex.** The convexity of $\mathscr{O}(\nu_0)$ follows immediately from the structure of $\mathscr{F}(\nu_0)$. Consider any $\mu_0, \mu_1 \in \mathscr{O}(\nu_0)$ any $\alpha \in [0,1]$, and define $\mu_\alpha = \alpha\mu_0 + (1-\alpha)\mu_1$. For any Borel $\mathsf{E} \subset \mathsf{X}$, we have that

$$\mathrm{proj}^{\mathsf{X}}_{\#}\mu_\alpha(\mathsf{E}) = \alpha\mathrm{proj}^{\mathsf{X}}_{\#}\mu_0(\mathsf{E}) + (1-\alpha)\mathrm{proj}^{\mathsf{X}}_{\#}\mu_1(\mathsf{E})$$

Since $\mu_0, \mu_1 \in \mathscr{F}(\nu_0)$, they solve (A.1), so we expand the RHS,

$$\mathrm{proj}^{\mathsf{X}}_{\#}\mu_\alpha(\mathsf{E}) = \alpha(1-\gamma)\nu_0(\mathsf{E}) + \alpha\gamma \int P_{x,a}(\mathsf{E}) \, \mathrm{d}\mu_0(x,a)$$

$$+ (1-\alpha)(1-\gamma)\nu_0(\mathsf{E}) + (1-\alpha)\gamma \int P_{x,a}(\mathsf{E}) \, \mathrm{d}\mu_1(x,a)$$

$$= (1-\gamma)\nu_0(\mathsf{E}) + \gamma \int P_{x,a}(\mathsf{E})(\alpha \, \mathrm{d}\mu_0(x,a) + (1-\alpha) \, \mathrm{d}\mu_1(x,a))$$

$$= (1-\gamma)\nu_0(\mathsf{E}) + \gamma \int P_{x,a}(\mathsf{E}) \, \mathrm{d}\mu_\alpha(x,a).$$

So $\mu_\alpha \in \mathscr{F}(\nu_0) = \mathscr{O}(\nu_0)$, as desired.

**Step 3: $\mathscr{O}(\nu_0)$ is closed under setwise convergence.** Let $(\mu_k)_{k\in\mathbb{N}} \subset \mathscr{F}(\nu_0)$ be a sequence that converges setwise to $\mu$. Since $(x,a) \mapsto P_{x,a}(\mathsf{E})$ is bounded and measurable for any Borel $\mathsf{E} \subset \mathsf{X}$,

$$\int P_{x,a}(\mathsf{E}) \, \mathrm{d}\mu_k(x,a) \to \int P_{x,a}(\mathsf{E}) \, \mathrm{d}\mu(x,a). \tag{A.2}$$

Likewise,

$$\mathrm{proj}^{\mathsf{X}}_{\#}\mu_k(\mathsf{E}) = \mu_k(\mathsf{E} \times \mathsf{A}) \to \mu(\mathsf{E} \times \mathsf{A}) = \mathrm{proj}^{\mathsf{X}}_{\#}\mu(\mathsf{E}), \tag{A.3}$$

as $\mu_k \to \mu$ setwise. Consequently, we have that

$$\mathrm{proj}^{\mathsf{X}}\mu(\mathsf{E}) = \lim_{k\to\infty} \mathrm{proj}^{\mathsf{X}}\mu(\mathsf{E})$$

$$= \lim_{k\to\infty} \left[ (1-\gamma)\nu_0(\mathsf{E}) + \gamma \int P_{x,a}(\mathsf{E}) \, \mathrm{d}\mu_k(x,a) \right]$$

$$= (1-\gamma)\nu_0(\mathsf{E}) + \gamma \int P_{x,a}(\mathsf{E}) \, \mathrm{d}\mu(x,a),$$

where the first equality follows from (A.3), the second follows as $\mu_k \in \mathscr{F}(\nu_0)$, and the final equality follows from (A.2). Thus, we see that $\mu \in \mathscr{F}(\nu_0) = \mathscr{O}(\nu_0)$. $\qquad\square$

Now we prove Lemma 2.1.

**Lemma 2.1.** *The functional $\mathcal{R} : \mathscr{P}(\mathsf{X} \times \mathsf{A}) \to \mathbb{R}$ is strictly convex.*  

*Proof.* Observe that

$$\mathcal{R}(\mu) = \mathrm{KL}(\mu \,\|\, \bar{\pi}_- \otimes \nu^\mu).$$

We prove this in two steps. First, for every Borel $f : \mathsf{X} \times \mathsf{A} \to [0, \infty)$, we have that

$$\int f(x,a) \frac{\mathrm{d}\pi_x^\mu}{\mathrm{d}\pi_x^{\mathsf{ref}}}(a) \, \mathrm{d}(\pi_-^{\mathsf{ref}} \otimes \nu^\mu)(x,a) = \int \left[ \int f(x,a) \frac{\mathrm{d}\pi_x^\mu}{\mathrm{d}\pi_x^{\mathsf{ref}}} \, \mathrm{d}\pi_x^{\mathsf{ref}}(a) \right] \mathrm{d}\nu^\mu(x)$$

$$= \int \left[ \int f(x,a) \, \mathrm{d}\pi_x^\mu(a) \right] \mathrm{d}\nu^\mu(x)$$

$$= \int f(x,a) \, \mathrm{d}(\pi_-^\mu \otimes \nu^\mu)(x,a).$$

Hence, $\mu = \pi_-^\mu \otimes \nu^\mu \ll \pi_-^{\mathsf{ref}} \otimes \nu^\mu$ if $\pi_x^\mu \ll \pi_x^{\mathsf{ref}}$ for $\nu^\mu$-almost every $x$, and

$$\frac{\mathrm{d}\mu}{\mathrm{d}(\pi_-^{\mathsf{ref}} \otimes \nu^\mu)}(x,a) = \frac{\mathrm{d}\pi_x^\mu}{\mathrm{d}\pi_x^{\mathsf{ref}}}(a).$$

Second, $\mu = \pi_-^\mu \otimes \nu^\mu \ll \pi_-^{\mathsf{ref}} \otimes \nu^\mu$ implies that $\pi_x^\mu \ll \pi_x^{\mathsf{ref}}$ for $\nu^\mu$-almost every $x$. Indeed, suppose that a set $\mathsf{S} \subset \mathsf{X}$ exists such that $\nu^\mu(\mathsf{S}) > 0$ and for each $x \in \mathsf{S}$, we have that

$$\pi_x^\mu(\mathsf{B}_x) > 0 \quad \text{but} \quad \pi^{\mathsf{ref}}(\mathsf{B}_x) = 0.$$

Let

$$\mathsf{E} := \bigcup_{x \in \mathsf{S}} \{x\} \times \mathsf{B}_x.$$

Then,

$$(\pi_-^{\mathsf{ref}} \otimes \nu^\mu)(\mathsf{E}) = \int_\mathsf{S} \pi_x^{\mathsf{ref}}(\mathsf{B}_x) \, \mathrm{d}\nu^\mu(x) = 0 \quad \text{and} \quad (\pi_-^\mu \otimes \nu^\mu)(\mathsf{E}) = \int_\mathsf{S} \pi_x^\mu(\mathsf{B}_x) \, \mathrm{d}\nu^\mu(x) > 0.$$

This is a contradiction. And so,

$$\mathcal{R}(\mu) = \int \left[ \int \log\left( \frac{\mathrm{d}\pi_x^\mu}{\mathrm{d}\pi_x^{\mathsf{ref}}}(a) \right) \mathrm{d}\pi_x^\mu(a) \right] \mathrm{d}\nu^\mu(x)$$

$$= \int \left[ \int \log\left( \frac{\mathrm{d}\mu}{\mathrm{d}(\pi_-^{\mathsf{ref}} \otimes \nu^\mu)}(x,a) \right) \mathrm{d}\pi_x^\mu(a) \right] \mathrm{d}\nu^\mu(x)$$

$$= \int \log\left( \frac{\mathrm{d}\mu}{\mathrm{d}(\pi_-^{\mathsf{ref}} \otimes \nu^\mu)}(x,a) \right) \mathrm{d}\mu(x,a)$$

$$= \mathrm{KL}(\mu \,\|\, \bar{\pi}_- \otimes \nu^\mu),$$

as desired.

Now recall that

$$\mathrm{KL}(t\mu_1 + (1-t)\mu_0 \,\|\, t\mu_1' + (1-t)\mu_0') \leq t\mathrm{KL}(\mu_1 \,\|\, \mu_1') + (1-t)\mathrm{KL}(\mu_0 \,\|\, \mu_0').$$

Moreover, note that

$$\nu^{t\mu_1 + (1-t)\mu_0} = t\nu^{\mu_1} + (1-t)\nu^{\mu_0}.$$

In turn,

$$\mathcal{R}(t\mu_1 + (1-t)\mu_0) = \mathrm{KL}(t\mu_1 + (1-t)\mu_0 \,\|\, \pi_-^{\mathsf{ref}} \otimes \nu^{t\mu_1 + (1-t)\mu_0})$$

$$= \mathrm{KL}(t\mu_1 + (1-t)\mu_0 \,\|\, \pi_-^{\mathsf{ref}} \otimes (t\nu^{\mu_1} + (1-t)\nu^{\mu_0}))$$

$$= \mathrm{KL}(t\mu_1 + (1-t)\mu_0 \,\|\, t(\pi_-^{\mathsf{ref}} \otimes \nu^{\mu_1}) + (1-t)(\pi_-^{\mathsf{ref}} \otimes \nu^{\mu_0}))$$

$$\leq t\mathrm{KL}(\mu_1 \,\|\, \pi_-^{\mathsf{ref}} \otimes \nu^{\mu_1}) + (1-t)\mathrm{KL}(\mu_0 \,\|\, \pi_-^{\mathsf{ref}} \otimes \nu^{\mu_0})$$

$$= t\mathcal{R}(\mu_1) + (1-t)\mathcal{R}(\mu_0).$$

Thus, $\mathcal{R}$ is convex. In particular, $\mathcal{R}$ is strictly convex as KL is strictly convex in its first argument. $\quad\square$

With Theorem A.1 and Lemma 2.1 in hand, we use the direct method from the Calculus of Variations to prove the well-posedness of $\tau$-ERL, in the tabular setting.

**Remark A.4.** The space $M_b(\mathsf{X} \times \mathsf{A})$ endowed with the supnorm is a Banach space. Note that $M_b(\mathsf{X} \times \mathsf{A})^* \cong ba(\mathsf{X} \times \mathsf{A})$, where $ba(\mathsf{X} \times \mathsf{A})$ denotes the set of finitely additive set functions on $\mathscr{B}(\mathsf{X} \times \mathsf{A})$ equipped with the total variation norm. Note that the set of probability measures on $\mathsf{X} \times \mathsf{A}$ is a subset of the closed unit ball in $ba(\mathsf{X} \times \mathsf{A})$, which is weak* compact, by Banach–Alaoglu. The duality pairing for any $\mu \in \mathscr{P}(\mathsf{X} \times \mathsf{A})$ and for any $\varphi \in M_b(\mathsf{X} \times \mathsf{A})$ is given by integration: $\langle \mu, \varphi \rangle := \int \varphi \, \mathrm{d}\mu$. In other words, weak* convergence is setwise convergence when $\mathscr{P}(\mathsf{X} \times \mathsf{A})$ is considered as a subset of the dual of $ba(\mathsf{X} \times \mathsf{A})$.

**Theorem A.5.** *Suppose that $r \in M_b(\mathsf{X} \times \mathsf{A})$, $\mathsf{X} \times \mathsf{A}$ is finite, and let $\nu_0 \in \mathscr{P}(\mathsf{X})$. A $\mu_\tau^\star \in \mathscr{O}(\nu_0)$ that achieves the supremum in* (2.2) *exists. Moreover, no other occupancy measure does so.*

*Proof.* Let the supremum in (2.2) be denoted by $\vartheta_\tau^\star$ and $(\mu_k)_{k \in \mathbb{N}} \subset \mathscr{O}(\nu_0)$ be such that

$$\vartheta_\tau^\star - \frac{1}{k} < \mathcal{J}_\tau(\mu_k) \leq \vartheta_\tau^\star.$$

In other words, let $(\mu_k)_{k \in \mathbb{N}} \subset \mathscr{O}(\nu_0)$ be a maximizing sequence. By Remark A.4, owing to the fact that $M_b(\mathsf{X} \times \mathsf{A})$ is separable (since $\mathsf{X} \times \mathsf{A}$ is finite), let $(\mu_{k_\ell})_{\ell \in \mathbb{N}}$ be a weakly* convergent subsequence, with weak* limit $\mu_\infty$. In particular, $\mu_{k_\ell} \to \mu_\infty$ setwise. As $\mathscr{O}(\nu_0)$ is closed under setwise convergence, by Theorem A.1, we have that $\mu_\infty \in \mathscr{O}(\nu_0)$. Furthermore, $\pi_-^{\mathsf{ref}} \otimes \nu^{\mu_{k_\ell}} \to \pi_-^{\mathsf{ref}} \otimes \nu^{\mu_\infty}$ setwise as well. As setwise convergence implies weak convergence and as the $\mathrm{KL}(\mu \,\|\, \mu')$ is lower-semicontinuous in the pair $(\mu, \mu')$ in the weak topology, we find that

$$\vartheta^{\tau,\star} \leq \limsup_{\ell \to \infty} \int r \, \mathrm{d}\mu_{k_\ell} - \tau \liminf_{\ell \to \infty} \mathrm{KL}(\mu_{k_\ell} \,\|\, \pi_-^{\mathsf{ref}} \otimes \nu^{\mu_{k_\ell}})$$

$$\leq \limsup_{\ell \to \infty} \int r \, \mathrm{d}\mu_{k_\ell} - \tau \mathrm{KL}(\mu_\infty \,\|\, \pi_-^{\mathsf{ref}} \otimes \nu^{\mu_\infty})$$

$$= \int r \, \mathrm{d}\mu_\infty - \tau \mathrm{KL}(\mu_\infty \,\|\, \pi_-^{\mathsf{ref}} \otimes \nu^{\mu_\infty})$$

$$= \mathcal{J}_\tau(\mu_\infty).$$

The penultimate equality uses that $r$ is bounded. Thus, $\mathcal{J}_\tau(\mu_\infty) = \vartheta_\tau^\star$. The previous argument applies to any sub-sequential weak* limit of our maximizing sequence. But as $\mathcal{R}$ is strictly convex, by Lemma 2.1, and $\mathscr{O}(\nu_0)$ is convex, by Theorem A.1, only one such limit exists. $\qquad \square$

We now move to prove Theorem 2.3. To do so, we state and prove some helpful results. We begin with policy evaluation.

For any $\pi \in \mathsf{K}(\mathsf{X}, \mathscr{P}(\mathsf{A}))$, define $q_\tau^\pi : \mathsf{X} \times \mathsf{A} \to \mathbb{R} \cup \{-\infty\}$ by

$$q_\tau^\pi(x, a) := \mathbf{E}\left[ r(X_0^\pi, A_0^\pi) + \sum_{t \geq 1} \gamma^t \Big( r(X_t^\pi, A_t^\pi) - \tau \mathrm{KL}(\pi_{X_t^\pi} \,\|\, \pi_{X_t^\pi}^{\mathsf{ref}}) \Big) \,\Big|\, (X_0^\pi, A_0^\pi) = (x, a) \right].$$

By the tower property of condition expectation, we have that

$$q_\tau^\pi(x, a) = r(x, a) + \gamma \int q_\tau^\pi(x', a') - \tau \mathrm{KL}(\pi_{x'} \,\|\, \pi_{x'}^{\mathsf{ref}}) \, \mathrm{d}\check{P}_{x,a}^\pi(x', a').$$

It is convenient to be able to evaluate a policy $\pi$ (find $q_\tau^\pi$) in an iterative fashion. This can be done via the *soft Bellman operator* $\mathcal{B}_\tau^\pi : M(\mathsf{X} \times \mathsf{A}) \to M(\mathsf{X} \times \mathsf{A})$ defined by

$$(\mathcal{B}_\tau^\pi q)(x, a) := r(x, a) + \gamma \int q(x', a') - \tau \mathrm{KL}(\pi_{x'} \,\|\, \pi_{x'}^{\mathsf{ref}}) \, \mathrm{d}\check{P}_{x,a}^\pi(x', a'),$$

but only on a restricted collection of policies.

**Lemma A.6.** *If $r \in M_b(\mathsf{X} \times \mathsf{A})$, $\gamma < 1$, and $\pi$ is such that* (4.1) *holds with $p = 1$, then the $\mathcal{B}_\tau^\pi$ is contractive on $M_b(\mathsf{X} \times \mathsf{A})$ endowed with the supnorm. Its unique fixed point is $q_\tau^\pi$.*

*Proof.* Observe that

$$\|\mathcal{B}_\tau^\pi q\|_{\sup} \le \|r\|_{\sup} + \gamma\|q\|_{\sup} + \gamma \sup_{x,a} \|\tau\mathrm{KL}(\pi_- \| \pi_-^{\mathsf{ref}})\|_{L^1(P_{x,a})} < \infty,$$

by (4.1), and

$$\|\mathcal{B}_\tau^\pi q - \mathcal{B}_\tau^\pi q'\|_{\sup} \le \gamma\|\mathcal{V}_\tau q - \mathcal{V}_\tau q'\|_{\sup} \le \gamma\|q - q'\|_{\sup}.$$

$\square$

Next, we proceed with policy improvement.

**Lemma A.7.** *If $r \in M_b(\mathsf{X}\times\mathsf{A})$ and $\gamma < 1$, then the soft Bellman optimality operator is a contraction on $M_b(\mathsf{X} \times \mathsf{A})$ endowed with the supremum norm. Thus, it has a unique fixed point $q_\tau^\star$.*

*Proof.* Observe that

$$\|\mathcal{B}_\tau^\star q\|_{\sup} \le \|r\|_{\sup} + \gamma\|q\|_{\sup} < \infty$$

and

$$\|\mathcal{B}_\tau^\star q - \mathcal{B}_\tau^\star q'\|_{\sup} \le \gamma\|\mathcal{V}_\tau q - \mathcal{V}_\tau q'\|_{\sup} \le \gamma\|q - q'\|_{\sup}.$$

$\square$

**Lemma A.8.** *The following equality holds true: $q_\tau^{\mathcal{G}_\tau q_\tau^\star} = q_\tau^\star$.*

*Proof.* Observe that

$$\begin{aligned}
\mathcal{B}_\tau^{\mathcal{G}_\tau q_\tau^\star} q_\tau^\star &= r(x,a) + \gamma \int q_\tau^\star - \tau\mathrm{KL}((\mathcal{G}_\tau q_\tau^\star)_- \| \pi_-^{\mathsf{ref}}) \,\mathrm{d}\check{P}_{x,a}^{\mathcal{G}_\tau q_\tau^\star} \\
&= r(x,a) + \gamma \int \mathcal{V}_\tau q_\tau^\star \,\mathrm{d}P_{x,a} \\
&= \mathcal{B}_\tau^\star q_\tau^\star \\
&= q_\tau^\star.
\end{aligned}$$

In words, $q_\tau^\star$ is a fixed point of the soft Bellman (policy evaluation) operator with $\pi = \mathcal{G}_\tau q_\tau^\star$. As $\mathcal{G}_\tau q_\tau^\star$ is a Boltzmann–Gibbs policy with a bounded potential, by Lemma A.6 and the preceding note, this operator is a contraction with a unique fixed point. Hence,

$$q_\tau^\star = q_\tau^{\mathcal{G}_\tau q_\tau^\star},$$

the unique fixed point of $\mathcal{B}_\tau^\pi$ with $\pi = \mathcal{G}_\tau q_\tau^\star$, as desired. $\square$

**Lemma A.9.** *For every $\pi \in \mathsf{K}(\mathsf{X}, \mathscr{P}(\mathsf{A}))$, we have that*

$$q_\tau^\star \ge q_\tau^\pi.$$

*Proof.* First, we prove that

$$\mathcal{B}_\tau^\star q_\tau^\pi \ge q_\tau^\pi. \tag{A.4}$$

By definition and the Donsker–Varadhan variational principle,

$$\begin{aligned}
q_\tau^\pi(x,a) &= r(x,a) + \gamma \int \left[ \int q_\tau^\pi(x',a')\,\mathrm{d}\pi_{x'}(a') - \tau\mathrm{KL}(\pi_x' \| \pi_{x'}^{\mathsf{ref}}) \right] \mathrm{d}P_{x,a}(x') \\
&\le r(x,a) + \gamma \int (\mathcal{V}_\tau q_\tau^\pi)(x')\,\mathrm{d}P_{x,a}(x') \\
&= (\mathcal{B}_\tau^\star q_\tau^\pi)(x,a).
\end{aligned}$$

Now we conclude. Let $q_{\tau,0}^\pi := \max\{q_\tau^\pi, 0\}$. By (A.4) and since $\mathcal{B}_\tau^\star$ is a monotone operator,

$$q_\tau^\pi \le \mathcal{B}_\tau^\star q_\tau^\pi \le \mathcal{B}_\tau^\star q_{\tau,0}^\pi \le \mathcal{B}_\tau^\star(\mathcal{B}_\tau^\star q_{\tau,0}^\pi) \le \cdots \le \lim_{n\to\infty}(\mathcal{B}_\tau^\star)^n q_{\tau,0}^\pi = q_\tau^\star,$$

where the final equality holds by Lemma A.7, noting that $\|q_{\tau,0}^\pi\|_{\sup} < \infty$. $\square$

Finally, we prove Theorem 2.3.

**Theorem 2.3.** *Let $\tau > 0$. The policy $\pi^{\tau,\star} := \mathcal{G}_\tau q_\tau^\star$ is optimal, and uniquely so. More precisely, for all $\nu_0, \nu_0' \in \mathscr{P}(\mathsf{X})$, we have that $\arg\max_{\mathscr{O}(\nu_0)} \mathcal{J}_\tau = \pi^{\tau,\star} = \arg\max_{\mathscr{O}(\nu_0')} \mathcal{J}_\tau$.* [Source]

*Proof.* For any $\pi \in \mathsf{K}(\mathsf{X}, \mathscr{P}(\mathsf{A}))$, let

$$v_\tau^\pi(x) := \int q_\tau^\pi(x, a)\, \mathrm{d}\pi_x(a) - \tau \mathrm{KL}(\pi_x \,\|\, \pi_x^{\mathsf{ref}}).$$

Note that

$$\mathcal{J}_\tau(\mu^\pi) = (1 - \gamma) \int v_\tau^\pi \, \mathrm{d}\nu_0$$

if $\mu^\pi \in \mathscr{O}(\nu_0)$. Hence, it suffices to show that

$$v_\tau^{\mathcal{G}_\tau q_\tau^\star} \geq \sup_\pi v_\tau^\pi. \tag{A.5}$$

Observe, by Lemma A.8,

$$v_\tau^{\mathcal{G}_\tau q_\tau^\star}(x) = \int q_\tau^\star(x, a)\, \mathrm{d}(\mathcal{G}_\tau q_\tau^\star)_x(a) - \tau\mathrm{KL}((\mathcal{G}_\tau q_\tau^\star)_x \,\|\, \pi_x^{\mathsf{ref}}) = (\mathcal{V}_\tau q_\tau^\star)(x).$$

Thus, by the Donsker–Varadhan variational principle,

$$v_\tau^{\mathcal{G}_\tau q_\tau^\star}(x) - v_\tau^\pi(x) = (\mathcal{V}_\tau q_\tau^\star)(x) - \int q_\tau^\pi(x, a)\, \mathrm{d}\pi_x(a) + \tau\mathrm{KL}(\pi_x \,\|\, \pi_x^{\mathsf{ref}})$$
$$\geq (\mathcal{V}_\tau q_\tau^\star)(x) - (\mathcal{V}_\tau q_\tau^\pi)(x).$$

Finally, by Lemma A.9, we have that

$$\mathcal{V}_\tau q_\tau^\star - \mathcal{V}_\tau q_\tau^\pi \geq 0,$$

for all $\pi$, as desired. $\qquad\square$

To conclude this section, we prove Theorem 2.5.

**Theorem 2.5.** *Suppose that $r \in M_b(\mathsf{X} \times \mathsf{A})$ and that $\mathsf{X} \times \mathsf{A}$ is finite. For every $\tau > 0$, let $\mu_\tau^\star$ be the maximizer of $\mathcal{J}_\tau$ over $\mathscr{O}(\nu_0)$. If Assumption 2.4 holds, the sequence $(\mu_\tau^\star)_{\tau>0}$ has a unique setwise limit as $\tau$ tends to zero. This limit $\mu_0^\star$ is the minimizer of $\mathcal{R}$ over $\arg\sup_{\mathscr{O}(\nu_0)} \mathcal{J}_0$.* [Source]

*Proof.* Let $\mu^\star \in \{\arg\sup_{\mathscr{O}(\nu_0)} \mathcal{J}_0\} \cap \{\mathcal{R} < \infty\}$. Then,

$$0 \leq \mathcal{J}_0(\mu^\star) - \mathcal{J}_0(\mu_\tau^\star) \leq \tau(\mathcal{R}(\mu^\star) - \mathcal{R}(\mu_\tau^\star)) < \infty.$$

In turn, for all $\tau > 0$, we deduce that $\mathcal{R}(\mu_\tau^\star) \leq \mathcal{R}(\mu^\star)$.

Now let $\mu_0$ be any limit of any setwise convergent subsequence of $(\mu_\tau^\star)_{\tau>0}$ (cf. the proof of Theorem A.5 and Remark A.4). As $\mathcal{R}$ is weakly lower semi-continuous we find that

$$\mathcal{R}(\mu_0) \leq \liminf_{\tau \to 0} \mathcal{R}(\mu_\tau^\star) \leq \mathcal{R}(\mu^\star).$$

Moreover, since $\mathcal{R}(\mu^\star) < \infty$, by Lemma B.2, and as $r \in M_b(\mathsf{X} \times \mathsf{A})$, we deduce that

$$\lim_{\tau \to 0} \tau(\mathcal{R}(\mu^\star) - \mathcal{R}(\mu_\tau^\star)) = 0 \quad \text{and} \quad \mathcal{J}_0(\mu_0) = \mathcal{J}_0(\mu^\star).$$

Therefore, $\mu_0 \in \arg\sup_{\mathscr{O}(\nu_0)} \mathcal{J}_0$ and minimizes $\mathcal{R}$ over $\arg\sup_{\mathscr{O}(\nu_0)} \mathcal{J}_0$.

Since $\mathcal{R}$ is strictly convex, by Lemma 2.1, and the set $\arg\sup_{\mathscr{O}(\nu_0)} \mathcal{J}_0$ is convex, $\mathcal{R}$ has at most one minimizer among this set. In turn, only one such limit $\mu_0$ exists, call it $\mu_0^\star$. Hence, $\mu_\tau^\star \to \mu_0^\star$ setwise, as desired. $\qquad\square$

# B  Proofs for Section 3

Before proving the results from Section 3, we introduce some helpful notation. For any $q : X \times A \to \mathbb{R}$, we define

$$(\mathcal{M}_\tau q)(x) := \int q(x, \cdot) \, \mathrm{d}(\mathcal{G}_\tau q)_x.$$

Additionally, we will define $M_\tau : L^\infty(X \times A) \to \mathbb{R} \cup \{\infty\}$ according to

$$M_\tau(q) := \sup_x \{\operatorname{ess\,sup}_{\pi_x^{\mathrm{ref}}} q(x, \cdot) - \mathcal{V}_\tau q(x)\}.$$

We start by proving that $\mathcal{B}_{\mathrm{ref}}^\star$ is contractive on $M_b(X \times A)$.

**Lemma 3.1.** *Let $r \in M_b(X \times A)$, $\gamma < 1$, and $\mathcal{B}_{\mathrm{ref}}^\star : M(X \times A) \to M(X \times A)$ be defined by*

$$(\mathcal{B}_{\mathrm{ref}}^\star q)(x, a) := r(x, a) + \gamma \int \operatorname{ess\,sup}_{\pi_{x'}^{\mathrm{ref}}} q(x', \cdot) \, \mathrm{d}P_{x,a}(x').$$

*Then $\mathcal{B}_{\mathrm{ref}}^\star$ is a contraction on $M_b(X \times A)$. Thus, it has a unique fixed point $q_{\mathrm{ref}}^\star$.*            [Source]

*Proof.* First, observe that

$$\|\mathcal{B}_{\mathrm{ref}}^\star q\|_{\sup} \leq \|r\|_{\sup} + \gamma \|q\|_{\sup}.$$

Second,

$$\begin{aligned}
\|\mathcal{B}_{\mathrm{ref}}^\star q - \mathcal{B}_{\mathrm{ref}}^\star q'\|_{\sup} &\leq \gamma \sup_{x'} |\operatorname{ess\,sup}_{\pi_{x'}^{\mathrm{ref}}} q(x', \cdot) - \operatorname{ess\,sup}_{\pi_{x'}^{\mathrm{ref}}} q'(x', \cdot)| \\
&\leq \gamma \sup_{x'}(\operatorname{ess\,sup}_{\pi_{x'}^{\mathrm{ref}}} |q(x', \cdot) - q'(x', \cdot)|) \\
&\leq \gamma \|q - q'\|_{\sup}.
\end{aligned}$$

The lemma follows by the Banach fixed point theorem.            $\square$

Next we prove value function convergence.

**Theorem 3.2.** *We have that $q_\tau^\star \to q_{\mathrm{ref}}^\star$ monotonically as $\tau \to 0$.*            [Source]

*Proof.* Since $q_\tau^\star$ is bounded (as the fixed point of a contractive operator on $M_b(X \times A)$, there exists $q_0 : X \times A \to \mathbb{R}$ such that $q_\tau^\star \to q_0$ monotonically and pointwise as $\tau \to 0$, as a direct consequence of Lemma B.1. Therefore, by the monotone convergence theorem,

$$\lim_{\sigma \to 0} \mathcal{V}_\tau q_\sigma^\star(x) = \lim_{\sigma \to 0} \log \|\exp(q_\sigma^\star(x, \cdot))\|_{L^{1/\tau}(\pi_x^{\mathrm{ref}})} = \log \|\exp(q_0(x, \cdot))\|_{L^{1/\tau}(\pi_x^{\mathrm{ref}})}.$$

Consequently,

$$\begin{aligned}
\lim_{\tau \to 0} \lim_{\sigma \to 0} \mathcal{V}_\tau q_\sigma^\star(x) &= \lim_{\tau \to 0} \log \|\exp(q_0(x, \cdot))\|_{L^{1/\tau}(\pi_x^{\mathrm{ref}})} \\
&= \log \|\exp(q_0(x, \cdot))\|_{L^\infty(\pi_x^{\mathrm{ref}})} \\
&= \operatorname{ess\,sup}_{\pi_x^{\mathrm{ref}}} q_0(x, \cdot).
\end{aligned}$$

The second step holds since for any $f \in L^\infty$, $\|f\|_p$ converges up to $\|f\|_\infty$ as $p \to \infty$. So, since the sequence $(\mathcal{V}_\tau q_\sigma^\star(x))_{\tau, \sigma \geq 0}$ is monotone and bounded, its limit exists, and coincides with that computed above:

$$\lim_{\tau \to 0} \mathcal{V}_\tau q_\tau^\star(x) = \operatorname{ess\,sup}_{\pi_x^{\mathrm{ref}}} q_0(x, \cdot).$$

Since $q_\tau^\star$ is the unique fixed point of $\mathcal{B}_\tau^\star$, by the monotone convergence theorem, we have

$$\begin{aligned}
q_0(x, a) &= \lim_{\tau \to 0} q_\tau^\star(x, a) \\
&= \lim_{\tau \to 0} (\mathcal{B}_\tau^\star q_\tau^\star)(x, a) \\
&= r(x, a) + \gamma \int \lim_{\tau \to 0} \mathcal{V}_\tau q_\tau^\star(x') \, \mathrm{d}P_{x,a}(x') \\
&= r(x, a) + \gamma \int \operatorname{ess\,sup}_{\pi_{x'}^{\mathrm{ref}}} q_0(x', \cdot) \, \mathrm{d}P_{x,a}(x'),
\end{aligned}$$

so that $q_0$ is a fixed point of $\mathcal{B}_{\mathrm{ref}}^\star$. Since the $\mathcal{B}_{\mathrm{ref}}^\star$ has a fixed point $q_{\mathrm{ref}}^\star$, it follows that $q_0 = q_{\mathrm{ref}}^\star$.            $\square$

Now we prove our core estimate.

**Theorem 3.6.** *Let $q, q' \in M(\mathsf{X} \times \mathsf{A})$. For any $\tau > 0$ and any $x \in \mathsf{X}$,*

$$\|(\mathcal{G}_\tau q)_x - (\mathcal{G}_\tau q')_x\|_{\mathrm{TV}}$$
$$\leq \min \left\{ \sqrt{\tau^{-1} \|q(x, \cdot) - q'(x, \cdot)\|_{L^\infty(\pi_x^{\mathrm{ref}})}}, \frac{1}{2} \sinh \left( 4\tau^{-1} \|q(x, \cdot) - q'(x, \cdot)\|_{L^\infty(\pi_x^{\mathrm{ref}})} \right) \right\}.$$

*In particular,*

$$\|(\mathcal{G}_\tau q)_x - (\mathcal{G}_\tau q')_x\|_{\mathrm{TV}} \leq \frac{2e - 3}{4} \tau^{-1} \|q(x, \cdot) - q'(x, \cdot)\|_{L^\infty(\pi_x^{\mathrm{ref}})},$$

*if $\|q(x, \cdot) - q'(x, \cdot)\|_{L^\infty(\pi_x^{\mathrm{ref}})} < \tau/2$.*    [Source]

*Proof.* Let $\pi := \mathcal{G}_\tau q, \pi' := \mathcal{G}_\tau q'$. By Lemma B.6, we have

$$\|\pi_x - \pi'_x\|_{\mathrm{TV}} \leq \sqrt{\tau^{-1} \|q(x, \cdot) - q'(x, \cdot)\|_{L^\infty(\pi^{\mathrm{ref}})}}.$$

Moreover, by Lemma B.9,

$$\|\pi_x - \pi'_x\|_{\mathrm{TV}} \leq \frac{1}{2} \sinh \left( 4\tau^{-1} \|q(x, \cdot) - q'(x, \cdot)\|_{L^\infty(\pi_x^{\mathrm{ref}})} \right).$$

This concludes the proof of the first claim. Next, we recall that

$$\sinh(y) = \sum_{k=0}^\infty \frac{y^{2k+1}}{(2k+1)!},$$

which is convergent for any $y \in \mathbb{C}$. Therefore, for $y \in (0, 1)$, we have

$$\sinh(y) \leq y + \frac{y^3}{3!} + \frac{y^5}{5!} + \dots$$
$$\leq y \left( 1 + e^y - \frac{5}{2} \right)$$
$$\leq y \left( e - \frac{3}{2} \right).$$

So, when $\|q(x, \cdot) - q'(x, \cdot)\|_{L^\infty(\pi^{\mathrm{ref}})} < \tau/2$, it follows that

$$\|\pi_x - \pi'_x\|_{\mathrm{TV}} \leq \frac{1}{2} \left( e - \frac{3}{2} \right) \left( 4\tau^{-1} \|q(x, \cdot) - q'(x, \cdot)\|_{L^\infty(\pi_x^{\mathrm{ref}})} \right)$$
$$= \frac{2e - 3}{4} \tau^{-1} \|q(x, \cdot) - q'(x, \cdot)\|_{L^\infty(\pi_x^{\mathrm{ref}})}.$$

$\square$

Finally, we prove policy and return distribution convergence.

**Theorem 3.9.** *Under Assumption 3.4, if $\sigma = \sigma(\tau)$ is such that $\lim_{\tau \to 0} \sigma/\tau = 0$, then $\pi_x^{\tau, \sigma} \to \pi_x^{\mathrm{ref}, \star}$ as $\tau \to 0$, for all $x \in \mathsf{X}$, in $\mathrm{TV}$ if $\mathsf{A}$ is discrete and weakly if $\mathsf{A}$ is continuous.*    [Source]

*Proof.* Recall $\pi^{\tau, \sigma} := \mathcal{G}_\tau q_\sigma^\star$. By Theorem 3.6 and Lemma B.10,

$$\lim_{\tau \to 0} \sup_x \|(\mathcal{G}_\tau q_\sigma^\star)_x - (\mathcal{G}_\tau q_{\mathrm{ref}}^\star)_x\|_{\mathrm{TV}} \lesssim - \lim_{\tau \to 0} \frac{\sigma}{\tau} \log p_{\mathrm{ref}}. \tag{B.1}$$

Consequently, $\pi_x^{\tau, \sigma} \to \pi_x^{\mathrm{ref}, \star}$ if and only if $(\mathcal{G}_\tau q_{\mathrm{ref}}^\star)_x \to \pi_x^{\mathrm{ref}, \star}$, and in whatever sense the later convergence occurs. In particular, if $\mathsf{A}$ is continuous, this is in the weak sense. While, if $\mathsf{A}$ is discrete, this in total variation.    $\square$

**Theorem 3.10.** *Suppose* $A$ *is discrete and Assumption 3.4 holds. If* $\sigma = \sigma(\tau)$ *is such that* $\sigma/\tau \to 0$ *as* $\tau \to 0$, *then, for any* $p, p' \in [1, \infty)$ *and* $\omega \in \mathscr{P}(X \times A)$, *as* $\tau \to 0$, *the return distribution functions* $\zeta^{\tau,\sigma}$ *of the temperature-decoupled policies* $\pi^{\tau,\sigma}$ *satisfy* $d_{p;p',\omega}(\zeta^{\tau,\sigma}, \zeta^{\pi^{\mathrm{ref},\star}}) \to 0$. 

*Proof.* By the distributional Bellman equation [5], we have that

$$d_p(\zeta^{\star}_{x,a}, \zeta^{\tau,\sigma}_{x,a})$$

$$\leq \int d_p\left((\mathtt{b}_{r(x,a),\gamma} \circ \mathtt{proj}^{\mathbb{R}})_\#(\zeta^{\star}_{x',\_} \otimes \pi^{\mathrm{ref},\star}_{x'}), (\mathtt{b}_{r(x,a),\gamma} \circ \mathtt{proj}^{\mathbb{R}})_\#(\zeta^{\tau,\sigma}_{x',\_} \otimes \pi^{\tau,\sigma}_{x'})\right) \mathrm{d}P_{x,a}(x')$$

$$= \int d_p\left((\mathtt{b}_{0,\gamma} \circ \mathtt{proj}^{\mathbb{R}})_\#(\zeta^{\star}_{x',\_} \otimes \pi^{\mathrm{ref},\star}_{x'}), (\mathtt{b}_{0,\gamma} \circ \mathtt{proj}^{\mathbb{R}})_\#(\zeta^{\tau,\sigma}_{x',\_} \otimes \pi^{\tau,\sigma}_{x'})\right) \mathrm{d}P_{x,a}(x')$$

$$= \gamma \int \mathrm{I}_{\tau,\sigma} \, \mathrm{d}P_{x,a}$$

where

$$\mathrm{I}_{\tau,\sigma}(x') = d_p\left((\mathtt{b}_{0,1} \circ \mathtt{proj}^{\mathbb{R}})_\#(\zeta^{\star}_{x',\_} \otimes \pi^{\mathrm{ref},\star}_{x'}), (\mathtt{b}_{0,1} \circ \mathtt{proj}^{\mathbb{R}})_\#(\zeta^{\tau,\sigma}_{x',\_} \otimes \pi^{\tau,\sigma}_{x'})\right).$$

We now derive a bound on $\mathrm{I}_{\tau,\sigma}$. Starting with a triangle inequality,

$$\mathrm{I}_{\tau,\sigma}(x') \leq d_p\left((\mathtt{b}_{0,1} \circ \mathtt{proj}^{\mathbb{R}})_\#(\zeta^{\star}_{x',\_} \otimes \pi^{\mathrm{ref},\star}_{x'}), (\mathtt{b}_{0,1} \circ \mathtt{proj}^{\mathbb{R}})_\#(\zeta^{\tau,\sigma}_{x',\_} \otimes \pi^{\mathrm{ref},\star}_{x'})\right)$$

$$+ d_p\left((\mathtt{b}_{0,1} \circ \mathtt{proj}^{\mathbb{R}})_\#(\zeta^{\tau,\sigma}_{x',\_} \otimes \pi^{\mathrm{ref},\star}_{x'}), (\mathtt{b}_{0,1} \circ \mathtt{proj}^{\mathbb{R}})_\#(\zeta^{\tau,\sigma}_{x',\_} \otimes \pi^{\tau,\sigma}_{x'})\right)$$

$$\overset{(a)}{\leq} \int d_p(\zeta^{\star}_{x',a'}, \zeta^{\tau,\sigma}_{x',a'}) \, \mathrm{d}\pi^{\mathrm{ref},\star}_{x'}(a')$$

$$+ d_p\left((\mathtt{b}_{0,1} \circ \mathtt{proj}^{\mathbb{R}})_\#(\zeta^{\tau,\sigma}_{x',\_} \otimes \pi^{\mathrm{ref},\star}_{x'}), (\mathtt{b}_{0,1} \circ \mathtt{proj}^{\mathbb{R}})_\#(\zeta^{\tau,\sigma}_{x',\_} \otimes \pi^{\tau,\sigma}_{x'})\right)$$

$$\overset{(b)}{\leq} \int d_p(\zeta^{\star}_{x',a'}, \zeta^{\tau,\sigma}_{x',a'}) \, \mathrm{d}\pi^{\mathrm{ref},\star}_{x'}(a')$$

$$+ 2^{\frac{p-1}{p}} \left(\int \left[\int |z|^p \, \mathrm{d}\zeta^{\tau,\sigma}_{x',a'}(z)\right] \mathrm{d}|\pi^{\mathrm{ref},\star}_{x'} - \pi^{\tau,\sigma}_{x'}|(a')\right)^{1/p}$$

$$\overset{(c)}{\leq} \int d_p(\zeta^{\star}_{x',a'}, \zeta^{\tau,\sigma}_{x',a'}) \, \mathrm{d}\pi^{\mathrm{ref},\star}_{x'}(a') + \frac{2^{\frac{p-1}{p}}}{1-\gamma} \|r\|_{\sup} \|\pi^{\mathrm{ref},\star}_{x'} - \pi^{\tau,\sigma}_{x'}\|^{1/p}_{\mathrm{TV}},$$

where $(a)$ follows by the convexity of the Wasserstein metrics [40, 6], $(b)$ applies [40, Theorem 6.15], and $(c)$ leverages that the support $\zeta^{\pi}_{x',a'}$ lives in a ball of radius $\|r\|_{\sup}/(1-\gamma)$, for any $\pi$ and $(x', a') \in X \times A$.

So, thus far, we have shown that

$$d_p(\zeta^{\star}_{x,a}, \zeta^{\tau,\sigma}_{x,a}) \leq \gamma \int d_p(\zeta^{\star}_{x',a'}, \zeta^{\tau,\sigma}_{x',a'}) \, \mathrm{d}\check{P}^{\pi^{\mathrm{ref},\star}}_{x,a}(x', a') + C\gamma \int \|\pi^{\mathrm{ref},\star}_{x'} - \pi^{\tau,\sigma}_{x'}\|^{1/p}_{\mathrm{TV}} \, \mathrm{d}P_{x,a}(x').$$

By Theorem 3.9, the total variation term tends to zero as $\tau$ tends to zero. Thus, defining $\iota(x', a') := \limsup_{\tau \to 0} d_p(\zeta^{\star}_{x',a'}, \zeta^{\tau,\sigma}_{x',a'})$, this implies that

$$\iota(x', a') \leq \gamma \int \iota(y, b) \, \mathrm{d}\check{P}^{\pi^{\mathrm{ref},\star}}_{x',a'}(y, b),$$

In turn, $\sup \iota \leq \gamma \sup \iota$, implying that $\iota \equiv 0$. Therefore, $d_p(\zeta^{\star}_{x,a}, \zeta^{\tau,\sigma}_{x,a}) \to 0$ pointwise over $X \times A$, so by the dominated convergence theorem, $d_{p;p',\omega}(\zeta^{\star}, \zeta^{\tau,\sigma}) \to 0$ for any $p' \in [1, \infty)$ and $\omega \in \mathscr{P}(X \times A)$. $\qquad\square$

## B.1   Supplemental Lemmas for Section 3

The following lemma translates immediately from the corresponding result in tabular MDPs; we prove it here for completeness.

**Lemma B.1.** *If $\tau \leq \sigma$, then $q_\sigma^\star \leq q_\tau^\star$.*

*Proof.* By the monotonicity of $\mathcal{B}_\tau^\star$,
$$q_\sigma^\star = r + \gamma \int \mathcal{V}_\sigma q_\sigma^\star \, \mathrm{d}P_{-,-} \leq r + \gamma \int \mathcal{V}_\tau q_\sigma^\star \, \mathrm{d}P_{-,-} = \mathcal{B}_\tau^\star q_\sigma^\star \leq \cdots \leq q_\tau^\star$$
(cf. the proof of Lemma A.9). $\qquad\square$

**Lemma B.2.** *Let $\sigma = \sigma(\tau)$ and suppose $\sigma \to 0$ as $\tau \to 0$. Then*
$$\tau \mathrm{KL}((\mathcal{G}_\tau q_\sigma^\star)_x \, \| \, \pi_x^{\mathsf{ref}}) \xrightarrow{\tau \downarrow 0} 0.$$

*Proof.* Expanding the KL, we have
$$\tau \mathrm{KL}((\mathcal{G}_\tau q_\sigma^\star)_x \, \| \, \pi_x^{\mathsf{ref}}) = \int_\mathsf{A} (q_\sigma^\star(x, \cdot) - \mathcal{V}_\tau q_\sigma^\star(x)) \, \mathrm{d}(\mathcal{G}_\tau q_\sigma^\star)_x$$
$$\leq \operatorname{ess\,sup}_{\pi_x^{\mathsf{ref}}} q_\sigma^\star(x, \cdot) - (\mathcal{V}_\tau q_\sigma^\star)(x)$$
$$\leq v_{\mathsf{ref}}^\star(x) - (\mathcal{V}_\tau q_\sigma^\star)(x).$$
where the final inequality holds by Lemma B.1. Since $\sigma = \sigma(\tau) \leq \tau$, we have
$$\mathcal{V}_\tau q_\sigma^\star(x) = \log \| \exp(q_\sigma^\star(x, \cdot)) \|_{L^{1/\tau}(\pi_x^{\mathsf{ref}})} \geq \log \| \exp(q_\tau^\star(x, \cdot)) \|_{L^{1/\tau}(\pi_x^{\mathsf{ref}})} = \mathcal{V}_\tau q_\tau^\star,$$
where the inequality again is due to Lemma B.1. Consequently, we have
$$\limsup_{\tau \to 0} \tau \mathrm{KL}(\pi_x^{\tau,\sigma} \, \| \, \pi_x^{\mathsf{ref}}) \leq v_{\mathsf{ref}}^\star(x) - \liminf_{\tau \to 0} \mathcal{V}_\tau q_\tau^\star(x) = v_{\mathsf{ref}}^\star(x) - v_{\mathsf{ref}}^\star(x) = 0,$$
where the penultimate step is due to the fact that $\mathcal{V}_\tau q_\tau^\star \to v_{\mathsf{ref}}^\star$ monotonically, as shown in the proof of Theorem 3.2. $\qquad\square$

**Lemma B.3.** *For every $q \in M_b(\mathsf{X} \times \mathsf{A})$, with the notation above, $M_\tau(q) \to 0$ as $\tau \to 0$. If Assumption 3.4 is satisfied, then for any $\sigma > 0$,*
$$M_\tau(q_\sigma^\star) \leq -\tau \log p_{\mathsf{ref}}.$$

*Proof.* First, we observe that
$$\lim_{\tau \to 0} \mathcal{V}_\tau q(x) = \lim_{\tau \to 0} \log \| \exp(q(x, \cdot)) \|_{L^{1/\tau}(\pi^{\mathsf{ref}})} = \log \| \exp(q(x, \cdot)) \|_{L^\infty(\pi^{\mathsf{ref}})} = \operatorname{ess\,sup}_{\pi_x^{\mathsf{ref}}} q(x, \cdot).$$
This is a monotone limit in $\tau$, as it is known that for any $f \in L^\infty$, $\|f\|_p$ converges up to $\|f\|_{L^\infty}$ as $p \to \infty$. Thus, we see that
$$M_\tau(q) = \sup_x \left( \operatorname{ess\,sup}_{\pi_x^{\mathsf{ref}}} q(x, \cdot) - \mathcal{V}_\tau q(x) \right) \to \sup_x (\operatorname{ess\,sup}_{\pi_x^{\mathsf{ref}}} q(x, \cdot) - \operatorname{ess\,sup}_{\pi_x^{\mathsf{ref}}} q(x, \cdot)) = 0$$
as claimed. Now, under Assumption 3.4, we have
$$M_\tau(q_\sigma^\star) = \sup_x (\operatorname{ess\,sup}_{\pi_x^{\mathsf{ref}}} q_\sigma^\star(x, \cdot) - \mathcal{V}_\tau q_\sigma^\star(x))$$
$$= \sup_x \left( \operatorname{ess\,sup}_{\pi_x^{\mathsf{ref}}} q_\sigma^\star(x, \cdot) - \tau \log \int \exp(\tau^{-1} q_\sigma^\star(x, \cdot)) \, \mathrm{d}\pi_x^{\mathsf{ref}} \right)$$
$$= \sup_x \left( -\tau \log \int \exp(\tau^{-1}(q_\sigma^\star(x, \cdot) - \operatorname{ess\,sup}_{\pi_x^{\mathsf{ref}}} q_\sigma^\star(x, \cdot)) \, \mathrm{d}\pi_x^{\mathsf{ref}} \right).$$
Let $\mathsf{B}_x = \{a \in \mathsf{A} : q_\sigma^\star(x, a) = \operatorname{ess\,sup}_{\pi_x^{\mathsf{ref}}} q_\sigma^\star(x, \cdot)\}$. Then,
$$M_\tau(q_\sigma^\star) = \sup_x \left[ -\tau \log \left( \int_{\mathsf{B}_x} \exp(\tau^{-1}(q_\sigma^\star(x, \cdot) - \operatorname{ess\,sup}_{\pi_x^{\mathsf{ref}}} q_\sigma^\star(x, \cdot)) \, \mathrm{d}\pi_x^{\mathsf{ref}} \right. \right.$$
$$\left. \left. + \int_{\mathsf{A} \backslash \mathsf{B}_x} \exp(\tau^{-1}(q_\sigma^\star(x, \cdot) - \operatorname{ess\,sup}_{\pi_x^{\mathsf{ref}}} q_\sigma^\star(x, \cdot)) \, \mathrm{d}\pi_x^{\mathsf{ref}} \right) \right]$$
$$= \sup_x \left[ -\tau \log \left( \pi_x^{\mathsf{ref}}(\mathsf{B}_x) + \int_{\mathsf{A} \backslash \mathsf{B}_x} \exp(\tau^{-1}(q_\sigma^\star(x, \cdot) - \operatorname{ess\,sup}_{\pi_x^{\mathsf{ref}}} q_\sigma^\star(x, \cdot)) \, \mathrm{d}\pi_x^{\mathsf{ref}} \right) \right]$$
$$\leq \sup_x -\tau \log \pi_x^{\mathsf{ref}}(\mathsf{B}_x)$$
$$\leq \tau \log p_{\mathsf{ref}},$$
where the final inequality invokes Assumption 3.4. $\qquad\square$

**Lemma B.4.** *For all $\tau > 0$ and any $q \in L^\infty(\mathsf{X} \times \mathsf{A})$,*

$$\mathcal{B}^\star_{\text{ref}} q \geq \mathcal{B}^\star_\tau q,$$

*where $\mathcal{B}^\star_{\text{ref}}$ denotes the Bellman optimality operator (cf. Lemma 3.1).*

*Proof.* A direct calculation gives

$$\mathcal{V}_\tau q(x) = \tau \log \left( \int \exp(\tau^{-1} q(x, a)) \, d\pi^{\text{ref}}_x(a) \right)$$

$$\leq \tau \log \left( \int \exp(\tau^{-1} \operatorname{ess\,sup}_{\pi^{\text{ref}}_x} q(x, \cdot)) \, d\pi^{\text{ref}}_x \right)$$

$$= \operatorname{ess\,sup}_{\pi^{\text{ref}}_x} q(x, \cdot).$$

Therefore, it immediate follows that

$$(\mathcal{B}^\star_{\text{ref}} q)(x, a) = r(x, a) + \gamma \int \operatorname{ess\,sup}_{\pi^{\text{ref}}_{x'}} q(x', \cdot) \, dP_{x,a}(x')$$

$$\geq r(x, a) + \gamma \int \mathcal{V}_\tau q(x') \, dP_{x,a}(x')$$

$$= (\mathcal{B}^\star_\tau q)(x, a).$$

$\square$

The follow proof is essentially the performance difference bound in [37].

**Lemma B.5.** *For all $n \geq 1$ and any $\tau > 0$,*

$$(\mathcal{B}^\star_{\text{ref}})^n q^\star_\tau - q^\star_\tau \leq \sum_{k=1}^{n} \gamma^k M_\tau(q^\star_\tau).$$

*If, additionally, Assumption 3.4 is satisfied, then*

$$(\mathcal{B}^\star_{\text{ref}})^n q^\star_\tau - q^\star_\tau \leq -\tau \log p_{\text{ref}} \sum_{k=1}^{n} \gamma^k.$$

*Proof.* We begin with the first statement. Recall that $q^\star_\tau$ is the fixed point of $\mathcal{B}^\star_\tau$, so that $q^\star_\tau = \mathcal{B}^\star_\tau q^\star_\tau$. We will proceed by induction on $n$. For $n = 1$, we observe that

$$(\mathcal{B}^\star_{\text{ref}} q^\star_\tau)(x, a) - q^\star_\tau(x, a) = (\mathcal{B}^\star_{\text{ref}} q^\star_\tau)(x, a) - (\mathcal{B}^\star_\tau q^\star_\tau)(x, a)$$

$$= \gamma \int \left( \operatorname{ess\,sup}_{\pi^{\text{ref}}_{x'}} q^\star_\tau(x', \cdot) - \mathcal{V}_\tau q^\star_\tau(x') \right) dP_{x,a}(x')$$

$$\leq \gamma M_\tau(q^\star_\tau),$$

recalling the notation established above. This proves the base case. Now, assume the statement holds for all $m \leq n$. We have

$$(\mathcal{B}^\star_{\text{ref}})^{n+1} q^\star_\tau - q^\star_\tau = (\mathcal{B}^\star_{\text{ref}})^{n+1} q^\star_\tau - \mathcal{B}^{\star\,n+1}_\tau q^\star_\tau$$

$$\leq \mathcal{B}^\star_{\text{ref}} \left( \mathcal{B}^{\star\,n}_\tau q^\star_\tau + \sum_{k=1}^{n} \gamma^k M_\tau(q^\star_\tau) \right) - \mathcal{B}^{\star\,n+1}_\tau q^\star_\tau$$

$$= \mathcal{B}^\star_{\text{ref}} q^\star_\tau + \sum_{k=1}^{n} \gamma^{k+1} M_\tau(q^\star_\tau) - \mathcal{B}^\star_\tau q^\star_\tau$$

$$\leq \gamma M_\tau(q^\star_\tau) + \sum_{k=2}^{n+1} \gamma^k M_\tau(q^\star_\tau)$$

$$= \sum_{k=1}^{n+1} \gamma^k M_\tau(q^\star_\tau),$$

where the first inequality invokes the induction hypothesis, and the second inequality is due to the base case. Thus, we have shown that the claimed statement holds for any $n \in \mathbb{N}$.

When Assumption 3.4 is satisfied, by Lemma B.3, we have $M_\tau(q_\tau^\star) \leq -\tau \log p_{\text{ref}}$, and the second statement follows. $\qquad\square$

**Lemma B.6.** *Let $q, q' \in L^\infty(\mathsf{X} \times \mathsf{A})$. Then for any $\tau > 0$ and any $x \in \mathsf{X}$,*

$$\|(\mathcal{G}_\tau q)_x - (\mathcal{G}_\tau q')_x\|_{\text{TV}} \leq \sqrt{\tau^{-1}\|q(x, \cdot) - q'(x, \cdot)\|_{L^\infty(\pi_x^{\text{ref}})}}.$$

*Proof.* Let $\pi = \mathcal{G}_\tau q$ and let $\pi' = \mathcal{G}_\tau q'$. By Pinsker's inequality, we have

$$\|\pi_x - \pi_x'\|_{\text{TV}} \leq \sqrt{\frac{1}{2}\text{KL}(\pi_x \,\|\, \pi_x')}.$$

Since $q, q' \in L^\infty(\mathsf{X} \times \mathsf{A})$, $\pi_x, \pi_x'$ are mutually absolutely continuous. Expanding the KL divergence, we have

$$\begin{aligned}
\text{KL}(\pi_x \,\|\, \pi_x') &= \int_\mathsf{A} \log \frac{\pi_x}{\pi_x'} \, \mathrm{d}\pi_x \\
&= \int_\mathsf{A} \log \frac{\pi_x^{\text{ref}}(a) \exp(\tau^{-1}(q(x, a) - \mathcal{V}_\tau q(x)))}{\pi_x^{\text{ref}}(a) \exp(\tau^{-1}(q'(x, a) - \mathcal{V}_\tau q'(x))} \, \mathrm{d}\pi_x(a) \\
&= \int_\mathsf{A} \tau^{-1}\left(q(x, a) - \mathcal{V}_\tau q(x) - q'(x, a) + \mathcal{V}_\tau q'(x)\right) \mathrm{d}\pi_x(a) \\
&\leq \tau^{-1}\|q(x, \cdot) - q'(x, \cdot)\|_{L^\infty(\pi_x^{\text{ref}})} + \tau^{-1}\|\mathcal{V}_\tau q(x) - \mathcal{V}_\tau q'(x)\|_{L^\infty(\pi_x^{\text{ref}})} \\
&\leq 2\tau^{-1}\|q(x, \cdot) - q'(x, \cdot)\|_{L^\infty(\pi_x^{\text{ref}})},
\end{aligned}$$

where the last inequality holds since $\mathcal{V}_\tau$ is 1-Lipschitz, as shown in the proof of Lemma B.4. Substituting back into Pinsker's inequality, we have

$$\|\pi_x - \pi_x'\|_{\text{TV}} \leq \sqrt{\tau^{-1}\|q(x, \cdot) - q'(x, \cdot)\|_{L^\infty(\pi_x^{\text{ref}})}},$$

as claimed. $\qquad\square$

**Lemma B.7.** *Let $\pi, \pi' \in \mathscr{P}(\mathsf{Y})$ for some measurable space $\mathsf{Y}$ be mutually absolutely continuous. Then*

$$\|\pi - \pi'\|_{\text{TV}} \leq \frac{1}{4}\left(\text{ess sup}_{\pi'} \frac{\mathrm{d}\pi}{\mathrm{d}\pi'} - \text{ess inf}_{\pi'} \frac{\mathrm{d}\pi}{\mathrm{d}\pi'}\right).$$

*Proof.* Define $h := \frac{\mathrm{d}\pi}{\mathrm{d}\pi'}$, and write $M := \text{ess sup}_{\pi'} h$, $m := \text{ess inf}_{\pi'} h$. Note that

$$0 = \int_\mathsf{Y} (\mathrm{d}\pi - \mathrm{d}\pi') = \int_\mathsf{Y} (h-1)\mathrm{d}\pi' = \int_\mathsf{E} (h-1)\mathrm{d}\pi' + \int_{\mathsf{Y}\backslash\mathsf{E}} (h-1)\mathrm{d}\pi',$$

for any measurable $\mathsf{E} \subset \mathsf{Y}$. Consequently, we have

$$\int_\mathsf{E} (h-1), \mathrm{d}\pi' = \int_{\mathsf{Y}\backslash\mathsf{E}} (1-h) \, \mathrm{d}\pi'.$$

Now, we derive the following upper bounds,

$$\pi(\mathsf{E}) - \pi'(\mathsf{E}) = \int_\mathsf{E} (h-1) \, \mathrm{d}\pi' \leq (M-1)\pi'(\mathsf{E})$$

$$\pi(\mathsf{E}) - \pi'(\mathsf{E}) = \int_{\mathsf{Y}\backslash\mathsf{E}} (1-h) \, \mathrm{d}\pi' \leq (1-m)\pi'(\mathsf{Y} \backslash \mathsf{E}).$$

Multiplying these inequalities by $\pi'(\mathsf{Y} \backslash \mathsf{E})$ and $\pi'(\mathsf{E})$, respectively, and adding the results, we have

$$\begin{aligned}
(\pi(\mathsf{E}) - \pi'(\mathsf{E}))(\pi'(\mathsf{Y} \backslash \mathsf{E}) + \pi'(\mathsf{E})) &\leq ((M-1) + (1-m))\pi'(\mathsf{E})\pi'(\mathsf{Y} \backslash \mathsf{E}) \\
\therefore \pi(\mathsf{E}) - \pi'(\mathsf{E}) &\leq (M-m)\pi'(\mathsf{E})\pi'(\mathsf{Y} \backslash \mathsf{E}).
\end{aligned}$$

In fact, the same bound can be achieved for $\pi'(\mathsf{E}) - \pi(\mathsf{E})$; to see this, note that

$$\pi'(\mathsf{E}) - \pi(\mathsf{E}) = \int_{\mathsf{E}} (1 - h)\mathrm{d}\pi' \leq (1 - m)\pi'(\mathsf{E})$$

$$\pi'(\mathsf{E}) - \pi(\mathsf{E}) = \int_{\mathsf{Y}\setminus\mathsf{E}} (h - 1)\mathrm{d}\pi' \leq (M - 1)\pi'(\mathsf{E}),$$

so by the same procedure as above, $\pi'(\mathsf{E}) - \pi(\mathsf{E}) \leq (M - m)\pi'(\mathsf{E})\pi'(\mathsf{Y} \setminus \mathsf{E})$. Therefore, we have shown that

$$|\pi(\mathsf{E}) - \pi'(\mathsf{E})| \leq (M - m)\pi'(\mathsf{E})\pi'(\mathsf{Y} \setminus \mathsf{E})$$

for any measurable $\mathsf{E} \subset \mathsf{Y}$. Since $\pi'(\mathsf{E})\pi'(\mathsf{Y} \setminus \mathsf{E})$ is maximized at $\pi'(\mathsf{E}) = \pi'(\mathsf{Y} \setminus \mathsf{E}) = 1/2$, we have

$$\|\pi - \pi'\|_{\mathrm{TV}} = \sup_{\mathsf{E}} |\pi(\mathsf{E}) - \pi'(\mathsf{E})| \leq \frac{1}{4}(M - m),$$

as claimed. $\qquad\square$

**Lemma B.8.** *Let $u, w \in L^\infty(\mathsf{Y})$ for some measurable space $\mathsf{Y}$, and let $\lambda$ be a measure on $\mathsf{Y}$. Define $\pi^u, \pi^w \in \mathscr{P}(\mathsf{Y})$ absolutely continuous with respect to $\lambda$ such that $\frac{\mathrm{d}\pi^\bullet}{\mathrm{d}\lambda} \propto e^{-\bullet}$ for $\bullet \in \{u, w\}$. Then*

$$\|\pi^u - \pi^w\|_{\mathrm{TV}} \leq \frac{1}{2} \sinh\left(2\|u - w\|_{L^\infty(\lambda)}\right).$$

*Proof.* Firstly, since $u, w \in L^\infty(\mathsf{Y})$, it follows that $\pi^u, \pi^w$ are mutually absolutely continuous. Now, define $h := \frac{\mathrm{d}\pi^u}{\mathrm{d}\pi^w}$, with $M := \operatorname{ess\,sup}_\lambda h$ and $m := \operatorname{ess\,inf}_\lambda h$. Note that

$$\mathrm{d}\pi^u(x) = \frac{e^{-u(x)}}{Z_u}\mathrm{d}\lambda(x), \quad \mathrm{d}\pi^w(x) = \frac{e^{-w(x)}}{Z_w}\mathrm{d}\lambda(x),$$

where $Z_u, Z_w \in \mathbb{R}$ are normalizing constants. Defining $f := u - w$, we have

$$h(x) = \frac{Z_w}{Z_u}e^{-f(x)}.$$

Additionally, we have

$$\frac{Z_w}{Z_u} = \frac{\int_{\mathsf{Y}} e^{-w(x)}\,\mathrm{d}\lambda(x)}{\int_{\mathsf{Y}} e^{-w(x)}e^{-f(x)}\,\mathrm{d}\lambda(x)} = \frac{1}{\mathbb{E}_{\pi^w}[e^{-f}]}.$$

Consequently, it holds that $\operatorname{ess\,inf}_\lambda h \geq e^{\operatorname{ess\,inf}_\lambda f - \operatorname{ess\,sup}_\lambda f}$ and $\operatorname{ess\,sup}_\lambda h \leq e^{\operatorname{ess\,sup}_\lambda f - \operatorname{ess\,inf}_\lambda f}$. So, by the definition of $f$, we have $m \geq e^{-2\|u - w\|_{L^\infty(\lambda)}}$ and $M \leq e^{2\|u - w\|_{L^\infty(\lambda)}}$. Then, invoking Lemma B.7, we have

$$\|\pi^u - \pi^w\|_{\mathrm{TV}} \leq \frac{1}{4}\left(e^{2\|u - w\|_{L^\infty(\lambda)}} - e^{-2\|u - w\|_{L^\infty(\lambda)}}\right) = \frac{1}{2}\sinh(2\|u - w\|_{L^\infty(\lambda)}).$$

$\qquad\square$

**Lemma B.9.** *Let $q, q' \in L^\infty(\mathsf{X} \times \mathsf{A})$. Then for any $\tau > 0$ and any $x \in \mathsf{X}$,*

$$\|(\mathcal{G}_\tau q)_x - (\mathcal{G}_\tau q')_x\|_{\mathrm{TV}} \leq \frac{1}{2}\sinh\left(2\tau^{-1}\|q(x, \cdot) - q'(x, \cdot)\|_{L^\infty(\pi_x^{\mathrm{ref}})}\right).$$

*Proof.* Note that, for any $q \in M_b(\mathsf{X} \times \mathsf{A})$, we have

$$\mathrm{d}(\mathcal{G}_\tau q)_x \propto \exp(\tau^{-1}q(x, \cdot))\mathrm{d}\pi_x^{\mathrm{ref}}.$$

So, invoking Lemma B.8 with $u = -\tau^{-1}q(x, \cdot)$, $v = -\tau^{-1}q'(x, \cdot)$, and $\lambda = \pi_x^{\mathrm{ref}}$, we have

$$\|(\mathcal{G}_\tau q)_x - (\mathcal{G}_\tau q')_x\|_{\mathrm{TV}} \leq \frac{1}{2}\sinh\left(2\tau^{-1}\|q - q'\|_{L^\infty(\pi_x^{\mathrm{ref}})}\right).$$

$\qquad\square$

**Lemma B.10.** *For every $\tau > 0$, recalling the notation above,*

$$0 \le q_{\text{ref}}^\star - q_\tau^\star \le \frac{\gamma}{1-\gamma} M_\tau(q_\tau^\star).$$

*If Assumption 3.4 is satisfied, then*

$$M_\tau(q_\tau^\star) \le -\tau \log p_{\text{ref}},$$

*and $q_\tau^\star$ converges uniformly up to $q_{\text{ref}}^\star$.*

*Proof.* By Lemma B.4, we have that for any $q \in L^\infty(\mathsf{X} \times \mathsf{A})$,

$$q_{\text{ref}}^\star - q_\tau^\star = \lim_{n \to \infty} (\mathcal{B}_{\text{ref}}^\star)^n q - \lim_{n \to \infty} \mathcal{B}_\tau^{\star n} q \ge 0.$$

Then, by Lemma B.5, we have

$$
\begin{aligned}
q_{\text{ref}}^\star - q_\tau^\star &= \lim_{n \to \infty} \left( (\mathcal{B}_{\text{ref}}^\star)^n q_\tau^\star - \mathcal{B}_\tau^{\star n} q_\tau^\star \right) \\
&\le \lim_{n \to \infty} M_\tau(q_\tau^\star) \sum_{k=1}^n \gamma^k \\
&= \frac{\gamma}{1-\gamma} M_\tau(q_\tau^\star),
\end{aligned}
$$

proving the first claim. When Assumption 3.4 is satisfied, we have $M_\tau(q_\tau^\star) \le -\tau \log p_{\text{ref}}$ by Lemma B.3, so that $M_\tau(q_\tau^\star)$ converges down to 0, and consequently $q_\tau^\star$ converges up to $q^\star$. □

# C   Proofs from Section 4

**Theorem 4.2.** *If $r \in M_b(\mathsf{X} \times \mathsf{A})$, $\gamma < 1$, and $\pi \in \mathsf{K}(\mathsf{X}, \mathscr{P}(\mathsf{A}))$ is such that*

$$\sup_{x,a} \|\tau \mathtt{kl}[\pi]\|_{L^p(P_{x,a})} < \infty, \tag{4.1}$$

*the soft distributional Bellman operator $\mathcal{T}_\tau^\pi$ is a $\gamma$-contraction in $\bar{d}_p$ for every $\tau \ge 0$. Thus, it has a unique solution to the fixed point equation $\bar\zeta = \mathcal{T}_\tau^\pi \bar\zeta$, which we denote by $\bar\zeta^{\pi,\tau}$.* [Source]

*Proof.* To begin, let us show that $\mathcal{T}_\tau^\pi$ maps elements of $\overline{\mathsf{K}}^p(\mathsf{X} \times \mathsf{A}, \mathscr{P}(\mathbb{R}))$ to $\overline{\mathsf{K}}^p(\mathsf{X} \times \mathsf{A}, \mathscr{P}(\mathbb{R}))$. For any $\zeta \in \overline{\mathsf{K}}^p(\mathsf{X} \times \mathsf{A}, \mathscr{P}(\mathbb{R}))$, observe that

$$
\begin{aligned}
\sup_{x,a} &\left( \int |z|^p \, \mathrm{d}(\mathcal{T}_\tau^\pi \zeta)_{x,a}(z) \right)^{1/p} \\
&= \sup_{x,a} \left( \int \left[ \int |r(x,a) - \gamma\tau \mathrm{KL}(\pi_{x'} \| \pi_{x'}^{\text{ref}}) + \gamma z|^p \, \mathrm{d}\zeta_{x',a'}(z) \right] \mathrm{d}\check{P}_{x,a}^\pi(x',a') \right)^{1/p} \\
&\le \|r\|_{\sup} + \gamma \sup_{x,a} \|\tau \mathtt{kl}[\pi]\|_{L^p(P_{x,a})} + \gamma \left( \sup_{x',a'} \int |z|^p \, \mathrm{d}\zeta_{x',a'}(z) \right)^{1/p} \\
&< \infty,
\end{aligned}
$$

by assumption, as desired.

Next, by the convexity of the Wasserstein metric [6, 40], we have

$$
\begin{aligned}
d_p&((\mathcal{T}_\tau^\pi \bar\zeta)_{x,a}, (\mathcal{T}_\tau^\pi \bar\zeta')_{x,a}) \\
&\le \int d_p \left( (\mathtt{b}_{r(x,a)-\gamma\mathrm{KL}(\pi_{x'} \| \pi_{x'}^{\text{ref}}),\gamma}) \# \bar\zeta_{x',a'}, (\mathtt{b}_{r(x,a)-\gamma\mathrm{KL}(\pi_{x'} \| \pi_{x'}^{\text{ref}}),\gamma}) \# \bar\zeta'_{x',a'} \right) \mathrm{d}\check{P}_{x,a}^\pi(x',a') \\
&\le \gamma \int d_p \left( \bar\zeta_{x',a'}, \bar\zeta'_{x',a'} \right) \mathrm{d}\check{P}_{x,a}^\pi(x',a') \\
&\le \gamma \sup_{x',a'} d_p(\bar\zeta_{x',a'}, \bar\zeta'_{x',a'}) \\
&= \gamma \bar{d}_p(\bar\zeta, \bar\zeta'),
\end{aligned}
$$

where the second inequality holds since the common transformation $\mathsf{b}_{r(x,a)-\gamma\tau\mathrm{KL}(\pi_{x'}\,\|\,\pi_{x'}^{\mathsf{ref}}),\gamma}$ is affine. As a consequence, we have that

$$\bar{d}_p(\mathcal{T}_\tau^\pi\bar{\zeta},\mathcal{T}_\tau^\pi\bar{\zeta}') \le \gamma\bar{d}_p(\bar{\zeta},\bar{\zeta}'),$$

which validates that $\mathcal{T}_\tau^\pi$ is a $\gamma$-contraction in $\bar{d}_p$. Consequently, since $(\overline{\mathsf{K}}^p(\mathsf{X}\times\mathsf{A},\mathscr{P}(\mathbb{R})),\bar{d}_p)$ is complete and separable [6], it follows that $\mathcal{T}_\tau^\pi$ has a unique fixed point. That $\bar{\zeta}^{\pi,\tau}$ coincides with this fixed point follows precisely by [6, Proposition 4.9]. $\qquad\square$

**Lemma 4.4.** *For any $\tau > 0$, $\mathcal{Q}\mathcal{T}_\tau^\star = \mathcal{B}_\tau^\star\mathcal{Q}$.* [Source]

*Proof.* For any $\bar{\zeta} \in \overline{\mathsf{K}}^p(\mathsf{X}\times\mathsf{A},\mathscr{P}(\mathbb{R}))$, we have

$$(\mathcal{Q}\mathcal{T}_\tau^\star\bar{\zeta})(x,a) = \iint (r(x,a) - \gamma\tau\mathrm{KL}((\mathcal{G}_\tau\mathcal{Q}\bar{\zeta})_{x'}\,\|\,\pi_{x'}^{\mathsf{ref}}) + \gamma z)\,\mathrm{d}\bar{\zeta}_{x',a'}(z)\,\mathrm{d}\check{P}_{x,a}^{\mathcal{G}_\tau\mathcal{Q}\bar{\zeta}}(x',a')$$

$$= \int \left(r(x,a) - \gamma\tau\mathrm{KL}((\mathcal{G}_\tau\mathcal{Q}\bar{\zeta})_{x'}\,\|\,\pi_{x'}^{\mathsf{ref}}) + \gamma\int z\,\mathrm{d}\bar{\zeta}_{x',a'}(z)\right)\,\mathrm{d}\check{P}_{x,a}^{\mathcal{G}_\tau\mathcal{Q}\bar{\zeta}}(x',a')$$

$$= \int \left(r(x,a) - \gamma\tau\mathrm{KL}((\mathcal{G}_\tau\mathcal{Q}\bar{\zeta})_{x'}\,\|\,\pi_{x'}^{\mathsf{ref}}) + \gamma(\mathcal{Q}\bar{\zeta})(x',a')\right)\,\mathrm{d}\check{P}_{x,a}^{\mathcal{G}_\tau\mathcal{Q}\bar{\zeta}}(x',a')$$

Defining $q := \mathcal{Q}\bar{\zeta}$, this is equivalent to

$$(\mathcal{Q}\mathcal{T}_\tau^\star\bar{\zeta})(x,a) = \int \left(r(x,a) - \gamma\tau\mathrm{KL}((\mathcal{G}_\tau q)_{x'}\,\|\,\pi_{x'}^{\mathsf{ref}}) + \gamma q(x',a')\right)\,\mathrm{d}\check{P}_{x,a}^{\mathcal{G}_\tau q}(x',a')$$

Moreover, note that

$$\mathrm{KL}((\mathcal{G}_\tau q)_x\,\|\,\pi_x^{\mathsf{ref}}) = \int \log\frac{\mathrm{d}(\mathcal{G}_\tau q)_x}{\mathrm{d}\pi_x^{\mathsf{ref}}}\,\mathrm{d}(\mathcal{G}_\tau q)_x$$

$$= \tau^{-1}\int (q(x,a) - \mathcal{V}_\tau q(x))\,\mathrm{d}(\mathcal{G}_\tau q)_x(a)$$

$$= \tau^{-1}\left(\int q(x,a)\,\mathrm{d}(\mathcal{G}_\tau q)_x(a) - \mathcal{V}_\tau q(x)\right).$$

Substituting, we have shown that

$$(\mathcal{Q}\mathcal{T}_\tau^\star\bar{\zeta})(x,a) = \int \left(r(x,a) - \gamma\int q(x',a''),\mathrm{d}\check{P}_{x,a}^{\mathcal{G}_\tau q} + \gamma\mathcal{V}_q^\tau(x') + \gamma q(x,a)\right)\,\mathrm{d}\check{P}_{x,a}^{\mathcal{G}_\tau q}(x',a')$$

$$= \int \left(r(x,a) + \gamma\mathcal{V}_q^\tau(x')\right)\,\mathrm{d}\check{P}_{x,a}^{\mathcal{G}_\tau q}(x',a')$$

$$\equiv \mathcal{B}_\tau^\star q(x,a)$$

$$= \mathcal{B}_\tau^\star\mathcal{Q}\bar{\zeta}(x,a).$$

$\qquad\square$

**Theorem 4.5.** *For any $\bar{\zeta} \in \overline{\mathsf{K}}^p(\mathsf{X}\times\mathsf{A},\mathscr{P}(\mathbb{R}))$ and temperature $\tau > 0$ define the iterates $(\bar{\zeta}^n)_{n\in\mathbb{N}}$ given by $\bar{\zeta}^{n+1} = \mathcal{T}_\tau^\star\bar{\zeta}^n$ for $\bar{\zeta}^0 = \mathcal{T}_\tau^\star\bar{\zeta}$. Then, for $\bar{\zeta}^{\tau,\star} := \bar{\zeta}^{\tau,\pi^{\tau,\star}}$,*

$$\bar{d}_p(\bar{\zeta}^n,\bar{\zeta}^{\tau,\star}) \le C_{p,\tau,\gamma}n\gamma^{n/p}\bar{d}_p(\bar{\zeta}^0,\bar{\zeta}^{\tau,\star}) \quad\text{and}\quad \bar{d}_1(\bar{\zeta}^n,\bar{\zeta}^{\tau,\star}) \le \frac{1}{(1-\gamma)\sqrt{\tau}}Cn\gamma^n\,\bar{d}_1(\bar{\zeta}^0,\bar{\zeta}^{\tau,\star}),$$

*where $C, C_{p,\tau,\gamma} < \infty$ are constants depending on $\|r\|_{\sup}$, $(p,\tau,\gamma,\|r\|_{\sup})$ respectively.* [Source]

*Proof.* We begin by defining some helper notation. For any $\bar{\zeta} \in \overline{\mathsf{K}}^p(\mathsf{X} \times \mathsf{A}, \mathscr{P}(\mathbb{R}))$, we define $\xi^{\bar{\zeta}} \in \overline{\mathsf{K}}^p(\mathsf{X}, \mathscr{P}(\mathbb{R} \times \mathsf{A}))$ where

$$\xi_x^{\bar{\zeta}} := (\mathtt{b}_{-\tau \mathrm{KL}((\mathcal{G}_\tau \mathcal{Q}\bar{\zeta})_x \,\|\, \pi_x^{\mathsf{ref}}),1} \circ \mathtt{proj}^{\mathbb{R}})_{\#}(\bar{\zeta}_{x,-} \otimes (\mathcal{G}_\tau \mathcal{Q}\bar{\zeta})_x). \tag{C.1}$$

In turn,

$$(\mathcal{T}_\tau^\star \bar{\zeta})_{x,a} = (\mathtt{b}_{r(x,a),\gamma} \circ \mathtt{proj}^{\mathbb{R}})_{\#}(\xi_-^{\bar{\zeta}} \otimes P_{x,a}). \tag{C.2}$$

Next, we define the following helpers,

$$\pi^n := \mathcal{G}_\tau \mathcal{Q}\bar{\zeta}^n, \quad \xi^n := \xi^{\bar{\zeta}^n}, \quad \xi^\star := \xi^{\bar{\zeta}^{\tau,\star}}.$$

By [40, Theorem 4.8], we have that for any $(x,a) \in \mathsf{X} \times \mathsf{A}$,

$$d_p(\bar{\zeta}_{x,a}^{n+1}, \bar{\zeta}^{\tau,\star}) \leq \gamma \int d_p(\xi_{x'}^n, \xi_{x'}^\star) \, \mathrm{d}P_{x,a}(x'). \tag{C.3}$$

Invoking the triangle inequality together with the expansion of the $\xi$ terms by definition, we have that for any $x \in \mathsf{X}$,

$$
\begin{aligned}
&d_p(\xi_x^n, \xi^\star) \\
&= d_p\left((\mathtt{proj}^{\mathbb{R}} - \tau \mathrm{KL}(\pi_x^n \,\|\, \pi_x^{\mathsf{ref}}))_{\#}(\bar{\zeta}_{x,-}^n \otimes \pi_x^n), (\mathtt{proj}^{\mathbb{R}} - \tau \mathrm{KL}(\pi_x^{\tau,\star} \,\|\, \pi_x^{\mathsf{ref}}))_{\#}(\bar{\zeta}_{x,-}^{\tau,\star} \otimes \pi_x^{\tau,\star})\right) \\
&\leq \overbrace{d_p\left((\mathtt{proj}^{\mathbb{R}} - \tau \mathrm{KL}(\pi_x^n \,\|\, \pi_x^{\mathsf{ref}}))_{\#}(\bar{\zeta}_{x,-}^n \otimes \pi_x^n), (\mathtt{proj}^{\mathbb{R}} - \tau \mathrm{KL}(\pi_x^n \,\|\, \pi_x^{\mathsf{ref}}))_{\#}(\bar{\zeta}_{x,-}^{\tau,\star} \otimes \pi_x^n)\right)}^{\mathrm{I}_n} \\
&\quad + \underbrace{d_p\left((\mathtt{proj}^{\mathbb{R}} - \tau \mathrm{KL}(\pi_x^n \,\|\, \pi_x^{\mathsf{ref}}))_{\#}(\bar{\zeta}_{x,-}^{\tau,\star} \otimes \pi_x^n)(\mathtt{proj}^{\mathbb{R}} - \tau \mathrm{KL}(\pi_x^{\tau,\star} \,\|\, \pi_x^{\mathsf{ref}}))_{\#}(\bar{\zeta}_{x,-}^{\tau,\star} \otimes \pi_x^{\tau,\star})\right)}_{\mathrm{II}_n}.
\end{aligned}
$$

Since the measures being compared in $\mathrm{I}_n$ are both translated by the same pushforward map, another application of [40, Theorem 4.8] yields the following inequality:

$$\mathrm{I}_n \leq \int d_p(\bar{\zeta}_{x,a}^n, \bar{\zeta}_{x,a}^{\tau,\star}) \, \mathrm{d}\pi_x^n(a) \leq \overline{d}_p(\bar{\zeta}^n, \bar{\zeta}^{\tau,\star}).$$

Next, we bound $\mathrm{II}_n$. Let $\mathscr{C}(\rho_1, \rho_2)$ be the set of couplings between measures $\rho_1, \rho_2$. Then

$$
\begin{aligned}
\mathrm{II}_n &\leq \inf_{\kappa \in \mathscr{C}(\bar{\zeta}_{x,-}^{\tau,\star} \otimes \pi_x^n, \bar{\zeta}_{x,-}^{\tau,\star} \otimes \pi_x^{\tau,\star})} \left(\int \left|\mathtt{b}_{-\tau\mathrm{KL}(\pi_x^n \,\|\, \pi_x^{\mathsf{ref}}),1}(z) - \mathtt{b}_{-\tau\mathrm{KL}(\pi_x^{\tau,\star} \,\|\, \pi_x^{\mathsf{ref}}),1}(z')\right|^p \mathrm{d}\kappa\right)^{1/p} \\
&\overset{(a)}{\leq} \inf_{\kappa \in \mathscr{C}(\bar{\zeta}_{x,-}^{\tau,\star} \otimes \pi_x^n, \bar{\zeta}_{x,-}^{\tau,\star} \otimes \pi_x^{\tau,\star})} \left(\int |z - z'|^p \, \mathrm{d}\kappa\right)^{1/p} + \tau \left|\mathrm{KL}(\pi_x^n \,\|\, \pi_x^{\mathsf{ref}}) - \mathrm{KL}(\pi_x^{\tau,\star} \,\|\, \pi_x^{\mathsf{ref}})\right| \\
&\overset{(b)}{\leq} C\gamma^{n/p} \|\mathcal{Q}\bar{\zeta}^0 - q_\tau^\star\|_{\sup}^{1/p},
\end{aligned}
$$

for some constant $C$ depending on $\tau, p, \gamma, \|r\|_{\sup}$ where $(a)$ applies Minkowski's inequality, noting that the KL terms are independent of $\kappa$, and $(b)$ invokes Lemma C.5 and Lemma C.6. Indeed, for $n$ large enough, Lemmas C.5 and C.6 assert that $C \lesssim \tau^{-1}$ for fixed $p$, and more generally that $C \lesssim \tau^{-1/2}$ for any $n$ (and fixed $p$). Substituting back into (C.3), we see that

$$\overline{d}_p(\bar{\zeta}^{n+1}, \bar{\zeta}^{\tau,\star}) \leq \gamma \overline{d}_p(\bar{\zeta}^n, \bar{\zeta}^{\tau,\star}) + C\gamma^{1+n/p} \|\mathcal{Q}\bar{\zeta}^0 - q_\tau^\star\|_{\sup}^{1/p}.$$

Let $a_n := \overline{d}_p(\bar{\zeta}^n, \bar{\zeta}^{\tau,\star})$. We have shown that $a_{n+1} \leq \gamma a_n + C'\gamma^{1+n/p}$, where $C' = C\|\mathcal{Q}\bar{\zeta}^0 - q_\tau^\star\|_{\sup}^{1/p}$ is a constant depending on $p$ and $\tau$. We will apply techniques of *generatingfunctionology* [41] to bound this sequence. We define $A : \mathbb{R} \to \mathbb{R}$ as the *formal power series* given by

$$A(y) = \sum_{n=0}^\infty a_n y^n,$$

and we will pick off the coefficients $a_n$ from the power series representation of $A$. Our recurrence above, upon multiplying through by $y^n$ and summing over $n$ yields

$$\sum_{n=0}^{\infty} a_{n+1} y^n \leq \gamma \sum_{n=0}^{\infty} a_n y^n + C' \gamma \sum_{n=0}^{\infty} \gamma^{n/p} y^n$$

$$\therefore \frac{1}{y} A(y) - a_0 \leq \gamma A(y) + C' \gamma \frac{1}{1 - \gamma^{1/p} x}$$

$$\therefore A(y) \leq \frac{a_0 y}{1 - \gamma y} + \frac{C' \gamma y}{(1 - \gamma^{1/p} y)(1 - \gamma y)},$$

where $a_0 = \overline{d}_p(\bar{\zeta}^0, \bar{\zeta}^{\tau, \star})$. Now, the formal power series expansion gives

$$A(y) \leq \begin{cases} \sum_{n=1}^{\infty} \left[ a_0 \gamma^n + C' \gamma \frac{\gamma^{n/p} - \gamma^n}{\gamma^{1/p} - \gamma} \right] y^n & p \neq 1 \\ \sum_{n=1}^{\infty} \left[ a_0 \gamma^n + C' n \gamma^n \right] y^n & p = 1. \end{cases}$$

Combining, we have

$$\overline{d}_p(\bar{\zeta}^n, \bar{\zeta}^{\tau, \star}) = a_n \leq (1 + \overline{d}_p(\bar{\zeta}^0, \bar{\zeta}^{\tau, \star})) C'' n \gamma^{n/p}$$

where $C'' = C'$ when $p = 1$, and $C'' = C'/(\gamma^{1/p} - \gamma)$ otherwise—in any case, $C''$ is a constant depending only on $p, \tau, \gamma$, and the proof is complete. $\qquad\square$

**Theorem 4.6.** *Suppose Assumption 3.4 holds. Let $p, p' \in [1, \infty)$ and $\omega \in \mathscr{P}(\mathsf{X} \times \mathsf{A})$. For any $\epsilon, \delta > 0$, there exists a $\tau > 0$ for which $d_{p;p',\omega}(\bar{\zeta}^{\tau, \pi^{\tau, \star}}, \zeta^{\pi^{\tau, \star}}) \leq \delta/2$ and $q^{\pi^{\tau, \star}}$ is $\epsilon/2$-reference-optimal. In turn, an $n_{\epsilon, \delta} = n_{\epsilon, \delta}(\tau) \in \mathbb{N}$ exists for which*

$$d_{p;p',\omega}(\bar{\zeta}^n, \zeta^{\pi^{\tau, \star}}) \leq \delta \quad \text{and} \quad \mathcal{G}_\tau \mathcal{Q} \bar{\zeta}^n \text{ is } \epsilon\text{-reference-optimal} \quad \forall n \geq n_{\epsilon, \delta}$$

*where $\bar{\zeta}^{n+1} = \mathcal{T}_\tau^\star \bar{\zeta}^n$ and $\bar{\zeta}^0 = \mathcal{T}_\tau^\star \bar{\zeta}$ for any $\bar{\zeta} \in \overline{\mathsf{K}}^p(\mathsf{X} \times \mathsf{A}, \mathscr{P}(\mathbb{R}))$.* [Source]

*Proof.* By Lemma B.10 and under Assumption 3.4,

$$\|q_\tau^\star - q_{\mathsf{ref}}^\star\|_{\sup} \leq \frac{\gamma \log p_{\mathsf{ref}}^{-1}}{1 - \gamma} \tau \leq \frac{\epsilon}{2},$$

choosing $\tau \leq \tau_\epsilon := \frac{\epsilon(1 - \gamma)}{2\gamma \log p_{\mathsf{ref}}^{-1}}$. Hence, by Lemmas 4.4 and A.7

$$\|\mathcal{Q} \bar{\zeta}^{n_\epsilon} - q_{\mathsf{ref}}^\star\|_{\sup} \leq \gamma^{n_\epsilon} \|\mathcal{Q} \bar{\zeta}^0 - q_\tau^\star\|_{\sup} + \|q_\tau^\star - q_{\mathsf{ref}}^\star\|_{\sup} \leq \gamma^{n_\epsilon} \|\mathcal{Q} \bar{\zeta}^0 - q_{\mathsf{ref}}^\star\|_{\sup} + \frac{\epsilon}{2} \leq \epsilon,$$

which holds when $n_\epsilon \geq (\log \gamma)^{-1} \log \frac{\epsilon}{2\|\mathcal{Q} \bar{\zeta}^0 - q_{\mathsf{ref}}^\star\|_{\sup}}$.

Next, we will show that the soft return distribution estimates will approximate $\zeta^{\pi^{\tau, \star}}$. For notational simplicity, define $X_t := X_t^{\pi^{\tau, \star}}$ and $A_t := A_t^{\pi^{\tau, \star}}$ for $t \in \mathbb{N}$. Recall that

$$\bar{\zeta}_{x,a}^{\tau, \pi^{\tau, \star}} = \mathrm{law}\left( r(x, a) + \sum_{t \geq 1} \gamma^t \left( r(X_t, A_t) - \tau \mathrm{KL}(\pi_{X_t}^{\tau, \star} \| \pi_{X_t}^{\mathsf{ref}}) \right) \,\Big|\, X_0 = x, A_0 = a \right).$$

Moreover, we define $\widetilde{\zeta}_{x,a}^{\tau, \pi^{\star}, \tau} := (-\tau \mathrm{KL}(\pi_x^{\tau, \star} \| \pi_x^{\mathsf{ref}}) + \mathtt{id})_\# \bar{\zeta}_{x,a}^{\tau, \pi^{\tau, \star}}$, so that

$$\widetilde{\zeta}^{\tau, \pi^{\star}, \tau} = \mathrm{law}\left( \sum_{t \geq 0} \gamma^t \left( r(X_t, A_t) - \tau \mathrm{KL}(\pi_{X_t}^{\tau, \star} \| \pi_{X_t}^{\mathsf{ref}}) \right) \,\Big|\, X_0 = x, A_0 = a \right).$$

Now, by the triangle inequality, we have

$$d_{p;p',\omega}^{p'}(\bar{\zeta}^{\tau, \pi^{\tau, \star}}, \zeta^{\pi^{\tau, \star}}) \leq 2^{p'-1} \int \left[ \underbrace{d_p^{p'}(\bar{\zeta}_{x,a}^{\tau, \pi^{\tau, \star}}, \widetilde{\zeta}_{x,a}^{\tau, \pi^{\tau, \star}})}_{\mathrm{I}_\tau(x,a)} + \underbrace{d_p^{p'}(\widetilde{\zeta}_{x,a}^{\tau, \pi^{\tau, \star}}, \zeta_{x,a}^{\pi^{\tau, \star}})}_{\mathrm{II}_\tau(x,a)} \right] \mathrm{d}\omega(x, a). \quad \text{(C.4)}$$

We proceed by analysing $I_\tau$. By coupling states and actions, we immediately have

$$I_\tau(x,a) \leq \left(\tau \mathrm{KL}(\pi_x^{\tau,\star} \parallel \pi_x^{\mathsf{ref}})\right)^{p'},$$

and so, since $\pi^{\tau,\star} = \mathcal{G}_\tau q_\tau^\star$, by virtue of Lemma B.2 we have

$$\limsup_{\tau \to 0} I_\tau(x,a) = 0.$$

Next, we bound $II_\tau$. Denote by $r_{\pi,\tau} : \mathsf{X} \times \mathsf{A} \to \mathbb{R}$ the reward function defined by

$$r_{\pi,\tau}(x,a) = r(x,a) - \tau \mathrm{KL}(\pi_x^{\tau,\star} \parallel \pi_x^{\mathsf{ref}}).$$

The work of [44] shows that, for any policy $\pi$, there is a unique $\daleth^\pi \in \mathsf{K}(\mathsf{X} \times \mathsf{A}, \mathscr{P}(\mathscr{P}(\mathsf{X} \times \mathsf{A})))$ for which $(\mu \mapsto (1-\gamma)^{-1}(\mu r)(x,a))_\# \daleth_{x,a}^\pi = \zeta_{x,a}^{\pi,r}$, where $\zeta^{\pi,r}$ denotes the return distribution function associated to the policy $\pi$ for the reward function $r$. Noting that $\widetilde{\zeta}^{\tau,\pi^{\tau,\star}} = \zeta^{\pi^{\tau,\star}, r_{\pi,\tau}}$, we have

$$II_\tau(x,a) = d_p^{p'}\left(\left(\mu \mapsto \frac{1}{1-\gamma}(\mu r_{\pi,\tau})(x,a)\right)_\# \daleth_{x,a}^{\pi^{\tau,\star}}, \left(\mu \mapsto \frac{1}{1-\gamma}(\mu r)(x,a)\right)_\# \daleth_{x,a}^{\pi^{\tau,\star}}\right)$$

$$\leq \frac{1}{1-\gamma}\left(\int \left[\int |r_{\pi,\tau}(x',a') - r(x',a')|^p \, d\mu(x',a')\right] d\daleth_{x,a}^{\pi^{\tau,\star}}(\mu)\right)^{p'/p}$$

$$= \frac{1}{1-\gamma}\left(\int \left[\int \tau \mathrm{KL}(\pi_{x'}^{\tau,\star} \parallel \pi_{x'}^{\mathsf{ref}})^p \, d\mu(x',a')\right] d\daleth_{x,a}^{\pi^{\tau,\star}}(\mu)\right)^{p'/p}$$

where the penultimate step is simply a coupling argument (coupling the samples of $\daleth^{\pi^{\tau,\star}}$). Once again, since $\limsup_{\tau \to 0} \tau \mathrm{KL}(\pi_x^{\tau,\star} \parallel \pi_x^{\mathsf{ref}}) = 0$, and $\mathrm{KL}(\pi_x^{\tau,\star} \parallel \pi_x^{\mathsf{ref}})$ is bounded by Lemma B.2, the dominated convergence theorem asserts that $\lim_{\tau \to 0} II_\tau(x,a) = 0$ pointwise.

Altogether, we have shown that $\lim_{\tau \to 0}(I_\tau(x,a) + II_\tau(x,a)) = 0$ pointwise, and is bounded as a consequence of Lemma B.2. Thus, by another application of the dominated convergence theorem together with (C.4), we have that

$$\lim_{\tau \to 0} d_{p;p',\omega}(\bar{\zeta}^{\tau,\pi^{\tau,\star}}, \zeta^{\pi^{\tau,\star}}) = 0.$$

It follows that there exists some $\tau_\delta > 0$ for which $d_{p;p',\omega}(\bar{\zeta}^{\tau,\pi^{\tau,\star}}, \zeta^{\pi^{\tau,\star}}) \leq \delta/2$ whenever $\tau \leq \tau_\delta$. For any such $\tau$, by Theorem 4.5, there exists $n_\delta \in \mathbb{N}$ for which

$$d_{p;p',\omega}(\bar{\zeta}^{n_\delta}, \bar{\zeta}^{\tau,\pi^{\tau,\star}}) \leq \overline{d}_p(\bar{\zeta}^{n_\delta}, \bar{\zeta}^{\tau,\pi^{\tau,\star}}) \leq \frac{\delta}{2}.$$

For this choice of $\tau$ and $n_\delta$, by the triangle inequality,

$$d_{p;p',\omega}(\bar{\zeta}^{n_\delta}, \zeta^\pi) \leq \delta.$$

Altogether, taking $\tau = \min\{\tau_\epsilon, \tau_\delta\}$ and $n = \max\{n_\epsilon, n_\delta\}$, we have that

$$d_{p;p',\omega}(\bar{\zeta}^{\tau,\pi^{\tau,\star}}, \zeta^\pi) \leq \frac{\delta}{2} \quad \text{and} \quad \|q^{\pi^{\tau,\star}} - q_{\mathsf{ref}}^\star\|_{\sup} \leq \frac{\epsilon}{2},$$

as well as

$$d_{p;p',\omega}(\bar{\zeta}^n, \zeta^\pi) \leq \delta \quad \text{and} \quad \|\mathcal{Q}\bar{\zeta}^n - q_{\mathsf{ref}}^\star\|_{\sup} \leq \epsilon.$$

To complete the proof, we note that

$$q^{\mathcal{G}_\tau \mathcal{Q}\bar{\zeta}^n} \geq q^{\mathcal{G}_\tau \mathcal{Q}\bar{\zeta}^n} \geq q_{\mathsf{ref}}^\star - \epsilon,$$

so that $\mathcal{G}_\tau \mathcal{Q}\bar{\zeta}^n$ is $\epsilon$-reference-optimal. $\qquad \square$

**Theorem 4.7.** *Suppose Assumption 3.4 holds and $\mathsf{A}$ is discrete. Let $p, p' \in [1, \infty)$ and $\omega \in \mathscr{P}(\mathsf{X} \times \mathsf{A})$. For any $\epsilon, \delta > 0$ and $\bar{\zeta}^0 \in \overline{\mathsf{K}}^p(\mathsf{X} \times \mathsf{A}, \mathscr{P}(\mathbb{R}))$, there exists $\tau > 0$, a decoupled $\sigma_\tau > 0$ and $n_{\mathsf{opt}}, n_{\mathsf{eval}} \in \mathbb{N}$ such that*

$$d_{p;p',\omega}(\hat{\zeta}^{n_{\mathsf{eval}}}, \zeta^{\pi^{\mathsf{ref},\star}}) \leq \delta \quad \text{and} \quad \mathcal{G}_\tau \mathcal{Q}\hat{\zeta}^{n_{\mathsf{eval}}} \text{ is } \epsilon\text{-reference-optimal}$$

*where $\bar{\zeta}^{n+1} = \mathcal{T}_\sigma^\star \bar{\zeta}^n$, $\hat{\pi}^{\tau,\sigma} = \mathcal{G}_\tau \bar{\zeta}^{n_{\mathsf{opt}}}$, and $\hat{\zeta}^{n+1} = \mathcal{T}_\tau^{\hat{\pi}^{\tau,\sigma}} \hat{\zeta}^n$, for $\hat{\zeta}^0 = \bar{\zeta}^{n_{\mathsf{opt}}}$.* [Source]

*Proof.* Appealing to Theorem 3.10, for any $\delta > 0$, any temperature decoupling gambit yields a $\tau_\delta > 0$ and an associated decoupled temperature $\sigma_\delta = \sigma(\tau_\delta) > 0$ such that

$$d_{p;p',\omega}(\zeta^{\tau,\sigma}, \zeta^\star) \leq \delta/3$$

whenever $\tau \leq \tau_\delta$. Moreover, as shown in the proof of Theorem 4.6, for small enough $\tau'_\delta$,

$$d_{p;p',\omega}(\zeta^{\tau,\sigma}, \bar{\zeta}^{\tau,\sigma}) \leq \delta/3$$

whenever $\tau \leq \tau'_\delta$—here, we recall that $\bar{\zeta}^{\tau,\sigma}$ is the *entropy-regularized* return distribution function for the decoupled policy $\pi^{\tau,\sigma}$.

Now, define $\hat{\zeta}^{\tau,\sigma} = (\mathcal{T}_\tau^{\hat{\pi}^{\tau,\sigma}})^{n_{\text{eval}}}\hat{\zeta}^{\sigma,\star}$, and $\hat{\zeta}^{\sigma,\star} = (\mathcal{T}_\tau^\star \sigma)^{n_{\text{opt}}}\bar{\zeta}^0$. By the triangle inequality, we have

$$
\begin{aligned}
d_{p;p',\omega}(\hat{\zeta}^{\tau,\sigma}, \bar{\zeta}^{\tau,\sigma}) &\leq d_{p;p',\omega}((\mathcal{T}_\tau^{\hat{\pi}^{\tau,\sigma}})^{n_{\text{eval}}}\hat{\zeta}^{\sigma,\star}, (\mathcal{T}_\tau^{\pi^{\tau,\sigma}})^{n_{\text{eval}}}\hat{\zeta}^{\sigma,\star}) \\
&\quad + d_{p;p',\omega}((\mathcal{T}_\tau^{\pi^{\tau,\sigma}})^{n_{\text{eval}}}\hat{\zeta}^{\sigma,\star}, \bar{\zeta}^{\tau,\sigma}) \\
&\overset{(a)}{\leq} d_{p;p',\omega}((\mathcal{T}_\tau^{\hat{\pi}^{\tau,\sigma}})^{n_{\text{eval}}}\hat{\zeta}^{\sigma,\star}, (\mathcal{T}_\tau^{\pi^{\tau,\sigma}})^{n_{\text{eval}}}\hat{\zeta}^{\sigma,\star}) \\
&\quad + d_{p;p',\omega}((\mathcal{T}_\tau^{\pi^{\tau,\sigma}})^{n_{\text{eval}}}\hat{\zeta}^{\sigma,\star}, (\mathcal{T}_\tau^{\pi^{\tau,\sigma}})^{n_{\text{eval}}}\bar{\zeta}^{\tau,\sigma}) \\
&\overset{(b)}{\leq} d_{p;p',\omega}((\mathcal{T}_\tau^{\hat{\pi}^{\tau,\sigma}})^{n_{\text{eval}}}\hat{\zeta}^{\sigma,\star}, (\mathcal{T}_\tau^{\pi^{\tau,\sigma}})^{n_{\text{eval}}}\hat{\zeta}^{\sigma,\star}) \\
&\quad + \gamma^{n_{\text{eval}}}\bar{d}_p(\hat{\zeta}^{\tau,\sigma}, \bar{\zeta}^{\tau,\sigma}) \\
&\overset{(c)}{\lesssim} \gamma^{n_{\text{opt}}/2p} + \gamma^{n_{\text{eval}}}.
\end{aligned}
$$

Here, $(a)$ leverages the fact that $\bar{\zeta}^{\tau,\sigma}$ is the fixed point of $\mathcal{T}_\tau^{\pi^{\tau,\sigma}}$ by definition, $(b)$ invokes the contractivity of $\mathcal{T}_\tau^{\pi^{\tau,\sigma}}$ shown in Theorem 4.2 appealing to the fact that $\pi^{\tau,\sigma}$ is a BG policy for reference $\pi^{\text{ref}}$, and $(c)$ follows by Lemma C.1. As a consequence, again since $|\gamma| < 1$, for sufficiently large $n_{\text{opt}}, n_{\text{eval}} \in \mathbb{N}$, we have

$$d_{p;p',\omega}(\hat{\zeta}^{\tau,\sigma}, \bar{\zeta}^{\tau,\sigma}) \leq \delta/3.$$

Altogether, by the triangle inequality once again, for the choices of $n_{\text{opt}}, n_{\text{eval}}, \tau, \sigma$ above,

$$
\begin{aligned}
d_{p;p',\omega}(\hat{\zeta}^{\tau,\sigma}, \zeta^\star) &\leq d_{p;p',\omega}(\hat{\zeta}^{\tau,\sigma}, \bar{\zeta}^{\tau,\sigma}) + d_{p;p',\omega}(\bar{\zeta}^{\tau,\sigma}, \zeta^{\tau,\sigma}) + d_{p;p',\omega}(\zeta^{\tau,\sigma}, \zeta^\star) \\
&\leq \frac{\delta}{3} + \frac{\delta}{3} + \frac{\delta}{3} \\
&= \delta.
\end{aligned}
$$

This completes the proof of the first claim. It remains to show now that $\mathcal{G}_\tau \mathcal{Q}\hat{\zeta}^{n_{\text{eval}}}$ is $\epsilon$-reference-optimal. Towards this end, we note that by Theorem 4.6 and 4.7 that there exists $\tau_\epsilon > 0$, $n_\epsilon \in \mathbb{N}$ such that

$$\bar{d}_1(\hat{\zeta}^{n_{\text{eval}}}, \bar{\zeta}^{\tau,\sigma}) \leq \epsilon/3 \tag{C.5}$$

whenever $\max\{n_{\text{eval}}, n_{\text{opt}}\} \geq n_\epsilon$ and $\tau \leq \tau_\epsilon$. To proceed, we note that for any $(x, a) \in \mathsf{X} \times \mathsf{A}$,

$$
\begin{aligned}
|q_\tau^{\pi^{\tau,\sigma}}(x, a) - q_{\text{ref}}^\star(x, a)| &\leq |q_\tau^{\pi^{\tau,\sigma}}(x, a) - q_\tau^\star(x, a)| + |q_\tau^\star(x, a) - q_{\text{ref}}^\star(x, a)| \\
&\leq |q_\tau^{\pi^{\tau,\sigma}}(x, a) - q_\tau^\star(x, a)| + \epsilon/3,
\end{aligned}
$$

where the last inequality holds for small enough $\tau$ by Theorem 3.2. Continuing, we have

$$
\begin{aligned}
|q_\tau^{\pi^{\tau,\sigma}}(x, a) - q_\tau^\star(x, a)| &= |q_\tau^{\pi^{\tau,\sigma}}(x, a) - q_\tau^{\pi^{\tau,\star}}(x, a)| \\
&= \gamma \left| \int_{\mathsf{X}} (\mathcal{V}_\tau q_\sigma^\star - \mathcal{V}_\tau q_\tau^\star) \, \mathrm{d}P_{x,a} \right| \\
&\leq \gamma \|\mathcal{V}_\tau q_\sigma^\star - \mathcal{V}_\tau q_\tau^\star\|_{\text{sup}} \\
&\leq \gamma \|q_\sigma^\star - q_\tau^\star\|_{\text{sup}},
\end{aligned}
$$

where the final inequality holds since $\mathcal{V}_\tau$, as a log-sum-exp, is 1-Lipschitz. Now, again by Theorem 3.2, for small enough $\tau$ (inducing small enough $\sigma$), we have

$$\gamma\|q_\sigma^\star - q_\tau^\star\|_{\sup} \le \gamma\|q_\sigma^\star - q_{\mathsf{ref}}^\star\|_{\sup} + \gamma\|q_\tau^\star - q_{\mathsf{ref}}^\star\|_{\sup} \le \epsilon/6 + \epsilon/6 = \epsilon/3.$$

Altogether, we have that

$$\sup_{x,a}|q_\tau^{\pi^{\tau,\sigma}}(x,a) - q_{\mathsf{ref}}^\star(x,a)| \le \epsilon/3 + \epsilon/3 = 2\epsilon/3.$$

Next, since $d_1(\rho_1, \rho_2) \ge \mathbf{E}_{(Z_1,Z_2)\sim\rho_1\otimes\rho_2}[|Z_1 - Z_2|]$, we have that

$$\|\mathcal{Q}\hat\zeta^{n_{\mathsf{eval}}} - q_\tau^{\pi^{\tau,\sigma}}\|_{\sup} \le \overline{d}_1(\hat\zeta^{n_{\mathsf{eval}}}, \bar\zeta^{\tau,\sigma}) \le \epsilon/3,$$

by (C.5). Now, by yet another triangle inequality,

$$\|\mathcal{Q}\hat\zeta^{n_{\mathsf{eval}}} - q_{\mathsf{ref}}^\star\|_{\sup} \le \|\mathcal{Q}\hat\zeta^{n_{\mathsf{eval}}} - q_\tau^{\pi^{\tau,\sigma}}\| + \|q_\tau^{\pi^{\tau,\sigma}} - q_{\mathsf{ref}}^\star\|_{\sup}$$
$$\le \epsilon/2 + 2\epsilon/3 = \epsilon.$$

Consequently, we have

$$q^{\mathcal{G}_\tau\mathcal{Q}\hat\zeta^{n_{\mathsf{eval}}}} \ge q_\tau\mathcal{G}_\tau\mathcal{Q}\hat\zeta^{n_{\mathsf{eval}}} \ge q_{\mathsf{ref}}^\star - \epsilon.$$

Thus, we have shown that $\mathcal{G}_\tau\mathcal{Q}\hat\zeta^{n_{\mathsf{eval}}}$ is $\epsilon$-reference-optimal, completing the proof. $\qquad\square$

**Lemma C.1.** *Let $\zeta \in \overline{K}^p(\mathsf{X}\times\mathsf{A}, \mathscr{P}(\mathbb{R}))$, $\tau, \sigma > 0$, $n_{\mathsf{eval}}, n_{\mathsf{opt}} \in \mathbb{N}$ be given. Define $\hat\zeta^{\sigma,\star} := (\mathfrak{T}_\sigma^\star)^{n_{\mathsf{opt}}}\zeta$, and let $\hat\pi^{\sigma,\star} = \mathcal{G}_\tau\hat\zeta^{\sigma,\star}$. Then, we have*

$$\overline{d}_p((\mathfrak{T}_\tau^{\hat\pi^{\sigma,\star}})^{n_{\mathsf{eval}}}\hat\zeta^{\sigma,\star}, (\mathfrak{T}_\tau^{\pi^{\tau,\sigma}})^{n_{\mathsf{eval}}}\hat\zeta^{\sigma,\star}) \lesssim \gamma^{n_{\mathsf{eval}}} + \gamma^{n_{\mathsf{opt}}/2p}.$$

*Proof.* By Lemma C.2, we have

$$\overline{d}_p((\mathfrak{T}_\tau^{\hat\pi^{\sigma,\star}})^{n_{\mathsf{eval}}}\hat\zeta^{\sigma,\star}, (\mathfrak{T}_\tau^{\pi^{\tau,\sigma}})^{n_{\mathsf{eval}}}\hat\zeta^{\sigma,\star})$$
$$\lesssim (\tau^{-1}\|\mathcal{Q}\hat\zeta^{\sigma,\star} - q_\sigma^\star\|_{\sup})^{1/2p} + \sqrt{\tau^{-1}\|\mathcal{Q}\hat\zeta^{\sigma,\star} - q_\sigma^\star\|_{\sup}} + \|\mathcal{Q}\hat\zeta^{\tau,\sigma} - q_\sigma^\star\|_{\sup}.$$

It remains to bound $\|\mathcal{Q}\hat\zeta^{\sigma,\star} - q_\sigma^\star\|_{\sup}$. However, by Lemma 4.4 and the contractivity of $\mathcal{B}_\sigma^\star$, we have that

$$\|\mathcal{Q}\hat\zeta^{\sigma,\star} - q_\sigma^\star\|_{\sup} \lesssim \gamma^{n_{\mathsf{opt}}}.$$

Since $|\gamma| < 1$, it follows that

$$\overline{d}_p((\mathfrak{T}_\tau^{\hat\pi^{\sigma,\star}})^{n_{\mathsf{eval}}}\hat\zeta^{\sigma,\star}, (\mathfrak{T}_\tau^{\pi^{\tau,\sigma}})^{n_{\mathsf{eval}}}\hat\zeta^{\sigma,\star}) \lesssim \gamma^{n_{\mathsf{opt}}/2p}.$$

$\qquad\square$

**Lemma C.2.** *Let $\zeta \in \overline{K}^p(\mathsf{X}\times\mathsf{A}, \mathscr{P}(\mathbb{R}))$, $\tau, \sigma > 0$, and $n_{\mathsf{eval}} \in \mathbb{N}$ be given. Then*

$$\overline{d}_p((\mathfrak{T}_\tau^{\hat\pi})^{n_{\mathsf{eval}}}\zeta, (\mathfrak{T}_\tau^\pi)^{n_{\mathsf{eval}}}\zeta) \lesssim (\tau^{-1}\|\mathcal{Q}\zeta - q_\sigma^\star\|_{\sup})^{1/2p} + \sqrt{\tau^{-1}\|\mathcal{Q}\zeta - q_\sigma^\star\|_{\sup}} + \|\mathcal{Q}\zeta - q_\sigma^\star\|_{\sup}.$$

*Proof.* For simplicity, we define $\hat\pi = \mathcal{G}_\tau\mathcal{Q}\zeta$ and $\pi = \mathcal{G}_\tau q_\sigma^\star$. We want to bound

$$\overline{d}_p((\mathfrak{T}_\tau^{\hat\pi})^{n_{\mathsf{eval}}}\zeta, (\mathfrak{T}_\tau^\pi)^{n_{\mathsf{eval}}}\zeta).$$

By Lemma C.3, we have

$$\overline{d}_p((\mathfrak{T}_\tau^{\hat\pi})^{n_{\mathsf{eval}}}\zeta, (\mathfrak{T}_\tau^\pi)^{n_{\mathsf{eval}}}\zeta)$$
$$\le \gamma\overline{d}_p((\mathfrak{T}_\tau^{\hat\pi})^{n_{\mathsf{eval}}-1}\zeta, (\mathfrak{T}_\tau^\pi)^{n_{\mathsf{eval}}-1}\zeta) + 2\gamma\sup_{y,b}\left[\|\mathtt{id}\|_{L^p(((\mathfrak{T}_\tau^\pi)^{n_{\mathsf{eval}}-1}\zeta)_{y,b})}c_1 + c_2\right]$$
$$\le \gamma^2\overline{d}_p((\mathfrak{T}_\tau^{\hat\pi})^{n_{\mathsf{eval}}-2}\zeta, (\mathfrak{T}_\tau^\pi)^{n_{\mathsf{eval}}-2}\zeta) + 2\gamma^2\sup_{y,b}\left[\|\mathtt{id}\|_{L^p(((\mathfrak{T}_\tau^\pi)^{n_{\mathsf{eval}}-2}\zeta)_{y,b})}c_1 + c_2\right]$$
$$\quad + 2\gamma\sup_{y,b}\left[\|\mathtt{id}\|_{L^p(((\mathfrak{T}_\tau^\pi)^{n_{\mathsf{eval}}-1}\zeta)_{y,b})}c_1 + c_2\right]$$
$$\le \gamma^{n_{\mathsf{eval}}}\overline{d}_p(\zeta, \zeta) + 2\sum_{k=1}^{n_{\mathsf{eval}}}\gamma^k\sup_{y,b}\left[\|\mathtt{id}\|_{L^p(((\mathfrak{T}_\tau^\pi)^{n_{\mathsf{eval}}-k}\zeta)_{y,b})}c_1 + c_2\right]$$

where

$$c_1 := \sup_x \|\hat{\pi}_x - \pi_x\|_{\mathrm{TV}}^{1/p} \quad \text{and} \quad c_2 := c_1^p \|q_\sigma^\star\|_{\sup} + 2\|\mathcal{Q}\zeta - q_\sigma^\star\|_{\sup}.$$

Now $((\mathfrak{T}_\tau^{\mathcal{G}_\tau q_\sigma^\star})^n \hat{\zeta}^{\sigma,\star})_{n\in\mathbb{N}}$ is the sequence of return distributions generated by iterative applications of a contractive operator on $\overline{\mathsf{K}}^p(\mathsf{X} \times \mathsf{A}, \mathscr{P}(\mathbb{R}))$. Thus,

$$\sup_n \sup_{y,b} \|\mathtt{id}\|_{L^p(((\mathfrak{T}_\tau^{\mathcal{G}_\tau q_\sigma^\star})^n \hat{\zeta}^{\sigma,\star})_{y,b})} \le c_3 < \infty,$$

where $c_3$ is a constant depending only on $p, \gamma, \sigma, \tau, \|r\|_{\sup}$. It remains to bound $c_1$ and $c_2$. By Theorem 3.6, we have

$$c_1 = \sup_x \|\mathcal{G}_\tau \mathcal{Q} \hat{\zeta}_x^{\sigma,\star} - \mathcal{G}_\tau q_\sigma^\star\|_{\mathrm{TV}}^{1/p}$$
$$\le (\tau^{-1}\|\mathcal{Q}\zeta - q_\sigma^\star\|_{\sup})^{1/2p}.$$

Thus, since $\|q_\sigma^\star\|_{\sup}$ is uniformly bounded for any $\sigma > 0$, we have shown that

$$\overline{d}_p((\mathfrak{T}_\tau^{\hat{\pi}})^{n_{\mathrm{eval}}}\zeta, (\mathfrak{T}_\tau^{\pi})^{n_{\mathrm{eval}}}\zeta) \lesssim (\tau^{-1}\|\mathcal{Q}\zeta - q_\sigma^\star\|_{\sup})^{1/2p} + \sqrt{\tau^{-1}\|\mathcal{Q}\zeta - q_\sigma^\star\|_{\sup}} + \|\mathcal{Q}\zeta - q_\sigma^\star\|_{\sup}.$$

$\square$

**Lemma C.3.** *Let $\tau > 0$, $p \in [1, \infty)$, $q, q' \in M_b(\mathsf{X} \times \mathsf{A})$, and $\zeta, \zeta' \in \overline{\mathsf{K}}^p(\mathsf{X} \times \mathsf{A}, \mathscr{P}(\mathbb{R}))$. Then,*

$$\overline{d}_p(\mathfrak{T}_\tau^{\mathcal{G}_\tau q}\zeta, \mathfrak{T}_\tau^{\mathcal{G}_\tau q'}\zeta')$$
$$\le \gamma \overline{d}_p(\zeta, \zeta')$$
$$+ 2\gamma \sup_{(x,y,b)\in\mathsf{X}\times\mathsf{X}\times\mathsf{A}} \left[\|\mathtt{id}\|_{L^p(\zeta_{y,b})}\|\pi_x^q - \pi_x^{q'}\|_{\mathrm{TV}}^{1/p} + c_{q,q'}\|\pi_x^q - \pi_x^{q'}\|_{\mathrm{TV}} + 2\|q - q'\|_{\sup}\right],$$

*where $c_{q,q'} := \min\{\|q\|_{\sup}, \|q'\|_{\sup}\}$.*

*Proof.* Observe

$$d_p(\mathfrak{T}_\tau^{\mathcal{G}_\tau q}\zeta, \mathfrak{T}_\tau^{\mathcal{G}_\tau q'}\zeta') \le d_p(\mathfrak{T}_\tau^{\mathcal{G}_\tau q}\zeta, \mathfrak{T}_\tau^{\mathcal{G}_\tau q'}\zeta) + d_p(\mathfrak{T}_\tau^{\mathcal{G}_\tau q'}\zeta, \mathfrak{T}_\tau^{\mathcal{G}_\tau q'}\zeta')$$
$$\le d_p(\mathfrak{T}_\tau^{\mathcal{G}_\tau q}\zeta, \mathfrak{T}_\tau^{\mathcal{G}_\tau q'}\zeta) + \gamma d_p(\zeta, \zeta').$$

So by Lemma C.4, we conclude. $\square$

**Lemma C.4.** *Let $\zeta \in \mathsf{K}(\mathsf{X} \times \mathsf{A}, \mathscr{P}(\mathbb{R}))$ and $q, q' \in M_b(\mathsf{X} \times \mathsf{A})$. For any $\tau > 0$, defining $\pi^\bullet = \mathcal{G}_\tau\bullet$ for $\bullet \in \{q, q'\}$, denoting $c_{q,q'} = \min\{\|q'\|_{\sup}, \|q\|_{\sup}\}$, we have*

$$\overline{d}_p(\mathfrak{T}_\tau^{\mathcal{G}_\tau q}\zeta, \mathfrak{T}_\tau^{\mathcal{G}_\tau q'}\zeta)$$
$$\le 2\gamma \sup_{(x,y,b)\in\mathsf{X}\times\mathsf{X}\times\mathsf{A}} \left[\|\mathtt{id}\|_{L^p(\zeta_{y,b})}\|\pi_x^q - \pi_x^{q'}\|_{\mathrm{TV}}^{1/p} + c_{q,q'}\|\pi_x^q - \pi_x^{q'}\|_{\mathrm{TV}} + 2\|q - q'\|_{\sup}\right]$$

*Proof.* For notational simplicity, we define

$$\xi_x^{\zeta,q} = (\mathtt{proj}^{\mathbb{R}} - \tau\mathtt{kl}[\mathcal{G}_\tau q] \circ \mathtt{proj}^{\mathsf{X}})_{\#}(\zeta_{x,\_} \otimes (\mathcal{G}_\tau q)_x).$$

Then, by the definition of $\mathfrak{T}_\tau^\pi$, we have

$$(\mathfrak{T}_\tau^{\mathcal{G}_\tau q}\zeta)_{x,a} = (\mathtt{b}_{r(x,a),\gamma} \circ \mathtt{proj}^{\mathbb{R}})_{\#}(\xi_\_^{\zeta,q} \otimes P_{x,a})$$

Following, by [40, Theorem 4.8], we have

$$d_p((\mathfrak{T}_\tau^{\mathcal{G}_\tau q}\zeta)_{x,a}, (\mathfrak{T}_\tau^{\mathcal{G}_\tau q'}\zeta)_{x,a}) \le \gamma \int d_p(\xi_\_^{\zeta,q}\xi_\_^{\zeta,q'}) \, \mathrm{d}P_{x,a}.$$

We will now estimate the integrand above. By the definition of $\xi^{\zeta,q}$, for any $x \in \mathsf{X}$, denoting $\pi^q := \mathcal{G}_\tau q$, we have

$$
d_p(\xi_x^{\zeta,q}, \xi_x^{\zeta,q'})
$$

$$
= d_p\left( (\mathtt{b}_{-\tau \mathrm{KL}(\pi_x^q \,\|\, \pi_x^{\mathsf{ref}}),1} \circ \mathtt{proj}^{\mathbb{R}})_{\#}(\zeta_{x,-} \otimes \pi_x^q), (\mathtt{b}_{-\tau \mathrm{KL}(\pi_x^{q'} \,\|\, \pi_x^{\mathsf{ref}}),1} \circ \mathtt{proj}^{\mathbb{R}})_{\#}(\zeta_{x,-} \otimes \pi_x^{q'}) \right)
$$

$$
\leq \underbrace{\inf_{\kappa_x \in \mathscr{C}(\zeta_x \otimes \pi_x^q, \zeta_x \otimes \pi_x^{q'})} \left( \int |z - z'|^p \, d\kappa_x \right)^{1/p}}_{\mathrm{I}(x)} + \underbrace{\tau |\mathrm{KL}(\pi_x^q \,\|\, \pi_x^{\mathsf{ref}}) - \mathrm{KL}(\pi_x^{q'} \,\|\, \pi_x^{\mathsf{ref}})|}_{\mathrm{II}(x)}.
$$

The inequality is due to a technique employed in the proof of Theorem 4.5. Next, by [40, Theorem 6.15], we bound I via

$$
\mathrm{I}(x) \leq 2^{\frac{p-1}{p}} \sup_{x',a'} \|\mathtt{id}\|_{L^p(\zeta_{x',a'})} \|\pi_x^q - \pi_x^{q'}\|_{\mathrm{TV}}^{1/p} \leq 2 \sup_{x',a'} \|\mathtt{id}\|_{L^p(\zeta_{x',a'})} \|\pi_x^q - \pi_x^{q'}\|_{\mathrm{TV}}^{1/p}
$$

Now, for II, we have

$$
\mathrm{II}(x) \leq \min\{\|q'\|_{\sup}, \|q\|_{\sup}\} \|\pi_x^q - \pi_x^{q'}\|_{\mathrm{TV}} + 2\|q - q'\|_{\sup},
$$

as shown in the proof of Lemma C.5. Therefore, we have shown that

$$
d_p((\mathcal{T}_\tau^{\mathcal{G}_\tau q} \zeta)_{x,a}, (\mathcal{T}_\tau^{\mathcal{G}_\tau q'} \zeta)_{x,a})
$$

$$
\leq \gamma \int \left[ \sup_{x',a'} \|\mathtt{id}\|_{L^p(\zeta_{x',a'})} \|\pi_x^q - \pi_x^{q'}\|_{\mathrm{TV}}^{1/p} + c_{q,q'} \|\pi_x^q - \pi_x^{q'}\|_{\mathrm{TV}} + 2\|q - q'\|_{\sup} \right] dP_{x,a}.
$$

$\square$

## C.1 Supplemental Lemmas for Section 4

**Lemma C.5.** *Let* $\bar{\zeta} \in \overline{\mathsf{K}}^1(\mathsf{X} \times \mathsf{A}, \mathscr{P}(\mathbb{R}))$, *and for any* $n \in \mathbb{N}$, *define* $\bar{\zeta}^{n+1} = \mathcal{T}_\tau^\star \bar{\zeta}^n$, *with* $\bar{\zeta}^0 = \mathcal{T}_\tau^\star \zeta$. *Also, define* $\pi^n := \mathcal{G}_\tau \mathcal{Q} \bar{\zeta}^n$. *Then for any* $x \in \mathsf{X}$, *denoting* $C_x := \|\mathcal{Q}\bar{\zeta}(x,\cdot) - q_\tau^\star(x,\cdot)\|_{L^\infty(\pi_x^{\mathsf{ref}})}$,

$$
\tau \left| \mathrm{KL}(\pi_x^n \,\|\, \pi_x^{\mathsf{ref}}) - \mathrm{KL}(\pi_x^\tau \,\|\, \pi_x^{\mathsf{ref}}) \right| \leq (2 + C_1\sqrt{\tau}) \max \left\{ \gamma^n C_x, \sqrt{\gamma^n C_x} \right\},
$$

*where* $C_1 < \infty$ *is a constant. If* $\tau \geq 2\gamma^n C_x$, *then for a constant* $C_2 < \infty$, *we have*

$$
\tau \left| \mathrm{KL}(\pi_x^n \,\|\, \pi_x^{\mathsf{ref}}) - \mathrm{KL}(\pi_x^{\tau,\star} \,\|\, \pi_x^{\mathsf{ref}}) \right| \leq (2 + C_2\tau^{-1}) C_x \gamma^n.
$$

*Proof.* First, observe that

$$
\tau \left| \mathrm{KL}(\pi_x^n \,\|\, \pi_x^{\mathsf{ref}}) - \mathrm{KL}(\pi_x^{\tau,\star} \,\|\, \pi_x^{\mathsf{ref}}) \right|
$$

$$
= \left| \int \mathcal{Q}\bar{\zeta}^n(x,a) \, d\pi_x^n(a) - \mathcal{V}_\tau \mathcal{Q}\bar{\zeta}^n(x) - \int q_\tau^\star(x,a) \, d\pi_x^{\tau,\star}(a) + \mathcal{V}_\tau q_\tau^\star(x) \right|
$$

$$
\leq \left| \int \mathcal{Q}\bar{\zeta}^n(x,a) \, d\pi_x^n(a) - \int q_\tau^\star(x,a) \, d\pi_x^{\tau,\star}(a) \right| + \left| \mathcal{V}_\tau \mathcal{Q}\bar{\zeta}^n(x) - \mathcal{V}_\tau q_\tau^\star(x) \right|
$$

$$
\leq \|q_\tau^\star(x,\cdot)\|_{L^\infty(\pi^{\mathsf{ref}})} \|\pi_x^n - \pi_x^{\tau,\star}\|_{\mathrm{TV}} + 2\|\mathcal{Q}\bar{\zeta}^n(x,\cdot) - q_\tau^\star(x,\cdot)\|_{L^\infty(\pi^{\mathsf{ref}})}
$$

By Lemma 4.4 and the $\gamma$-contractivity of $\mathcal{B}_\tau^\star$, we note that

$$
\|\mathcal{Q}\bar{\zeta}^n(x,\cdot) - q_\tau^\star(x,\cdot)\|_{L^\infty(\pi^{\mathsf{ref}}} \leq \gamma^n \|\mathcal{Q}\bar{\zeta}^0(x,\cdot) - q_\tau^\star(x,\cdot)\|_{L^\infty(\pi^{\mathsf{ref}}}.
$$

Then, by Theorem 3.6, we have

$$
\|\pi_x^n - \pi_x^{\tau,\star}\|_{\mathrm{TV}} \leq \begin{cases} \frac{2e-3}{4} \gamma^n \tau^{-1} \|\mathcal{Q}\bar{\zeta}(x,\cdot) - q_\tau^\star(x,\cdot)\|_{L^\infty(\pi_x^{\mathsf{ref}})} & \|\mathcal{Q}\bar{\zeta}(x,\cdot) - q_\tau^\star\|_{L^\infty(\pi_x^{\mathsf{ref}})} \leq \frac{1}{2}\gamma^{-n}\tau \\ \sqrt{\tau^{-1}\gamma^n \|\mathcal{Q}\bar{\zeta} - q_\tau^\star\|_{L^\infty(\pi_x^{\mathsf{ref}})}} & \text{otherwise.} \end{cases}
$$

Note that $\|q_\tau^\star\|_{\sup} \leq \|r\|_{\sup}/(1-\gamma)$. Indeed, the upper bound is free; the lower bound comes from comparing $q_\tau^\star$ with $q_\tau^\pi$ for $\pi = \pi^{\mathsf{ref}}$. Altogether, we have that

$$
\tau |\mathrm{KL}(\pi_x^n \,\|\, \pi_x^{\mathsf{ref}}) - \mathrm{KL}(\pi_x^{\tau,\star} \,\|\, \pi_x^{\mathsf{ref}})| \leq \left( 2 + \frac{\|r\|_{\sup}\tau^{-1/2}}{1-\gamma} \right) \max \left\{ \gamma^n C_x, \sqrt{\gamma^n C_x} \right\}
$$

for $C_x = \|\mathcal{Q}\bar{\zeta}(x, \cdot) - q_\tau^\star(x, \cdot)\|_{L^\infty(\pi_x^{\mathrm{ref}})}$.

If $\tau \geq 2\gamma^n C_x$, then we have the stronger bound

$$\tau|\mathrm{KL}(\pi_x^n \,\|\, \pi_x^{\mathrm{ref}}) - \mathrm{KL}(\pi_x^{\tau,\star} \,\|\, \pi_x^{\mathrm{ref}})| \leq (2 + C'\tau^{-1})C_x\gamma^n,$$

where $C' = (2e - 3)\|r\|_{\mathrm{sup}}/4(1 - \gamma)$. $\qquad\qquad\qquad\qquad\qquad\qquad\qquad\qquad\square$

**Lemma C.6.** *Let $\bar{\zeta} \in \overline{\mathsf{K}}^p(\mathsf{X} \times \mathsf{A}, \mathscr{P}(\mathbb{R}))$. For any $n \in N$, define $\bar{\zeta}^{n+1} = \mathfrak{T}_\tau^\star \bar{\zeta}^n$, with $\bar{\zeta}^0 = \mathfrak{T}_\tau^\star \bar{\zeta}$. Denoting by $\mathscr{C}(\rho_1, \rho_2)$ the space of all couplings between the measures $\rho_1, \rho_2$, for all $x \in \mathsf{X}$ we have*

$$\inf_{\kappa \in \mathscr{C}(\bar{\zeta}_{x,-}^{\tau,\star} \otimes \pi_x^n, \bar{\zeta}_{x,-}^{\tau,\star} \otimes \pi_x^{\tau,\star})} \int |z - z'|^p \,\mathrm{d}\kappa \leq C_p \frac{\gamma^{n/2}}{(1 - \gamma)^p} \sqrt{\tau^{-1}\|\mathcal{Q}\bar{\zeta}(x, \cdot) - q_\tau^\star(x, \cdot)\|_{L^\infty(\pi_x^{\mathrm{ref}})}},$$

*where $\pi^n := \mathcal{G}_\tau \mathcal{Q}\bar{\zeta}^n$ and $C_p < \infty$ is a constant depending only on $p$ and $\|r\|_{\mathrm{sup}}$. Moreover, when $n > \log\gamma^{-1}(\log 2\|\mathcal{Q}\bar{\zeta}(x, \cdot) - q_\tau^\star(x, \cdot)\|_{L^\infty(\pi_x^{\mathrm{ref}})} - \log\tau)$, we have*

$$\inf_{\kappa \in \mathscr{C}(\bar{\zeta}_{x,-}^{\tau,\star} \otimes \pi_x^n, \bar{\zeta}_{x,-}^{\tau,\star} \otimes \pi_x^{\tau,\star})} \int |z - z'|^p \,\mathrm{d}\kappa \leq C_p' \frac{\gamma^n}{(1 - \gamma)^p} \tau^{-1}\|\mathcal{Q}\bar{\zeta}(x, \cdot) - q_\tau^\star(x, \cdot)\|_{L^\infty(\pi_x^{\mathrm{ref}})},$$

*where $C_p' = (2e - 3)C_p/4$.*

*Proof.* For notational convenience, define $\varpi_x^\bullet := \bar{\zeta}_{x,-}^{\tau,\star} \otimes \pi_x^\bullet$, for $\bullet \in \{n, (\tau, \star)\}$. Then,

$$
\begin{aligned}
\mathrm{W}_n^p := \inf_{\kappa \in \mathscr{C}(\varpi_x^n, \varpi_x^{\tau,\star})} \int |z - z'|^p \,\mathrm{d}\kappa &= d_p^p(\varpi_x^n, \varpi_x^{\tau,\star}) \\
&\overset{(a)}{\leq} 2^{p-1} \int |z|^p \,\mathrm{d}|\varpi_x^n - \varpi_x^{\tau,\star}| \\
&= 2^{p-1} \int \left[ \int |z|^p \,\mathrm{d}\bar{\zeta}_{x,a}^{\tau,\star}(z) \right] \mathrm{d}|\pi_x^n - \pi_x^{\tau,\star}|(a) \\
&\overset{(b)}{\leq} \frac{3^{2p-1}\|r\|_{\mathrm{sup}}^p}{(1 - \gamma)^p} \|\pi_x^n - \pi_x^{\tau,\star}\|_{\mathrm{TV}},
\end{aligned}
$$

where $(a)$ applies [40, Theorem 6.15], and $(b)$ uses that the support of $\bar{\zeta}^{\tau,\star}$ is contained in a ball of radius $3\|r\|_{\mathrm{sup}}/(1 - \gamma)$. By Lemma B.6, it follows that

$$
\begin{aligned}
\mathrm{W}_n^p &\leq \sqrt{\tau^{-1}\|\mathcal{Q}\bar{\zeta}^n(x, \cdot) - q_\tau^\star(x, \cdot)\|_{L^\infty(\pi_x^{\mathrm{ref}})}} \\
&\leq C_p \frac{\gamma^{n/2}}{(1 - \gamma)^p} \sqrt{\tau^{-1}\|\mathcal{Q}\bar{\zeta}(x, \cdot) - q_\tau^\star(x, \cdot)\|_{L^\infty(\pi_x^{\mathrm{ref}})}},
\end{aligned}
$$

where the last inequality holds by Lemma 4.4 and the $\gamma$-contractivity of $\mathcal{B}_\tau^\star$ with $C_p := 3^{2p-1}\|r\|_{\mathrm{sup}}^p$.

Moreover, if $n > \log\gamma^{-1}(\log 2\|\mathcal{Q}\bar{\zeta}(x, \cdot) - q_\tau^\star(x, \cdot)\|_{L^\infty(\pi_x^{\mathrm{ref}})} - \log\tau)$, then

$$\|\mathcal{Q}\bar{\zeta}^n(x, \cdot) - q_\tau^\star(x, \cdot)\|_{L^\infty(\pi_x^{\mathrm{ref}})} < \tau/2$$

for each $x \in \mathsf{X}$, so by Theorem 3.6,

$$\mathrm{W}_n^p \leq \frac{2e - 3}{4}C_p \frac{\gamma^n}{(1 - \gamma)^p} \tau^{-1}\|\mathcal{Q}\bar{\zeta}(x, \cdot) - q_\tau^\star(x, \cdot)\|_{L^\infty(\pi_x^{\mathrm{ref}})}.$$

$\qquad\qquad\qquad\qquad\qquad\qquad\qquad\qquad\qquad\qquad\qquad\qquad\qquad\qquad\qquad\qquad\qquad\square$

# D  Comparison between vanishing temperature limits of ERL with and without temperature decoupling

In this section, we compare and contrast the properties of vanishing temperature limits of standard ERL (assuming they exist) with those achieved by the temperature decoupling gambit. As we showed in Theorem 2.3 and Theorem 3.9, both schemes achieve reference-optimality in the limit; yet, their

limits may be notably distinct according to criteria beyond the RL objective, as we saw in Sections 3.1 and 4.3.

In the remainder of this section, we will define $\zeta^{\mathrm{ref},\star} := \zeta^{\pi^{\mathrm{ref},\star}}$ as the return distribution function corresponding to the limiting temperature-decoupled policy, and $\zeta^{\mathrm{ERL},\star} := \zeta^{\pi^{\mathrm{ERL},\star}}$ as the return distribution function corresponding to the limiting ERL policy $\pi^{\mathrm{ERL},\star}$, assuming such a limit exists.

A very nice property of $\pi^{\mathrm{ref},\star}$ is that it is easy to characterize as the *optimality-filtered reference*, as per Definition 3.8. In particular, $\pi^{\mathrm{ref},\star}$ is characterized *entirely* in terms of the optimal action-value function $q^\star$ and the reference policy $\pi^{\mathrm{ref}}$. On the other hand, as we see explicitly in Section 3.1, $\pi^{\mathrm{ERL},\star}$ *does not* have such a simple characterization: it is influenced also by the transition dynamics of the MDP (as well as the $q^\star$ and $\pi^{\mathrm{ref}}$).

A notable consequence of this fact is that one can reason about $\pi^{\mathrm{ref},\star}$ generically across MDPs, which is not the case for $\pi^{\mathrm{ERL},\star}$. For instance, in *any* MDP, if $\pi^{\mathrm{ref}}$ is the uniform policy, $\pi^{\mathrm{ref},\star}$ is the uniform policy on optimal actions. Thus, one can say definitively that all actions leading to optimal behavior are played equally under $\pi^{\mathrm{ref},\star}$. But this is not true of $\pi^{\mathrm{ERL},\star}$; in general, it is difficult to characterize exactly how $\pi^{\mathrm{ERL},\star}$ behaves: among a set of MDPs with equal $q^\star$, the corresponding $\pi^{\mathrm{ERL},\star}$ can vary significantly.

Similarly, this property of $\pi^{\mathrm{ref},\star}$ enables one to easily influence the optimal policy that is achieved via temperature decoupling by intervening on $\pi^{\mathrm{ref}}$. Again, this is possible due to the simple characterization of $\pi^{\mathrm{ref},\star}$ as the optimality-filtered reference. Suppose, for example, there exists a particular action $a^{\mathrm{scary}}$ that you want to avoid whenever possible (e.g., certain controversial phrases in language generation). It may be undesirable to filter this action out completely (say, by choosing $\pi^{\mathrm{ref}}$ to never play $a^{\mathrm{scary}}$), because perhaps from some states this action is necessary to achieve optimal return. Instead, with temperature-decoupling, you can choose $\pi^{\mathrm{ref}}$ to play this action with very low probability (e.g., $\pi_x^{\mathrm{ref}}(a^{\mathrm{scary}}) = p_{\mathrm{ref}}$ for each $x$). By Theorem 3.9, $a^{\mathrm{scary}}$ will only ever be played when it achieves optimal returns, and moreover, as long as other actions exist that achieve optimal returns, $a^{\mathrm{scary}}$ will be played with much lower probability.

The same logic *does not* hold, in general, for $\pi^{\mathrm{ERL},\star}$. As we saw in Section 3.1, $\pi^{\mathrm{ERL},\star}$ may continue to play $a^{\mathrm{scary}}$ with high probability even if $\pi^{\mathrm{ref}}$ plays it with low probability. Suppose, for instance, that after playing $a^{\mathrm{scary}}$ in state $x$, it is optimal to play $\pi^{\mathrm{ref}}$ subsequently for the rest of the episode. Then $\pi^{\mathrm{ERL},\star}$ may strongly prefer to play $a^{\mathrm{scary}}$ from state $x$, even if other actions can achieve the same expected return. In fact, depending on the transition kernel, the scale of the rewards, and the discount factor, $\pi^{\mathrm{ERL},\star}$ may play $a^{\mathrm{scary}}$ from state $x$ with arbitrarily high probability.

