# OpenReview forum: "Convergence Theorems for Entropy-Regularized and Distributional Reinforcement Learning"
_NeurIPS.cc/2025/Conference — NeurIPS 2025 poster_

### Official Review · Reviewer_79Aa · 2025-06-24

**Clarity:** 3
**Significance:** 2
**Originality:** 3
**Rating:** 4
**Confidence:** 3

**Summary:**

This study focuses on limit theorems of the entropy-regularized RL with a vanishing temperature, including the convergence and (reference-)optimality, and then considers the return distribution convergence. They extend the existing analysis in entropy-regularized RL to the distributional context, partially connecting the theoretical results in distributional RL. They proposed a novel temperature decoupling mechanism to facilitate the proof.

**Questions:**

I read some parts of the appendix, which seems rigorous. Please see my concerns in the Weakness part above.

**Ethical Concerns:**

["NO or VERY MINOR ethics concerns only"]

**Final Justification:**

The theoretical results seem to be a potential contribution, but overall I feel this is still a borderline paper without a clear conclusion among the provided theoretical results. I keep my rating for now, but I become open-minded in the following discussion.

**Limitations:**

NAN

**Quality:**

3

**Strengths And Weaknesses:**

## Strength

1. The proof and writing are rigorous, aligning with an RL theory requirement.

2. Albeit simple, some numerical demonstrations are given in some sections, which are useful even in a theory-oriented paper.


## Weakness

1. The title is too broad. It focuses on the convergence analysis instead of the asymptotic analysis in classical statistics, such as sample efficiency, which also plays a key role in limit theorems. It is suggested to change the title to emphasize the contribution more precisely.

2. The theoretical contribution should be posited better. The convergence analysis of entropy-regularized RL is relatively mature. For example, [33] provides the faster convergence rate even though it is a tabular MDP. It seems that this study extends [33] to continuous MDP qualitatively without giving a rate, but it extends to the distributional RL setting, is that correct? Compared with [34], the conclusion of this study may not be novel, but the analysis is slightly general. Is my understanding correct? Since RL theory literature is relatively packed, it is very important to highlight the contribution among the existing literature.

3. In the introduction part, some descriptions are ambiguous or not clear. For example, I find it difficult to understand the ‘expected-value optimal policies.’ At least, it is not commonly used in distributional RL literature. The conclusion that entropy regularized RL ensures a unique optimal policy needs some reference.

4. This paper concerns a continuous MDP beyond a tabular MDP, but it seems that we need to require a continuous action space in the analysis. Is there a fundamental gap between the paper’s result with the continuous MDP with a discrete action, or is it also able to extend to the discrete action space?

5. I understand this study tries to be rigorous, but it reduces the readability among potential readers. Maybe it is more suggested to focus on the necessary notations to elaborate the conclusions instead of giving too many notations, e.g., the measure-theoretic language in Section 2.

6. One point that makes me feel disappointed is the convergence for control in distributional RL, which is viewed as a very hard problem. However, I only find the convergence in an entropy-regularized distributional RL instead of distributional RL directly. I think there is still a fundamental gap. Correct me if there is a misunderstanding.

7. The authors motivate the temperature decoupling mechanism given the TV distance is not vanishing in an annealing temperature. Is that consistent among different choices of distances? How do we understand this result? In addition, the authors are suggested to highlight again the motivation of this new mechanism in terms of the analysis. Also, the numerical results in Section 3.1 only suggest the learned policy under this mechanism is different. Are they also able to verify the motivation of this mechanism in some sense? I struggle to interpret Section 3.1 with the motivation of this new technique.

8. A general comment is whether the established results can be useful to guide practical algorithms? If so, the contribution may be more broad.

---

> ### Author Rebuttal · Authors · 2025-07-31
>
> We thank the reviewer for their very insightful feedback, and for appreciating the mathematical rigor and the pedagogy of our numerical demonstrations.
>
> Below, we speak to the remaining points that you have raised.
>
> > The title is too broad.
>
> Thank you for pointing this out, we would appreciate any suggestions you have and we’d be happy to change the title. If you have objections to “Convergence Theorems in Entropy-Regularized and Distributional Reinforcement Learning”, please let us know.
>
> > [...] [33] provides the faster convergence rate even though it is a tabular MDP. It seems that this study extends [33] to continuous MDP qualitatively without giving a rate, but it extends to the distributional RL setting, is that correct?
>
> In [33], the authors provide fast convergence rates of policies to an entropy-regularized optimal policy in a tabular MDP with fixed (nonzero) temperature. You correctly identify that our work extends such a result to general MDPs and additionally to convergence of return distributions, albeit with a slower rate. While our rate is slower, it is for the convergence of fundamentally different objects, so not quite comparable.
>
> Beyond this, our work asserts convergence of a scheme for approximating optimal policies (and return distributions) in the vanishing-temperature limit. [33] only has such a result for policy convergence in the bandit setting, while our results hold in much more generality.
>
> > Compared with [34], the conclusion of this study may not be novel, but the analysis is slightly general. Is my understanding correct?
>
> This is incorrect. [34] provides policy convergence in a vanishing temperature limit in a different problem setting: they add an alternative form of (non-vanishing) regularization towards a prior distribution over neural network weights. Moreover, [34] also does not consider convergence of return distributions, in contrast to our work. Our work, on the other hand, establishes convergence to a classically optimal policy in the vanishing temperature limit, as well as the return distributions for this policy.
>
> > In the introduction part, some descriptions are ambiguous or not clear. For example, I find it difficult to understand the ‘expected-value optimal policies.’ [...]
>
> Thank you for pointing this out, we agree that certain terms in the introduction were perhaps too technical before the context had been set. We will fix these. For reference, “expected-value optimal policies” refers to policies that achieve optimal expected returns (e.g., the standard risk-neutral definition of policy optimality).
>
> > The conclusion that entropy regularized RL ensures a unique optimal policy needs some reference.
>
> We are unaware of any reference that rigorously proves the existence and uniqueness of optimal policies in ERL (for fixed nonzero temperature) outside of the discrete action space setting. As a result, we have proved the existence and uniqueness of optimal policies in ERL in general MDPs (see Theorem 2.2). If you can point us to a reference that already does this, we are happy to cite it.
>
> > [...] it seems that we need to require a continuous action space in the analysis. Is there a fundamental gap between the paper’s result with the continuous MDP with a discrete action, or is it also able to extend to the discrete action space?
>
> None of our results require a continuous state or action space. They all hold for both discrete/continuous state spaces as well as discrete/continuous action spaces (unless specifically mentioned).
>
> > [...] Maybe it is more suggested to focus on the necessary notations to elaborate the conclusions instead of giving too many notations, e.g., the measure-theoretic language in Section 2.
>
> We have endeavored to only put notation in the main body of the paper that is needed to understand the statements we make in the main body of the paper. If there is something that can be moved to the appendix without making it so that the reader has to go to the appendix in order to read something in the main body, we are happy to move it. Please let us know if you have found something like this. Our measure-theoretic language is essential for understanding the definitions and statements of Sec 4. We use it to recast more familiar, classical objects in RL in Sec 2 in order to give the reader some time to become acquainted with it.
>
> > [In] the convergence for control in distributional RL, which is viewed as a very hard problem[,...] I only find the convergence in an entropy-regularized distributional RL instead of distributional RL directly. [...] Correct me if there is a misunderstanding.
>
> Your understanding is correct. That said, our approach leveraging vanishing-temperature limits of distributional ERL is still a major step towards solving the long-standing open problem in distributional RL: achieving consistent and stable estimates of the performance of a policy undergoing optimization. To clarify:
>
> 1. Thm 4.5 establishes (with rates) the convergence of an exact entropy-regularized distributional optimality operator at fixed nonzero temperature. This is indeed not the classic RL setting.
>
> 2. Thm 4.7, leveraging Thm 4.5, considers an approximate operator to qualitatively allow one to approximate the (reference-)optimal return distribution (in the standard sense, without entropy regularization) to an arbitrary tolerance.
>
> Under existing distributional dynamic programming methods, one cannot hope for anything more than an approximation either (since the distributional Bellman operator can be applied at most finitely-many times). Beyond that, the classic distributional optimality operator does not produce convergent iterates. So it cannot be used to approximate any particular optimal return distribution function.
>
> > The authors motivate the temperature decoupling mechanism given the TV distance is not vanishing in an annealing temperature. Is that consistent among different choices of distances? How do we understand this result?
>
> Our focus on the TV distance is tied to our proofs regarding return distributions, wherein we find that policy convergence in TV yields return distribution convergence (see the proof of Thm 3.13, inequality (c), for example). In general MDPs with continuous states and actions, the convergence of entropy-regularized optimal policies under any metric is an open problem.
>
> > the authors are suggested to highlight again the motivation of [the temperature-decoupling mechanism] in terms of the analysis.
>
> Thank you, we will do this for sure. We will, for example, move Eqn (B.1) and surrounding text to the main body of the paper, in our revision.
>
> > [T]he numerical results in Section 3.1 only suggest the learned policy under this mechanism is different. Are they also able to verify the motivation of this mechanism in some sense? I struggle to interpret Section 3.1 with the motivation of this new technique.
>
> Indeed, we can explicitly provide closed forms of the limiting policies in standard ERL and the temperature-decoupling mechanism, and they line up precisely with those depicted in Figure 3.1. The derivations aren’t difficult to compute, though they are tedious and will not fit here. Please let us know if you’d like us to provide derivations in a subsequent comment.
>
> This demonstration illustrates the diversity of the limiting temperature-decoupled policy. As predicted, the temperature-decoupled limit is uniform on the set of optimal actions. In contrast, the limiting standard ERL policy (which we can reason about since this is a tabular MDP) carries over influence from the entropy regularizer: it effectively ignores the green action from the root state, despite the fact that it can achieve optimal return for the un-regularized objective.
>
> > A general comment is whether the established results can be useful to guide practical algorithms? If so, the contribution may be more broad.
>
> Please see our response to Reviewer 4Nx8 wherein we discuss some ways in which our results can inform practical algorithms.

---

> > ### Comment · Reviewer_79Aa · 2025-08-05
> >
> > Thank the authors for their response, which is helpful. The theoretical results seem to be a potential contribution, but overall I feel this is still a borderline paper without a clear conclusion among the provided theoretical results. I keep my rating for now, but I become open-minded in the following discussion.

---

### Official Review · Reviewer_4Nx8 · 2025-06-26

**Clarity:** 3
**Significance:** 2
**Originality:** 2
**Rating:** 3
**Confidence:** 3

**Summary:**

In generic MDPs, there can be more than one optimal policy exist and not all of them behave as expected. One of the current methods that handle this issue is entropy-regularized RL which enforces the uniqueness of the optimal policy. This paper studies the evolution of ERL algorithms as temperature vanishes beyond tabular MDPs.

**Questions:**

* Why it is difficult to understand the behavior of ERL methods beyond the tabular setting? In other words, what makes continuous MDPs fundamentally different from tabular MDPs in the context of ERL? Can you give an insight to it without using any math?

* Can you provide some more practical examples such as MuJoCo and Atari, other than the toy problems presented in the paper?

**Ethical Concerns:**

["NO or VERY MINOR ethics concerns only"]

**Final Justification:**

It is solely an RL theory paper. The theoretical results look good to me, but I am not an expert in theory and may have missed something. The empirical results are rather limited, as pointed out by others, too. Overall I am neutral on this work and would leave the decision to other reviewers and the AC.

**Limitations:**

Yes, they are discussed.

**Paper Formatting Concerns:**

None.

**Quality:**

2

**Strengths And Weaknesses:**

**Strengths**

* The problem studied in this paper is well-motivated from the theoretical perspective.

* The paper is written in a clear and mathematically rigorous way.

* Related works are adequately discussed.

**Weaknesses**

* As claimed in the introduction, this paper aims to study ERL beyond tabular MDPs. However, both numerical examples provided in section 3.1 and 4.3 are tabular with only a couple of states and actions.

* It is less intuitive to relate the theoretical analysis presented in this draft to the practical problems one may encounter in practice. While I acknowledge the full rigor the authors made throughout the entire paper, the practical significance of this work is not very clear to me.

---

> ### Author Rebuttal · Authors · 2025-07-31
>
> We thank the reviewer for their feedback, and for their appreciation of the mathematical rigor and the motivation of our work.
>
> Below, we speak to the remaining points that you have raised.
>
> > It is less intuitive to relate the theoretical analysis presented in this draft to the practical problems one may encounter in practice. While I acknowledge the full rigor the authors made throughout the entire paper, the practical significance of this work is not very clear to me.
>
> While our paper’s primary focus is theory, our results do shed light on the behavior of existing algorithms and open the door to new algorithmic concepts that can be applied to, e.g., deep RL.
>
> Many practitioners use ERL as a means to approximate a classically optimal policy. But we show that, in reality, ERL can only recover a reference-optimal policy, which could be rather far from optimal in the classic sense.
>
> Moreover, safety is an increasingly important consideration in the design of autonomous agents. Hence, understanding the distribution over outcomes under policies is critical. Our techniques inform ways of ensuring the stability of distributional return estimates over the course of policy improvement for the first time, which can be crucial in downstream safety approaches.
>
> Finally, here are two ways in which our techniques might be applied:
>
> 1. Consider Soft Q-Learning where the soft Q-function is estimated with a temperature schedule $\sigma(\tau\_t)$ (say, with $\sigma(\tau)=\tau^2$) using standard methods, but when interacting with the environment, the softmax policy with temperature $\tau\_t$ is deployed. This strategy induces more exploration. Furthermore, it's likely that this “action-policy” will converge to $\pi^{\mathrm{ref},\star}$ in the vanishing temperature limit, given our theory.
>
> 1. Consider, e.g., in Soft Actor-Critic, using a temperature decoupled policy as a target policy as a means to further stabilize training.
>
> Actually implementing either of these ideas (or any other idea one might conceive) and the necessary work surrounding any new practical algorithm, of course, is non-trivial. Hence, while we are excited to do this work, we defer it to the future, where it can receive its due care.
>
> > […] this paper aims to study ERL beyond tabular MDPs. However, both numerical examples…are tabular with only a couple of states and actions.
>
> The purpose of our demonstrations was to illustrate how our method differs from standard ERL. The only regime in which ERL is known to produce convergent policies in the vanishing temperature limit is the tabular setting. So the tabular setting is the best case scenario for the “baseline”. Our numerical demonstrations show that, even in this ideal case, there is a notable difference between ERL with and without temperature-decoupling—they produce different policies, and our temperature decoupling method is more robust to numerical precision errors.
>
> Moreover, beyond the tabular setting, there would be a confounding factor in our demonstrations. Any observed discrepancy between the standard ERL policies and the temperature decoupled policies could be the result of the standard ERL policies not converging! Working in the tabular case removes this confounder. So any difference between the ERL policies and the temperature-decoupled policies in our demonstrations is due only to the policy optimization scheme.
>
> > Why is it difficult to understand the behavior of ERL methods beyond the tabular setting […] what makes continuous MDPs fundamentally different from tabular MDPs in the context of ERL?
>
> There are many factors that make ERL more complicated in continuous MDPs than in tabular ones, much like in standard RL. In ERL, we have a new stream of challenges that deal with ensuring that the regularization penalty is bounded. When the action space is continuous and bounded, for example, the uniform reference policy does not guarantee finite regularization penalty anymore. If the set of optimal actions is very sparse, for example, and the density of those actions under the reference is small, the penalty will explode.
>
> In addition, as we show, the quality of the best policy that ERL can approximate (that is, even outside the methods we introduce) in the vanishing temperature limit depends on the “smoothness” of the optimal Q-function when the action space is continuous. For instance, if the true (unregularized) optimal Q-function has large spikes at particular actions (say, the action values are all $0$ except at a particular one), then vanishing temperature limits of ERL cannot recover optimal policies (or in the parlance of our paper, reference-optimality is not the same as optimality). This is a fundamental limit of ERL that had not been known before our work.
>
> > Can you provide some more practical examples such as MuJoCo and Atari, other than the toy problems presented in the paper?
>
> In this submission, we establish a theoretical foundation upon which such work can be done. We view this kind of important, non-trivial empirical work as part of follow-up to our theory-oriented work.

---

> > ### Comment · Reviewer_4Nx8 · 2025-08-03
> >
> > Thank you for the rebuttal, now I have a better sense of your approach. However, I still feel that the experimental results are insufficient. While I understand that non-trivial efforts are needed to implement a new practical algorithm, it is the only and the most convincing evidence that the proposed approach really works in practice. Other than that, the theoretical results in this paper look good and solid.

---

### Official Review · Reviewer_x7BH · 2025-06-26

**Clarity:** 3
**Significance:** 3
**Originality:** 3
**Rating:** 4
**Confidence:** 2

**Summary:**

This paper develops a rigorous theoretical framework for analyzing the limiting behavior of entropy-regularized reinforcement learning as the temperature parameter vanishes. It introduces a novel **temperature decoupling mechanism** that ensures convergence to an interpretable, diversity-preserving optimal policy. The authors further extend their results to **distributional reinforcement learning (DRL)**, presenting the first convergent dynamic programming schemes for estimating return distributions in the control setting.

**Questions:**

1. Can the authors provide any empirical intuition or heuristics for selecting the temperature-decoupling schedule $\sigma(\tau)$? For instance, what are reasonable scaling laws?
2. *In practical RL with neural policies, does the temperature-decoupled policy retain its convergence properties? Have the authors considered the effect of function approximation?
3. How sensitive is the convergence to the choice of $\pi_{ref}$, especially when $\pi_{ref}$ poorly overlaps with the support of the optimal policy? Can regularization adaptive to support mismatch be incorporated?
4. For continuous actions, can the authors elaborate more on the implications of weak convergence? What does this imply for action sampling and training?

**Ethical Concerns:**

["NO or VERY MINOR ethics concerns only"]

**Final Justification:**

The authors have provided detailed explanations addressing my initial concerns, which have resolved them fully. Overall, this is a solid piece of work with strong theoretical results, exploring distributional RL and its convergence to a particular optimal policy. However, I believe the paper would benefit from including experiments to demonstrate the effectiveness of the proposed algorithms, as well as an empirical sensitivity analysis. Therefore, I am maintaining my rating of borderline accept.

**Limitations:**

see weakness and question part

**Quality:**

3

**Strengths And Weaknesses:**

### Strengths
1. The paper rigorously establishes convergence of both value functions and return distributions in the vanishing temperature limit—solving long-standing open questions in DRL.
2. The temperature decoupling idea is theoretically elegant and provides a clear path to interpretable and diverse policy learning.
3. The paper includes complete, detailed, and technically solid derivations that should be reproducible by other researchers.
4. It offers the first convergent algorithm for distributional control, which could significantly influence future work on safe and robust RL.
5. Despite the mathematical density, the flow is clear and well-motivated.

### Weaknesses
1. The toy examples, while illustrative, do not demonstrate practical effectiveness in realistic RL benchmarks.
2. There is no concrete guidance on selecting $\sigma(\tau)$ in practice, nor any adaptive scheme proposed.
3. Convergence of policies in continuous action spaces remains weakly addressed. While weak convergence is claimed, its implications for learning with function approximation are not explored.
4. The framework relies heavily on the referenc policy. Although reference-optimality is well-defined, performance heavily depends on the support and smoothness of the reference policy. This may limit practical deployment in high-dimensional settings.

---

> ### Author Rebuttal · Authors · 2025-07-31
>
> We thank the reviewer for their feedback and appreciation of the quality, elegance, and significance of our work.
>
> Below, we speak to the remaining points that you have raised.
>
> > The toy examples, while illustrative, do not demonstrate practical effectiveness in realistic RL benchmarks.
>
> In this submission, we establish a theoretical foundation upon which such work can be done. We view this kind of important, non-trivial empirical study as a follow-up to our theory-oriented work.
>
> We are certainly excited about the prospects of leveraging our findings to inform the design of new algorithms. Please see, e.g., our response to Reviewer 4Nx8 for some of our ideas regarding this. If you’d like to continue this discussion, please let us know.
>
> > There is no concrete guidance on selecting $\sigma(\tau)$ in practice [...]
>
> As stated in Def 3.10, we simply require $\sigma(\tau)/\tau \to 0$ as $\tau \to 0$. In practice, any choice that obeys this relationship within either a fixed or adaptive annealing scheme is viable. We will be sure to state this explicitly in our revision. Note that in our illustrations, we set $\sigma(\tau) = \tau^2$.
>
> > Can the authors provide any empirical intuition or heuristics for selecting the temperature-decoupling schedule $\sigma(\tau)$? For instance, what are reasonable scaling laws?
>
> Heuristically, the faster $\sigma(\tau)/\tau \to 0$, e.g., $\sigma(\tau) = \tau^3$ rather than $\tau^2$, the closer the resulting policy is to the Boltzmann–Gibbs policy associate to $q^\star_{\rm ref}$ at temperature $\tau$. This "theoretical scaling-law” is made precise in Eqn (B.1). We will be sure to move this equation and surrounding text to the main body of the paper, in our revision.
>
> > While weak convergence is claimed, its implications for learning with function approximation are not explored.
>
> Our focus is to understand the nature of the soft optimal policies themselves, independently of how they are represented. While exploring the implications of weak convergence for learning with function approximation in ERL is certainly an interesting topic, we leave this to future work.
>
> > In practical RL with neural policies, does the temperature-decoupled policy retain its convergence properties? Have the authors considered the effect of function approximation?
>
> Please see above.
>
> > For continuous actions, can the authors elaborate more on the implications of weak convergence? What does this imply for action sampling and training?
>
> One implication of only having weak convergence of the policies is that the associated return distributions may not converge. On the other hand, our results show that we can upgrade TV convergence of policies to convergence of return distributions (see the proof of Thm 3.13, inequality (c), for example).
>
> That said, would you please clarify what you are referring to regarding action sampling and training?
>
> > The framework relies heavily on the reference policy […] performance heavily depends on the support and smoothness of the reference policy.
>
> While you raise important points, the limitations you speak to are in fact fundamental limitations of every ERL method, not just our own. Moreover, we are the first to uncover this truth (see, e.g., Thm 3.2). Indeed, our results show that the best one can hope for in ERL is to approach a reference-optimal policy. (And this is true even when a uniform reference policy is used. Please see the discussion after Def 3.3.)
>
> > How sensitive is the convergence to the choice of $\pi_{{\rm ref}}$, especially when $\pi_{{\rm ref}}$ poorly overlaps with the support of the optimal policy? Can regularization adaptive to support mismatch be incorporated?
>
> If the reference policy does not suitably overlap with some optimal policy, there is little hope for ERL to recover a classically optimal policy, even beyond our novel techniques. (Please see the discussion after Def 3.3.) Furthermore, since ERL agents only ever see actions supported by the reference policy, any scheme that might compensate or correct for a support mismatch would require external information.

---

> > ### Comment · Reviewer_x7BH · 2025-08-04
> >
> > The reviewer thanks the authors for their efforts in preparing the rebuttal. I do not have any further concerns at this point, and I will maintain my current score.

---

### Official Review · Reviewer_4ZKB · 2025-07-02

**Clarity:** 2
**Significance:** 2
**Originality:** 2
**Rating:** 2
**Confidence:** 4

**Summary:**

In this paper, the authors analyze the convergence properties of the optimal policy and the associated return distributions in entropy-regularized reinforcement learning as the regularization temperature approaches zero. They show that, under a temperature decoupling mechanism, the optimal policies converge to a "reference-optimal" policy, and the corresponding return distributions converge to "reference-optimal" return distributions.  They use toy examples to verify their theoretical findings.

**Questions:**

See the Strengths And Weaknesses part above.

**Ethical Concerns:**

["NO or VERY MINOR ethics concerns only"]

**Final Justification:**

After reviewing the authors’ rebuttal, I find that they have not made sufficient efforts to address my concerns or improve the quality of their paper. While the work contains some interesting theoretical findings, I maintain my negative evaluation of the paper.

**Limitations:**

yes

**Quality:**

3

**Strengths And Weaknesses:**

Pros:

* The paper demonstrates the limiting behavior of the optimal policy and the associated return distributions for entropy-regularized RL when the temprature vanishes. The theoretical findings are novel.
* I checked most parts of the proofs. The theoretical analysis is rigorous and sound.

Cons:

* **- Motivation and Contribution:**
   The motivation of the paper is unclear, and the contributions appear limited in scope. On the theoretical side, the proposed temperature decoupling mechanism seems artificial and lacks intuitive justification. The paper does not address the fundamental question of whether convergence can occur in standard entropy-regularized reinforcement learning (without the temperature decoupling mechanism) as the regularization temperature vanishes. On the practical side, the theoretical results seem disconnected from practical applications. Although the authors claim that their theoretical insights lead to a practical algorithm with convergence guarantees, the algorithm described below Theorem 4.7 relies on a fixed ($\tau$, $\sigma$, $n_{\text{opt}}$, $n_{\text{eval}}$), and the resulting policy sequence will not converge to the stated "reference-optimal" policy under the fixed choice of  ($\tau$, $\sigma$, $n_{\text{opt}}$, $n_{\text{eval}}$). Furthermore, Theorem 4.7 only asserts the existence of certain parameters ($\tau$, $\sigma$, $n_{\text{opt}}$, $n_{\text{eval}}$) for which the algorithm performs well, but provides no guidance on how to select them in practice.

  **- Clarity and Presentation:**
   The paper is written in an extremely technical style that makes it difficult to follow. Understanding the theoretical results and verifying the proofs is a struggle due to the dense technical details and uncommon notations. I recommend that the authors adopt more standard notation used in the RL literature and reduce the introduction of new symbols such as $ \mathcal{G}_\tau, \mathcal{J}_\tau, \mathcal{B}_\tau, \mathcal{V}_\tau, \mathcal{Q}, \mathcal{O}, \textup{N}_{\text{ref}}, \odot, \otimes $, and various subscripts/superscripts. Additionally, the authors should highlight their main theoretical results more clearly and consider moving unimportant lemmas and technical details to the appendix.

  **- Empirical Evaluation:**
   The empirical evaluation is limited to simple toy environments, and no experiments are presented to validate the proposed control algorithm. This significantly weakens the practical impact of the paper.

---

> ### Author Rebuttal · Authors · 2025-07-31
>
> We thank the reviewer for their feedback and appreciate their verification of the proofs.
>
> Below, we speak to the remaining points that you have raised.
>
> > The motivation of the paper is unclear, and the contributions appear limited in scope.
>
> Can you please share something concrete about your concerns regarding the motivation and scope that we might address specifically?
>
> > On the theoretical side, the proposed temperature decoupling mechanism seems artificial and lacks intuitive justification.
>
> Can you please explain what you mean by “artificial” in this context? Our temperature decoupling mechanism is justified by its ability to realize a unique interpretable, diversity-preserving (reference-)optimal policy in the vanishing temperature limit. The mechanism itself arises directly from our mathematical analysis of ERL—is this not sufficient?
>
> > The paper does not address the fundamental question of whether convergence can occur in standard entropy-regularized reinforcement learning [...] as the [temperature] vanishes.
>
> We do address this question. Remark 3.6 speaks to its underlying mathematical difficulty. Thm 3.12 (precisely, Eqn (B.1)) establishes an explicit relationship between ERL-optimal policy convergence and ERL-optimal action-value function convergence. And Figure 4.4 shows how standard ERL is numerically unstable at very small temperatures. While we do not close this open problem, we nonetheless provide a clean formula (via temperature decoupling) that realizes a (reference-)optimal policy---the best one can hope for, as we show---that is more interpretable than a potential vanishing temperature limit of ERL-optimal policies.
>
> > On the practical side, the theoretical results seem disconnected from practical applications.
>
> Please see our response to Reviewer 4Nx8 wherein we discuss some ways in which our results can inform practical algorithms.
>
> > Although the authors claim that their theoretical insights lead to a practical algorithm with convergence guarantees, the algorithm described below Theorem 4.7 relies on a fixed $(\tau, \sigma, n_{{\rm opt}}, n_{{\rm eval}})$, and the resulting policy sequence will not converge [...] under the fixed choice of $(\tau, \sigma, n_{{\rm opt}}, n_{{\rm eval}})$. Furthermore, Theorem 4.7 only asserts the existence of certain parameters […] for which the algorithm performs well, but provides no guidance on how to select them in practice.
>
> Thm 4.7 and the discussion following asserts the existence of an approximation scheme. Rather than directly lead to  “a practical algorithm with convergence guarantees”, as you have stated, it “outlines an algorithm for estimating” the return distribution associated with $\pi^{{\mathrm{ref}}, \star}$, as we have stated. Specifically, we prove that by decreasing $\tau$ (and consequently, $\sigma$) and increasing $n_{\rm opt}$ and $n_{\rm eval}$, estimation accuracy increases.
>
> In practice, one can simply run the algorithm suggested by Thm 4.7 with increasing $n_{\rm opt}$ and/or decreasing $\tau$, as we did in Sec 4.3, until the resulting distributions stabilize.
>
> In contrast, before our work, the best one could do is iterate the distributional optimality operator. This is highly unstable (Bellemare et al. 2017). In particular, there is no number $n$ of iterates beyond which we get accurate approximations of a particular return distribution function.
>
> > [...] I recommend that the authors adopt more standard notation used in the RL literature and reduce the introduction of new symbols such as $\mathcal{G}\_\tau, \mathcal{J}\_\tau, \mathcal{B}\_\tau, \mathcal{V}\_\tau, \mathcal{Q}, \mathcal{O}, \mathsf{N}\_{\text{ref}}^\star, \odot, \otimes $, and various subscripts/superscripts.
>
> Whenever possible, we adopt standard notation from distributional RL, particularly with regard to the operators such as $\mathcal{G}_\tau$ (see Distributional Reinforcement Learning by Bellemare, Dabney, Rowland, 2023; Chapter 7). The notation $\mathcal{J}$ is standard for describing the performance of a policy in RL (Sutton and Barto, 2018).
>
> Other symbols and operators, such as $\mathcal{V}_\tau$ and $\mathcal{Q}$, are crucial for lifting ERL operations to distributions; Def 4.3 and Lem 4.4, for example, would be unruly without them. Again, dealing with such issues via the introduction of operators is standard practice in distributional RL (Bellemare, Dabney, Rowland, 2023; Chapter 4).
>
> We concede that we have taken some artistic license with other notation, but only when no universally agreed-upon notation exists (e.g., $\mathcal{O}$ for the set of occupancy measures).
>
> Finally, as a practical matter, our proofs and statements would be much longer without our notational choices like $\otimes$ and $\odot$.
>
> > [T]he authors should highlight their main theoretical results more clearly and consider moving unimportant lemmas and technical details to the appendix.
>
> Would you please point to which of our main theoretical results could be highlighted more clearly?
>
> Moreover, are there any specific lemmas or technical details you think would be better suited for the appendix? If so, we would be happy to move them, if moving them did not disrupt the flow of our paper by, for example, requiring the reader to go to the appendix to be able to read another statement in the main body of the text.
>
> > The empirical evaluation is limited to simple toy environments, and no experiments are presented to validate the proposed control algorithm. [...]
>
> We stress that the purpose of our experiments was to illustrate the different behavior induced by our schemes compared to standard ERL schemes, not for evaluation. Our proof of Thm 4.5 that iterative applications of the soft Bellman optimality operator converge conclusively validates this theoretical algorithm. Likewise, our proofs of Thms 4.6 and 4.7 validate the theoretical algorithms they define. Realizing any of these as a practical, deep RL algorithm, for instance, is important and non-trivial. Hence, while we are excited to do this work, we defer it to the future, where it can receive its due care.

---

> > ### Comment · Reviewer_4ZKB · 2025-08-04
> >
> > Thanks for the responses. Given my original concerns and the response to them, I maintain my initial recommendation.

---

### Note · Authors · 2025-08-11

To conclude our correspondence, we would like to thank you all for your time and effort in reviewing our work as well as your acknowledgement of our contributions and their strengths:

* The paper “[solves] long-standing open questions in DRL”, “the temperature decoupling idea is theoretically elegant”, “the paper includes “complete, detailed, and technically solid derivations that should be reproducible by other researchers”. (Reviewer x7BH)
* “The theoretical findings are novel”, and “the theoretical analysis is rigorous and sound”. (Reviewer 4ZKB)
* The paper is “well-motivated from the theoretical perspective”, “written in a clear and mathematically rigorous way”, and “the theoretical results in this paper look good and solid”. (Reviewer 4Nx8)
* “The proof and writing are rigorous, aligning with an RL Theory requirement”. (Reviewer 79Aa).

In addition, we are particularly excited that you feel "[the results] could significantly influence future work on safe and robust RL".

Finally, in the remaining discussion, we ask that you first consider our work and its merits as a stand-alone theory paper before considering its (deliberately) limited empirics. While we respect and share your desire for evaluation and empirical study, we wish to reiterate that we leave this to future work, given the many ways in which our theory can inform practice, as discussed in our rebuttals.

---

### Decision · Program_Chairs · 2025-09-17

**Decision:**

Accept (poster)

**Comment:**

This paper studies the convergence of entropy regularized distributional RL with a diminishing temperature. The authors provide a comprehensive theoretical characterization of the problem, including the hardness. To ensure convergence, the authors provide a novel  decoupling of the temperature. Namely, in the most naive approach, the same temperature is used for both defining the optimal value function and computing the softmax. The authors instead propose to used two different temperature and one decays significantly faster than the other. This resembles the two-timescale learning rates idea and I find this novel and interesting. Sufficient theoretical analysis is provided to support the effectiveness of this approach.

The reviewers all agree that this is a technically solid paper. The main concerns lie in motivation, in presentation being too technical, and in lack of larger empirical study. I consider this paper as a theoretical paper and believe the current empirical results are sufficient for a theoretical paper. I do agree with the concerns about motivation and presentation being too technical. After calibration, I feel the novel and principled decoupling mechanism is a valuable idea to the community and outweighs the concerns. I, therefore, recommend accept.

That being said, I do have a few suggestions. (1) First, I suggest the authors to significantly tone down the claims. For example, the first few sentences in the abstract read like this paper is going to address a general RL problem. But it turns out that the paper is addressing a very specific problem in distributional RL and the way the paper addresses the problem (the decoupling mechanism) is far away from what practitioners do. It would be better to make the scope and limitation of the work absolutely clear as early as possible. (2) The paper is indeed very technical. Since the decoupling mechanism is the most important contribution, the author may consider demonstrating this contribution first in a finite state action MDP as a warm-up. This can significantly improve the readability of the work and help the reader to understand the main contribution easily.